# Development of the WRF-CO2 4DVar assimilation system v1.0

Tao Zheng[1,4], Nancy H.F. French[2], and Martin Baxter[3]

[1]Department of Geography, Central Michigan University, Mount Pleasant, MI. USA
[2]Michigan Tech Research Institute, Michigan Technological University, Ann Arbor, MI. USA
[3]Department of Earth and Atmospheric Sciences, Central Michigan University, Mount Pleasant, MI. USA
[4]Institute of Great Lakes Research, Central Michigan University, Mount Pleasant, MI. USA

*Correspondence to:* Tao Zheng (zheng1t@cmich.edu)

**Abstract.** Regional atmospheric $CO_2$ inversions commonly use Lagrangian particle trajectory model simulations to calculate the required influence function, which quantifies the sensitivity of a receptor to flux sources. In this paper, an adjoint based four-dimensional variational (4DVar) assimilation system, WRF-CO2 4DVar, is developed to provide an alternative approach. This system is developed based on the Weather Research and Forecasting (WRF) modeling system, including WRF-Chem, WRFPLUS, and WRFDA, all in version 3.6. In WRF-CO2 4DVar, $CO_2$ is modeled as a tracer and its feedback to meteorology is ignored. This configuration allows most WRF physical parameterizations to be used in the assimilation system without incurring a large amount of code development. WRF-CO2 4DVar solves for the optimized $CO_2$ flux scaling factors in a Bayesian framework. Two variational optimization schemes are implemented for the system: the first uses the L-BFGS-B and the second uses the Lanczos conjugate gradient (CG) in an incremental approach. WRFPLUS forward, tangent linear, and adjoint models are modified to include the physical and dynamical processes involved in the atmospheric transport of $CO_2$. The system is tested by simulations over a domain covering the continental United States at 48 km × 48 km grid spacing. The accuracy of the tangent linear and adjoint models are assessed by comparing against finite difference sensitivity. The system's effectiveness for $CO_2$ inverse modeling is tested using pseudo-observation data. The results of the sensitivity and inverse modeling tests demonstrate the potential usefulness of WRF-CO2 4DVar for regional $CO_2$ inversions.

## 1 Introduction

While rising atmospheric $CO_2$ has been well documented by observations, major uncertainties still exist in attributing it to specific processes (Gurney et al., 2002; Peylin et al., 2013). Atmospheric $CO_2$ inversions estimate surface carbon fluxes from atmospheric $CO_2$ measurements. Since the early study by Enting et al. (1995), a large amount of effort has been devoted to developing and applying atmospheric $CO_2$ inversion methods. Most of these inversions are based on a Bayesian framework, and a wide range of different approaches have been used, including: synthesis inversion (Rayner et al., 1999; Bousquet et al., 1999; Peylin et al., 2002; Gurney et al., 2002), geostatisitical estimation (Michalak et al., 2004; Gourdji et al., 2012), Kalman smoother (Bruhwiler et al., 2005), Ensemble Kalman smoother (Peters et al., 2005), and 4d variational inversion (Chevallier et al., 2005; Baker et al., 2010). All of these inversion approaches are optimization systems which yield an optimal estimate of $CO_2$ fluxes by minimizing a Bayesian cost function. Within these optimization systems, the observation vector is formed

by atmospheric $CO_2$ measurements, and the state vector is formed by $CO_2$ fluxes and lateral boundary conditions (only for regional inversion systems). The relationship between $CO_2$ fluxes and atmospheric $CO_2$ is described by the influence function, which is also called the footprint or adjoint sensitivity. Because all of the inversion approaches use a chemistry transport model (CTM) to relate $CO_2$ fluxes to atmospheric $CO_2$, the influence function in theory can be calculated by the CTM using a finite difference method. However the practical limits imposed by computational costs often necessitate the aggregation of flux to reduce the state vector size, which leads to aggregation errors (Bocquet, 2009; Kaminski et al., 2001; Turner and Jacob, 2015). In practice, different inversion systems use different approaches to determine the influnce function. Some inversion systems, including synthesis inverison, geostatistical estimation, and Kalman smoother, require the influence function to be explicitly constructed before the inversion. In comparison, Ensemble Kalman smoother and 4DVar inversion do not require precalculation of the influence function. Precalculation of the influence function is typically carried out using a finite difference approach with the CTM when the state vector is smaller than observation vector, or by the adjoint model of the CTM when the observation vector is smaller than the state vector. While most of the Lagrangian CTM models have their adjoint developed together (Uliasz, 1993; Lin et al., 2003; Stohl et al., 2005; Stein et al., 2015), separate and considerable efforts were often needed to develop and maintain the adjoint for Eulerian CTM models (Hourdin et al., 2006; Meirink et al., 2008). An ensemble Kalman smoother requires neither an adjoint model nor precalculation of the influence function. Instead it creates an ensemble of $CO_2$ flux fields and runs a CTM for each ensemble member. By sampling the ensemble flux fields and their corresponding atmospheric $CO_2$, the ensemble Kalman smoother calculates the Kalman gain matrix without explicitly constructing the influence function (Peters et al., 2005), and it provides posterior flux error estimates. The main disadvantage of the ensemble Kalman smoother is that use of a small number of ensemble members is likely to lead to misrepresentation of the true prior error variance.

Like ensemble Kalman smoothers, 4DVar inversions do not require precalculation of the influence function, but they do require the adjoint of a CTM to calculate the observation cost function. 4DVar inversions use a CTM and prior $CO_2$ fluxes to calculate the simulated $CO_2$, which is compared with observations to obtain the innovative vector. The adjoint of the CTM is then used to calculate the cost function gradient based on the innovative vector. Through iterative minimization of the cost function, 4DVar inversions estimate the optimal posterior fluxes. By avoiding calculation of the influence function, 4DVar inversions enable $CO_2$ fluxes to be estimated at much higher resolution provided that sufficent observations are available. The major disadvantages of 4DVar inversions are: they do not explicitly provide posterior flux error estimates (additional computation is required), and their convergence is not always guaranteed.

4DVar systems have been widely used for $CO_2$ inversion at global and regional scales. Examples of global 4DVar inversions include the following: The off-line transport model Parameterized Chemistry Tracer Model (PCTM) (Kawa et al., 2004) and its adjoint have been used for $CO_2$ inversions (Baker et al., 2010, 2006; Butler et al., 2010; Gurney et al., 2005). Chevallier et al. (2005) developed a 4DVar system based on the LMDZ model (Hourdin et al., 2006) to assimilate $CO_2$ observation data from the Television Infrared Observation Satellite Operational Vertical Sounder (TOVS). This system has also been used to invert

surface $CO_2$ observations (Chevallier, 2007; Chevallier et al., 2010). The TM5 4DVar system (Meirink et al., 2008), based on the TM5 global two-way nested transport model (Krol et al., 2005), is included in the TransCom satellite intercomparison experiment (Saito et al., 2011). TM5 4DVar has also been used to investigate total column $CO_2$ seasonal amplitude (Basu et al., 2011) and to assimilate the Greenhouse Gases Observing Satellite (GOSAT) observations (Basu et al., 2013). Another widely

used inversion system is the GEOS-Chem 4DVar (Henze et al., 2007; Kopacz et al., 2009) with its $CO_2$ module updated by Nassar et al. (2010). GEOS-Chem 4DVar has been used to estimate $CO_2$ fluxes from the Tropospheric Emission Spectrometer (TES) and the GOSAT $CO_2$ observations (Nassar et al., 2011; Deng et al., 2014), and it is also part of JPL's (Jet Propulsion Laboratory) Carbon Monitoring System (Liu et al., 2014)

Regional 4DVar inversion studies have been driven in part by the need to resolve biosphere-atmosphere carbon exchange at smaller scales (Gerbig et al., 2009), and by the need to address policy-relevant objectives, such as assessing emission reduction effectiveness (Ciais et al., 2014) and the impact of regional scale sources like wildland fire (French et al., 2011). For instance, Gerbig et al. (2003) used an analytical approach to minimize the cost function and the STILT (Lin et al., 2003) model driven by meteorological analyses to calculate the influence function. In a later study, STILT driven by the European Center for Medium-

Range Weather Forecasts (ECMWF) meteorological data is used to calculate the influence function to investigate the impacts of vertical mixing error (Gerbig et al., 2008). More recently, Lauvaux et al. (2012) also used an analytical solution for cost function minimization and LPDM (Uliasz, 1993) to compute the influence function. In another study, Pillai et al. (2012) used STILT driven by meteorological data from WRF to calculate the influence function for comparing Lagrangian and Eulerian models for regional $CO_2$ inversions. To improve accuracy, STILT has been coupled to WRF, in which the latter provides online

meteorology to STILT to avoid interpolation error (Nehrkorn et al., 2010). More recently, Alden et al. (2016) investigated biopheric $CO_2$ flux in the Amazon using an analytical inversion approach (Yadav and Michalak, 2013) with the influence function calculated by the STILT and Flexpart (Stohl et al., 2005) models. Also, Chan et al. (2016) applied a regional $CO_2$ inversion in Canada with both analytical and Markov chain Monte Carlo (MCMC) LPDM based approaches. The influence function is also calculated with the Flexpart model in this study.

In this paper, a regional $CO_2$ inversion system with online meteorology, WRF-CO2 4DVar, was developed by modifying the WRFDA and WRFPLUS system (v3.6) in an approach similar to that used for black carbon emission inversion by Guerrette and Henze (2015, 2017) (hereafter GH15/17). WRFDA is a meteorology data assimilation system, which includes a 4DVar assimilation system (Barker et al., 2012; Huang et al., 2009) and related adjoint and tangent linear models (WRFPLUS) (Zhang

et al., 2013). Designed to improve weather forecasts, WRFDA 4DVar optimizes meteorological initial and boundary conditions by assimilating a variety of observational data. WRFPLUS was modified to include $CO_2$ transport processes and the cost function was configured so that the state vector consists of $CO_2$ fluxes instead of meteorological fields. In developing WRFDA-Chem for black carbon inversion, GH15/17 excluded radiation, cumulus, and microphysics parameterization schemes from the tangent linear and adjoint models because developing these procedures for black carbon would incur a large amount of new

code development. In WRF-CO2 4DVar, $CO_2$ is a tracer, meaning its impacts on meteorology are ignored. This configuration

allows inclusion of the full physics schemes in WRF-CO2 4DVar's tangent linear and adjoint models with limited new code development (see Section 2.4.1). As transport model error is detrimental to 4DVar inversion accuracy (Fowler and Lawless, 2016; Gerbig et al., 2009), it is beneficial to use the full physics schemes in the tangent linear and adjoint models for WRF-CO2 4DVar. In addition, while GH15/17 excluded convective transport of chemistry species in WRFDA-Chem, the tangent

linear and adjoint code for this process was developed for WRF-CO2 4DVar to reduce the vertical mixing error (see Section 2.4.3). Two optimization schemes were developed for WRF-CO2 4DVar: an incremental optimization with Lanczos-CG and an L-BFGS-B based optimization.

In the WRF system, $CO_2$ mixing ratio variations could impact meteorology fields through the radiation scheme, which
proivdes temperature tendency to the dynamical core (Iacono et al., 2008; Skamarock et al., 2008). None of the radiation schemes (as of Version 3.6) use the simulated $CO_2$ from the chemistry module, but instead use climatological values. A sensitivity test was conducted to assess the short term impacts of $CO_2$ variation on meteorology. The results show that with $CO_2$ mixing ratio changed from 391 ppm to 500 ppm, the mean difference in horizontal wind (U,V) and air temperature at the end of the 48-hour simulations are $0.0794$ ms$^{-1}$, $0.0791$ ms$^{-1}$, and $0.0366$ K, respectively. Although these differences grow with time,
their magnitude are considerably smaller compared with the contribution from other factors for the short assimilation window (days to weeks) that WRF-CO2 4DVar is designed for. Based on the above analysis, the impact of $CO_2$ on meteorology is ignored in WRF-CO2 4DVar and $CO_2$ is modeled as a passive tracer. This simplification allows WRF-CO2 4DVar to use the full version of most WRF physics schemes in its tangent linear and adjoint models to minimize the linearization error (Tremolet, 2004).

Compared with offline regional inversion systems, WRF-CO2 4DVar has an advantage provided by the close one-way coupling between meteorological processes and chemistry transport. For example, adequately representing $CO_2$ vertical transport and eddy turbulent mixing in high resolution regional simulations is crucial, as vertical motions in the atmosphere exhibit significant temporal variability. Grell et al. (2004) shows that less than 40% of the total vertical velocity variability in a 3 km
resolution simulation is captured by a 1-hour output interval. He estimated that the meteorological output interval must be less than 10 minutes in order to capture more than 85% of the variability in cloud resolving simulations. In WRF-CO2 4DVar, $CO_2$ transport runs at the same time step as the meteorology, avoiding the problems facing its offline counterparts.

The remainder of this paper is organized as follows: Section 2 details the implementation of the two variational optimization
schemes for cost function minimization, and the modification to the tangent linear and adjoint models. Section 3 examines the accuracy of sensitivity calculated by the tangent linear and adjoint models, and the system's effectiveness in inverse modeling. Section 4 describes the treatment of $CO_2$ lateral boundary conditions in the WRF-CO2 4DVar system. Finally, a summary and outlook are presented in Section 5.

## 2   Method

This section describes the WRF-CO2 4DVar cost function configuration and the associated minimization schemes, followed by a description of the forward, tangent linear, and adjoint models.

### 2.1   Cost function configuration

WRF-CO2 4DVar is designed to optimize the $CO_2$ flux by assimilating $CO_2$ observations into an atmospheric chemistry transport model. The $CO_2$ flux is optimized through use of a linear scaling factor:

$$E = k_{co2}\widetilde{E} \tag{1}$$

Where $\widetilde{E}$ is the $CO_2$ flux read from flux files, $k_{co2}$ is the flux scaling factor, and $E$ is the effective $CO_2$ flux. It is the effective flux $E$ that is used in WRF-Chem's emission driver to update $CO_2$ mixing ratio ($q_{co2}$). The flux scaling factor $k_{co2}$, its tangent linear variable $g\_k_{co2}$, and its adjoint variable $a\_k_{co2}$ are used in calculating model sensitivity and minimizing the cost function defined in Eq. (2).

The cost function $J(\mathbf{x})$ of WRF-CO2 4DVar follows the Bayes framework widely used in atmospheric chemistry and numerical weather prediction (NWP) data assimilations:

$$J(\mathbf{x}) = J_b(\mathbf{x}) + J_o(\mathbf{x}) \tag{2}$$

where the background cost function $J_b(\mathbf{x})$ is defined as

$$J_b(\mathbf{x}) = \frac{1}{2}(\mathbf{x}^n - \mathbf{x}^b)^T \mathbf{B}^{-1}(\mathbf{x}^n - \mathbf{x}^b) \tag{3}$$

and the observation cost function $J_o(\mathbf{x})$ is defined as

$$J_o(\mathbf{x}) = \frac{1}{2}\sum_{k=1}^{K}\{H[M(\mathbf{x}^n)] - \mathbf{y}_k\}^T \mathbf{R}^{-1}\{H[M(\mathbf{x}^n)] - \mathbf{y}_k\} \tag{4}$$

In Eqs. (3-4), $\mathbf{B}$ is the background error covariance matrix, $\mathbf{R}$ is the observation error covariance matrix, $M$ is the transport model, and $H$ is the observational operator, $\mathbf{y}_k$ is the observation vector, $\mathbf{x}^b$ is the prior state vector. The superscript $n$ indicates that $\mathbf{x}^n$ is the optimized state vector at the $n^{\text{th}}$ iteration. For a full list of variables used in this paper, please refer to Table 1. Throughout the paper, bold face lower case characters represent vectors and bold face upper case characters represent matrices.

Like other data assimilation systems, WRF-CO2 4DVar is an optimization scheme. Its state vector $\mathbf{x}$ consists of the flux scaling factors $k_{co2}$ and lateral boundary condition scaling factors. The summation $K$ in Eq. (4) indicates the entire assimilation time period is evenly split into $K$ observation windows during which observational data are ingested into the assimilation system. Details about how observations are ingested in the variational optimization schemes are given in Section 2.2 and 2.3

Two optimization schemes are implemented in WRF-CO2 4DVar to minimize the cost function. The first scheme uses a limited memory BFGS minimization algorithm (L-BFGS-B) (Byrd et al., 1995) and the second uses the Lanczos version of the conjugate gradient (Lanczos-CG) (Lanczos, 1950) minimization algorithm. Both schemes are iterative processes, and they call on WRF-CO2 4DVar model components (the forward, tangent linear, and adjoint models) to calculate the model sensitivity

$\partial q_{co2}/\partial k_{co2}$ between the iterations. The two optimization schemes are described in Section 2.2 and 2.3, respectively, and the three model components are described in Section 2.4.

## 2.2   L-BFGS-B optimization

L-BFGS-B (Byrd et al., 1995) is a quasi-Newton method for nonlinear optimization with bound constraints. L-BFGS-B has been used in a number of atmospheric chemistry inverse modeling systems, including the GEOS-Chem adjoint model system

(Henze et al., 2007) and the TM5 4DVar system (Meirink et al., 2008). The diagram in Fig. 1 demonstrates the steps involved in the L-BFGS-B based optimization scheme. The scheme is an iterative process which searches for the optimized $k_{co2}$ by minimizing the cost function defined in Eq. (2-4). Between its iterations, the minimization algorithm L-BFGS-B requires the values of the cost function and its gradient, which are supplied by the forward model and the adjoint model as indicated in Fig. 1.

The calculation of the cost function is carried out based on Eq. (2-4). Starting with the prior estimate of $k_{co2}$, the forward model run generates the CO$_2$ mixing ratio $q_{co2}$, which is transformed from the WRF model space to the observation space by the forward observation operator $H$. This results in the $H(M(\mathbf{x}^n))$ term in Eq. (4), which is then paired with the observation vector $\mathbf{y}_k$ to calculate the innovation vector $\mathbf{d}_k = H(M(\mathbf{x}^n)) - \mathbf{y}_k$. Next, the innovation vector and observation error covariance $\mathbf{R}$ are used to calculate the observation cost function $J_o(\mathbf{x})$ as expressed in Eq. (4). Finally, the background cost function

$J_b(\mathbf{x})$ is calculated according to Eq. (3), and combined with the observation cost function $J_o(\mathbf{x})$ to form the total cost function $J(\mathbf{x})$ according to Eq. (2).

L-BFGS-B requires the values of the cost function $J(\mathbf{x})$ and its gradient $\nabla J(\mathbf{x})$ in searching for the optimized $k_{co2}$. The gradient is calculated using Eq. (5).

$$\nabla J(\mathbf{x}) = \sum_{k=1}^{K} \widetilde{M}^T \widetilde{H}^T R^{-1} \{ H[M(\mathbf{x}^n) - \mathbf{y}_k] \} + B^{-1}(\mathbf{x}^n - \mathbf{x}^b) \tag{5}$$

The first term on the right hand side of Eq. (5) is the observation gradient and the second is the background gradient. The observation gradient is calculated in two steps: (1) The innovation vector is scaled by $\mathbf{R}^{-1}$ and transformed to the WRF model space by the adjoint observation operator, resulting in $\widetilde{H}^T \mathbf{R}^{-1}(H(M(\mathbf{x}^n)) - \mathbf{y}_k)$, which is the adjoint forcing. (2) The adjoint forcing is then ingested by the WRF-CO2 adjoint model during its backward (in time) integration, which yields the

30 observation gradient. Supplied with the values of the cost function and gradient, the L-BFGS-B algorithm finds a new value of $k_{co2}$, which is used for the next iteration. The iterative optimization process continues until a given convergence criterion is met. The L-BFGS-B based optimization in WRF-CO2 4DVar is implemented based on the Fortran code of Algorithm 788

version Lbfgsb.2.1 (Zhu et al., 1997). Version Lbfgsb.3.0 (Luis Morales and Nocedal, 2011) will be implemented in the next model update.

## 2.3 Incremental optimization

The second optimization scheme implemented for WRF-CO2 4DVar is the incremental approach commonly used in NWP data assimilation systems, including ECWMF 4DVar (Rabier et al., 2000) and WRFDA (Barker et al., 2012). A major difference between the L-BFGS-B based optimization and the incremental optimization is that the former optimizes for the state vector while the latter optimizes for the state vector analysis increment. The incremental assimilation scheme uses a linear approximation to transform the observation cost function from what is defined in Eq. (4) to Eq. (6):

$$J_o(\mathbf{x}) = \frac{1}{2} \sum_{k=1}^{K} \{H[M(\mathbf{x}^{n-1})] - \mathbf{y}_k + \widetilde{H}[\widetilde{M}(\mathbf{x}^n - \mathbf{x}^{n-1})]\}^T \mathbf{R}^{-1} \{H[M(\mathbf{x}^{n-1})] - \mathbf{y}_k + \widetilde{H}[\widetilde{M}(\mathbf{x}^n - \mathbf{x}^{n-1})]\} \tag{6}$$

Compared to Eq. (4), Eq. (6) approximates the innovation vector by a sum of two parts. The first part, $H(M(\mathbf{x}^{n-1})) - \mathbf{y}_k$, is the innovation vector from the previous iteration. The second part, $\widetilde{H}(\widetilde{M}(\mathbf{x}^n - \mathbf{x}^{n-1}))$, is the state vector analysis increment $(\mathbf{x}^n - \mathbf{x}^{n-1})$ transformed by the tangent linear model $\widetilde{M}$ and tangent linear observation operator $\widetilde{H}$. With the linear approximation of the cost function the gradient is calculated by

$$\nabla J(\mathbf{x}) = \sum_{k=1}^{K} \widetilde{M}^T \widetilde{H}^T \mathbf{R}^{-1} \{H[M(\mathbf{x}^{n-1}) - \mathbf{y}_k]\} + \mathbf{B}^{-1}(\mathbf{x}^{n-1} - \mathbf{x}^b) +$$
$$\sum_{k=1}^{K} \widetilde{M}^T \widetilde{H}^T \mathbf{R}^{-1} \{\widetilde{H}[\widetilde{M}(\mathbf{x}^n - \mathbf{x}^{n-1})]\} + \mathbf{B}^{-1}(\mathbf{x}^n - \mathbf{x}^{n-1}) \tag{7}$$

In WRF-CO2 4DVar, the incremental optimization is implemented as a double loop in which the outer loop calculates the first and second items on the right hand side of Eq. (7), while the inner loop calculates the third and fourth items. The superscript $n-1$ indicates that $\mathbf{x}^{n-1}$ is the optimized state vector in the last outer loop, and superscript $n$ indicates that $\mathbf{x}^n$ is the optimized state vector in the inner loop. The outer loop first calls the forward model $M$ and adjoint model $\widetilde{M}^T$ to calculate $\widetilde{M}^T \widetilde{H}^T R^{-1}(H(M(\mathbf{x}^{n-1}) - \mathbf{y}_k))$ and $B^{-1}(\mathbf{x}^{n-1} - \mathbf{x}^b)$, which remain unchanged during the subsequent inner loop calculation. The analysis increment $(\mathbf{x}^n - \mathbf{x}^{n-1})$ is optimized in the inner loop, which calls the tangent linear and adjoint models to calculate the third and fourth items of Eq. (7). Inner loop calculation is carried out by Lanczos-CG (Lanczos, 1950), which can optionally estimate eigenvalues of the cost function Hessian matrix $(\nabla^2 J(\mathbf{x}))$. The diagram in Fig. 2 shows the structure of the Lanczos-CG based incremental optimization implemented in WRF-CO2 4DVar.

## 2.4 Forward, tangent linear, and adjoint models

WRFPLUS consists of three model components: the WRF model, its tangent linear model, and its adjoint model (Barker et al., 2012; Huang et al., 2009). The three models are used by WRFDA to optimize the initial meteorological condition in order to improve numerical weather prediction. Unlike WRFDA, WRF-CO2 4DVar is designed to optimize the $CO_2$ flux, instead of the

meteorological initial and boundary conditions. This difference means the physical and dynamical processes involved in the atmospheric $CO_2$ transport are needed in WRF-CO2 4DVar's model components. To include these processes, the chemistry module was added to the forward model. The chemistry module includes chemistry, deposition, photolysis, advection, diffusion, and convective transport of chemistry species (Grell et al., 2005). These processes are included in different modules of

5   WRF-Chem: ARW (Advanced Research WRF) dynamical core, physics driver, and chemistry driver. GHG (Greenhouse Gas) tracer option was removed and $CO_2$ is treated as an inert tracer. In the emission driver, CarbonTracker 2016 version (Peters et al., 2007) replaces the online biogenic $CO_2$ model Vegetation Photosynthesis and Respiration Model (VPRM) (Mahadevan et al., 2008). This change is made because WRF-CO2 4DVar optimizes the $CO_2$ flux instead of online emission model parameters.

### 2.4.1   Variable dependence analysis

The tangent linear and adjoint models of WRFPLUS needed to be modified to include the physical and dynamical processes involved in the atmospheric transport of $CO_2$, so that they will be consistent with the forward model. A thorough variable dependence analysis was conducted and the results are summarized in Table 2, which groups WRF-Chem processes into three

categories regarding $CO_2$ tracer transport. The first category includes the chemistry processes that do not apply to $CO_2$, including gas and aqueous phase chemistry, dry and wet deposition, and photolysis. These processes are simply excluded from the forward, tangent linear, and adjoint models in WRF-CO2 4DVar.

The second category is comprised of the physical parameterizations that do not provide $CO_2$ tendency, but provide mete-

orological tendency. This category includes radiation, surface, cumulus, and microphysics parameterizations. While the full physics schemes of surface, cumulus, planetary boundary layer (PBL), and microphysics are used in the forward model of WRFPLUS, simplified versions of these schemes are used in its tangent linear and adjoint models. In addition, WRFPLUS uses full radiation schemes (longwave and shortwave) in its forward model, but it excludes radiation schemes from its tangent linear and adjoint models. The difference in the physical parameterizations between the forward model and tangent linear/adjoint

models in a 4DVar system is a source of linearization error. For instance, Tremolet (2004) found linearization error in ECMWF 4DVar larger than expected and recommended more accurate linear physics for higher resolution 4DVar systems. Because WRF-CO2 4DVar ignores the impacts of $CO_2$ mixing ratio variation on the meteorological fields, no tangent linear and adjoint variables for meteorological fields are needed in its tangent linear and adjoint models. Since this second category of processes is not directly involved in $CO_2$ transport, there is no need for their tangent linear and adjoint procedures in WRF-CO2 4DVar. In

WRFPLUS's tangent linear model, the tangent linear code of the simplified versions of the cumulus, surface, and microphysics schemes, was removed and replaced with the code for the full schemes as used in the forward model. In WRFPLUS's adjoint model, the forward sweep updates the state variables and local variables just as in the forward model, but it also stores these variables' values for the subsequent backward sweep, which updates the adjoint variables of the state variables. The simplified versions of the cumulus, surface, and microphysics schemes used in the forward sweep of WRFPLUS's adjoint model were

removed and replaced with the full schemes used in the forward model. Since these processes do not directly modify $CO_2$ mixing ratio, their corresponding adjoint code was removed from the backward sweep of the adjoint model, as indicted by the 'X' in Table 2.

5    The third category includes advection, diffusion, emission, and turbulence mixing in the PBL, along with convective transport of $CO_2$. Because these processes directly modify $CO_2$ mixing ratio, their tangent linear code and adjoint code are needed for WRF-CO2 4DVar. The modifications made for advection and diffusion are described in Section 2.4.2, and those for emission, turbulent mixing in the PBL, and convective transport of $CO_2$ are detailed in Section 2.4.3.

### 2.4.2   Advection and diffusion of $CO_2$

10  WRF includes the advection and diffusion of inert tracers along with other scalars in its ARW dynamical core. The tangent linear and adjoint code for these processes has been implemented in WRFPLUS. It should be noted that the variables for these inert tracers are part of WRF, instead of WRF-Chem. WRF-Chem uses a separate array for its chemistry species. Since WRF-Chem is used as the forward model of WRF-CO2 4DVar, the $CO_2$ mixing ratios are included in the chemistry array. In the GHG option of WRF-Chem used for WRF-CO2 4DVar, $CO_2$ from different sources (anthropogenic, biogenic, biomass 15  burning, and oceanic) are represented by separate variables in the chemistry array. Following the treatment for the inert tracers in WRFPLUS, subroutines solve_em_tl and solve_em_ad were modified to add the tangent linear and adjoint code for the advection and diffusion of the chemistry array. The modifications made include adding calls to the procedures that calculate advection and diffusion tendencies, updating the chemistry array with the tendencies and boundary conditions, and addressing the Message Passing Interface (MPI) communications. The new upgrade to WRFPLUS described in (Zhang et al., 2013) greatly 20  expedited this part of development for WRF-CO2 4DVar. The 'Add' in Table 2 for advection and diffusion emphasizes that their tangent linear and adjoint code are added to WRF-CO2 4DVar based on the existing WRFPLUS code without substantial new code development.

### 2.4.3   Vertical mixing of $CO_2$ in PBL and convective transport

An accurate representation of vertical mixing is important for inversion accuracy, because misrepresentation causes transport 25  error, which manifests itself in the innovation vector and causes error in posterior estimation (Fowler and Lawless, 2016). For instance, Stephens et al. (2007) pointed out that global chemistry transport model error in vertical mixing and boundary layer thickness could cause significant overestimation of northern terrestrial carbon uptake. A comparison of four global models found that model transport uncertainty exceeds the target requirement for the A-SCOPE mission of 0.02 Pg C yr$^{-1}$ per $10^6$ km$^2$ (Houweling et al., 2010). In addition, Jiang et al. (2008) reported that convective flux is likely underestimated in boreal winter 30  and spring based on simulated upper tropospheric $CO_2$ from 2000 to 2004 using three chemistry transport models.

In WRF-Chem, vertical mixing of chemical species is treated in three separate parts: in the vertical diffusion (subgrid scale filter) in the dynamical core, in the PBL scheme in the physics driver, and in the convective transport in the chemistry driver.

The subgrid scale filter in the dynamical core treats both horizontal and vertical diffusion, but vertical diffusion is turned off if a PBL scheme is used.

For PBL parameterization, ACM2 (Pleim, 2007) was chosen for WRF-CO2 4DVar. ACM2 is a hybrid local-nonlocal closure
PBL scheme, and it updated the non-local scheme ACM1 (Pleim and Chang, 1992) by adding an eddy diffusion component. Because ACM2 explicitly defines local and nonlocal mass fluxes, it is particularly applicable for atmospheric chemistry simulations. In a one-dimensional model evaluation, ACM2 showed a very good agreement with large-eddy simulations for PBL heights with a very slight low bias (Pleim, 2007). In addtion to WRF, ACM2 has been implemented in the fifth-generation Pennsylvania State University-NCAR Mesoscale Model (MM5) and the Community Multiscale Air Quality (CMAQ) model.
An evaluation using PBL heights derived from radar wind profiles showed that the MM5-ACM2 is capable of realistic simulation of PBL heights (Pleim, 2007). Hu et al. (2010) evaluated three WRF PBL schemes and found that ACM2 resulted in less bias than the local closure scheme Mellor-Yamada-Janjic (MYJ) when compared with surface and boundary layer observations. Furthermore, model evaluations also showed that ACM2 performed well with CMAQ for air pollution simulations (Nolte et al., 2015; Appel et al., 2017) .
Convective transport of chemistry species in WRF-Chem is not treated by the cumulus scheme in the physics driver, but by a separate convective transport module (module_ctrans_grell) in the chemistry driver (Grell et al., 2004). This convective transport module includes a deep convective process and a shallow convective process. The deep convective transport process requires the convective precipitation rate calculated by the cumulus scheme (in the physics module of WRF): It calculates the base mass flux based on the convective precipitation rate. Compared to the ensemble stochastic approach used in the Grell-
Freitas cumulus scheme (Grell and Freitas, 2014), this is a rather crude representation of the vertical convective transport. The shallow convective process requires three parameters passed in from the cumulus scheme in the physics module: updraft originating level, cloud top level, and total base mass flux. Only two cumulus schemes in WRF provide these parameters: the Grell-Freitas (Grell and Freitas, 2014) and Grell 3D Ensemble (Grell and Devenyi, 2002).

Because the ACM2 PBL and chemistry convective transport are not included in WRFPLUS, their tangent linear and adjoint code were developed for WRF-CO2 4DVar. First the automatic differentiation tool TAPENADE (Hascoet and Pascual, 2013) was used to generate the tangent linear and adjoint code based on the forward code: module_bl_acm for the ACM2 PBL and module_ctrans_grell for the chemistry convective transport. Then the TAPENADE generated code was manually modified to remove redundancy and unnecessary loops. It should be pointed out that these code developments were made significantly
simpler because the meteorological state variables are merely passive variables in the tangent linear and adjoint code. For instance, to calculate the moist static energy and environmental values on cloud levels, the chemistry convective transport code (module_ctrans_grell) in the chemistry driver calls a number of subroutines in the cumulus parameterization code in the physics driver. Because these subroutines in the cumulus parameterization only involve meteorology state variables and not the chemistry array, no tangent linear or adjoint code is needed for them in WRF-CO2 4DVar.

## 3    Results

This section presents an accuracy assessment of the newly developed WRF-CO2 4DVar system. First the simulation model setup is described, then the sensitivity tests and inverse modeling experiments are presented.

### 3.1    Model setup

5    WRF-CO2 4DVar is setup with a domain covering the continental United States with 48 km $\times$ 48 km grid spacing and 50 vertical levels (Fig. 3). The domain dimension is 110 points in east-west and 66 points in north-south direction. Model configuration includes: Rapid Radiative Transfer Model (RRTM) longwave radiation (Mlawer et al., 1997), Goddard shortwave radiation (Chou and Suarez, 1999), Pleim surface layer (Pleim, 2006), Pleim-Xiu land surface model (Pleim and Xiu, 2003), ACM2 PBL (Pleim, 2007), Grell-Freitas cumulus (Grell and Freitas, 2014), and Thompson microphysics (Thompson et al., 10   2008). Positive-definite transport is applied to the transport of scalars and $CO_2$.

CO$_2$ fluxes used for the simulations are from the CarbonTracker 2016 version (hereafter CT2016) (Peters et al., 2007). These fluxes are the optimized surface fluxes at a 3-hour interval and at $1 \times 1$ degree spatial resolution. The four individual CO$_2$ fluxes (biosphere, fossil fuel, fire, and ocean) are spatially interpolated to the WRF grid, and saved in chemistry input files. In the 15   following sensivitity tests and inverse experiments, the emission scaling facotr $k_{co2}$ is applied only to the biosphere flux. Daily mean biospheric fluxes are calculated as the arithmatic mean of the 3-hourly CT2016 fluxes at each surface grid cell, and the scaling factor $k_{co2}$ is applied as in Eq. (1). The daily mean biospheric flux used for the 24 hour simulation is shown in Figure 4. The model configuration and emission data used are summarized in Table 3. Although the daily mean biospheric flux was used for the forward and inverse modeling in this paper, the WRF-CO2 4DVar implementation allows flexibility in configuring the 20   prior fluxes. For instance, fluxes from respiration and photosynthesis can be estimated independently, and at higher temporal resolution (Gourdji et al., 2012). When using these options with real observations, the balance between the degrees of freedom in the state vector and the constraints provided by the observations need to be carefully considered.

Model simulations span 24 hours from 00 UTC 02 June to 00 UTC 03 June, 2011. Meteorological initial and lateral bound-25   ary conditions are prepared using the NCEP Climate Forecast System Version 2 (CFSv2) $1 \times 1$ degree 6-hourly products (Saha et al., 2014). CO$_2$ initial and lateral boundary conditions are from the CT2016 global $3 \times 2$ degree CO$_2$ mole fraction. A method similar to PREP-CHEM-SRC (Freitas et al., 2010) was used to horizontally and vertically interpolate CT2016 mole fraction data to the WRF grid.

30   First, the forward model (WRF-Chem) was run for 24 hours with the CO$_2$ emission as described in the last section. Trajectory files that contain model state variables including both meteorology and CO$_2$ mixing ratio are saved at model dynamical time step intervals (120 seconds). These files are required for the subsequent tangent linear and adjoint model runs. Figure 5 shows the instantaneous values of Sea Level Pressure (SLP) and horizontal wind at the model's lowest vertical level at every

6 hours. The figure shows that a high pressure system was located off the west coast, causing a northerly surface wind off southern California, and a westerly wind for most of the Pacific Northwest. A low pressure system intensified over Montana and North Dakota during the 24 hours, causing a strong southerly wind over the Midwest. In the northeast, as a low pressure system moved eastward out of the domain, the surface wind shifted from southwesterly to westerly.

In the model setup, the initial and boundary meteorological conditions are generated by downscaling the CFSv2 data. Downscaling coarse resolution global reanalysis data could lead to poor WRF performance. Although this potential problem is not a concern for the present pseudo-observation based inversion experiments, it must be properly treated in future applications with real observations. Error in the initial condition will lead to erroneous flux attribution, especially for inversions with a short assimilation window.

In order to be useful for applications which employ real observational data, WRF-CO2 4DVar requires accurate simulations of the meteorological fields by the forward model. Because transport error can only be partially accounted for in the 4DVar system through the observation error covariance matrix, it is imperative to minimize errors due to inaccurate simulation of meteorological processes as much as possible. Although the present paper uses pseudo-observation data (which have zero transport error by definition) in its inversion experiments, future applications with true observational data will require vigorous evaluation of the model simulated meteorology and associated transport error. In the following, the forward model simulated horizontal winds at the surface and 500 hPa constant pressure surface are evaluated using in-situ measurements from weather stations and radiosondes.

For the surface level, WRF simulated 10m winds are compared against surface weather station measurements archived in the NOAA Integrated Surface Dataset (Smith et al., 2011). Hourly surface wind measurements from more than 2,000 stations within the WRF domain are used for the evaluation. Comparisons of wind speed and wind direction are carried out at the top of each hour during the 24 hour simulation period starting at 00:00 UTC 02 June 2011. Excluding missing observations, this results in 31,745 valid data pairs, which are summarized in the histograms of Fig. 6. RMSE for the hourly wind speed is 2.16 m s$^{-1}$ and the mean difference in the hourly wind direction is 29.4º.

For the upper level, WRF simulated 500 hPa horizontal winds were compared against radiosonde measurements from 90 stations obtained from the NOAA/ESRL radiosonde database (https://ruc.noaa.gov/raobs/). Since most stations release balloons at 00:00 and 12:00 UTC, WRF winds were compared against the radiosonde measurements at a 12 hour interval during the 24 hour simulation period. The results are shown in Figure 7: RMSE of wind speed is 2.54, 4.0, and 5.11 m s$^{-1}$, at 2 June 00:00 UTC, 2 June 12:00 UTC, and 3 June 00:00 UTC, respectively. Wind direction difference between WRF and radiosonde is 11.5º, 16.4º, and 19.1º at the three times. Locations of the weather stations and radiosonde sites used in the evaluations can be found in the supplement document.

The above described evaluations using in-situ measurements indicate that the meteorological simulation is of adequate accuracy for the pseudo observations based inverse modeling tests conducted in this paper. When the 4DVar system is applied with real observations, the error and bias must be considered. In WRF-4DVar's cost function configuration, the observation error matrix $\mathbf{R}$ is a combination of three error sources: measurement error, aggregation error, and transport model error. The uncertainty of the $CO_2$ measurements is about 0.05%, while the transport and aggregation errors are typically an order of magnitude larger (Bruhwiler et al., 2005). For real observation applications, the variance and covariance in $\mathbf{R}$ need to represent the transport error. Furthermore, Fig. 6 shows that WRF simulated 10m wind speed is biased high, which is likely to result in bias in the simulated atmospheric $CO_2$ mixing ratio. Because Bayes inversion framework assumes unbiased observation error, it may be imperative to correct the error for inversions. One approach is to nudge the meteorology fields toward the observations. For instance, Gupta et al. (1997) found that nudging the model simulated winds in the boundary layer to the radar wind profile observations substantially improved estimates of plume dispersion. An alternative approach is to use a combined 4DVar inversion of meteorology and $CO_2$ fluxes. For instance, Bocquet et al. (2015) discussed data assimilation using coupled chemistry meteorology models (CCMM). If the $CO_2$ impact on meteorology is not considered, the current implementation of WRF-CO2 4DVar can be extended to a joint meteorology and $CO_2$ assimilation system. Since the adjoint code for meteorology has been developed and tested in WRFPLUS and WRFDA (Zhang et al., 2013; Barker et al., 2012), the major modification would be in the optimizaiton schemes where the combined state vector of meteorology and $CO_2$ is optimized. It should be noted that in such a joint assimilation framework, optimization of meteorology is an initial condition problem, whereas the $CO_2$ flux optimization is a boundary condition problem (bottom and lateral boundaries).

### 3.2 Accuracy of tangent linear and adjoint sensitivities

Next, the accuracy of the newly developed tangent linear and adjoint models was evaluated by comparing their sensitivity calculations against finite difference sensitivity calculated by the forward model. Grid cells involved in the sensitivity calculations are shown in Fig. 3, in which the 35 blue stars are the source cells, and the 20 red triangles are 20 tower sites where the receptors are placed. All the 35 sources are placed at the grid's bottom vertical level. Receptors are placed at the 1[st], 5[th], and 10[th] vertical level at each of the 20 tower sites, resulting in 60 receptor cells.

A tangent linear model run for a grid cell will calculate the tangent linear sensitivity $\partial\mathbf{q_{co2}}/\partial k_{co2}$, which approximates a column vector of the forward model's Jacobian matrix and quantifies the influence of the cell's flux change on $CO_2$ mixing ratio of its receptor cells downwind. In comparison, an adjoint model run for a grid cell will calculate adjoint sensitivity $\partial q_{co2}/\partial \mathbf{k_{co2}}$, which approximates a row vector of the forward model's Jacobian matrix and quantifies the influence on the cell's $CO_2$ mixing ratio by its source cells upwind. Because $k_{co2}$ multiplies emission in Eq. (1), the magnitude of the sensitivity is determined by both the magnitude of emission and meteorological transport.

To calculate tangent linear sensitivity at a grid cell, $g\_k_{co2}$ is set to unity at the cell and zero at all other cells at the start of a tangent linear model run. Upon completion, the values of $\mathbf{g\_q_{co2}}$ are the tangent linear sensitivities $\partial\mathbf{q_{co2}}/\partial k_{co2}$. To calculate

adjoint sensitivity at a cell, an adjoint model run starts with $a\_q_{co2}$ set to unity at the cell and zero at all others, and the values of $\mathbf{a\_k_{co2}}$ at the end of the simulation are the adjoint sensitivities. The adjoint model running in this mode is analogous to using a Lagrangian particle transport model in backward trajectory mode to compute the footprint of a receptor, such as shown in Fig. 4 of Gerbig et al. (2008).

The tangent linear sensitivity is first compared against the finite difference sensitivity. After confirming the accuracy of the tangent linear model, the adjoint sensitivity is compared against the tangent linear sensitivity. Finite difference sensitivities are calculated using the two-sided formula (Eq. (8)).

$$\frac{\partial f}{\partial x} = \frac{f(x + \Delta x) - f(x - \Delta x)}{2\Delta x} \tag{8}$$

The magnitude of $\Delta x$ used in Eq. (8) is determined by comparing the result from a range of different values. The finite sensitivities were calculated at the 35 sites using $\Delta x$ set to 0.01, 0.1, and 1.0, and the results show that the magnitude of all differences is less than $10^{-10}$ (results not shown) because WRF-CO2 is largely linear. For all subsequent calculations, $\Delta x = 0.1$ is used for Eq. (8).

Since both finite difference and tangent linear sensitivities form columns of the Jacobian matrix, their values can be compared cell by cell for all receptor cells for a given site. Figure 8 shows the comparison between the finite difference and tangent linear sensitivities at 9 of the 35 source cells. The dark straight lines in the figures are the 1:1 line. The maximum and minimum of the difference between finite difference and tangent linear sensitivities are given for each source cell. Results at the rest of the sources are similar (not shown). All differences are less than $10^{-10}$, confirming that the tangent linear model is accurate.

The adjoint model is next evaluated by comparing adjoint sensitivities against the tangent linear sensitivities. Because finite difference sensitivities form columns of the Jacobian matrix while adjoint sensitivities form rows of the Jacobian matrix, they can only be compared at the intersections of the rows and columns of the Jacobian matrix, meaning there are 2160 ($35 \times 60$) pairs of comparison. We organized these 2160 pairs into three groups based on the vertical levels a receptor is placed at and the
result is shown in Fig. 9. The minimum and maximum value of the difference between tangent linear and adjoint sensitivities in all three groups are no greater than $10^{-6}$, which is comparable to the accuracy tests from other adjoint model developments (Meirink 2008; Henze 2007), indicating that the adjoint model has been correctly implemented.

### 3.3    Spatial patterns of adjoint sensitivities

Adjoint sensitivity $q_{co2}/\mathbf{k}_{co2}$ quantifies how $q_{co2}$ of a given receptor is impacted by the flux scaling factor of all surface cells.
It is similar to the receptor footprint typically calculated using LPDM, such as Fig. 4 of Gerbig et al. (2008) and Fig. 1 of Alden et al. (2016). But $q_{co2}/\mathbf{k}_{co2}$ differs from footprint in that the former contains the combined impact of tracer transport and flux magnitude, while the latter is determined by tracer transport alone. The spatial patterns of the adjoint sensitivity were examined to discern the impacts of tracer transport. Figure 10 shows $q_{co2}/\mathbf{k}_{co2}$ of Centerville, Iowa (top row) and WLEF,

Wisconsin (bottom row). At each tower site, $q_{co2}/\mathbf{k}_{co2}$ of the receptor placed at the 1st and 10th vertical levels is plotted.

The adjoint sensitivities of the Centerville tower site indicate its $q_{co2}$ results primarily from surface flux located immediately south of the site. This pattern agrees with the fact that low level wind during the simulation period is predominantly southerly, tranporting tracers northward. There is also a marked difference in the adjoint sensitivity of the same tower site when the receptor is placed at a different height. The figure in the top left panel of Fig. 10 shows that the highest magnitude of $q_{co2}/\mathbf{k}_{co2}$ is closest to the tower itself, indicating a large impact from local fluxes. In comparison, when the receptor is placed at the 10th vertical level, the peak magnitude of its adjoint sensitivity is located further south of the tower site. Results from the WLEF site shows the adjoint sensitivity are located to the southeast of the site, matching the southeasterly wind patterns around Wisconsin during the simulation period. There are also clear difference between the receptors at the different vertical levels. Results from other sites all show similar pattern of impacts of transport and receptor placement height (not shown).

To provide a comparative view of the source-receptor relations, backward trajectories of particles released from the Centerville and WLEF sites were also calculated using the Lagrangian model HYSPLIT (Stein et al., 2015). WRF-CO2 forward model simulated meteorology saved at 1-hour intervals was used to drive the HYSPLIT trajectory calculations. To compare with the adjoint model result, two sets of simulations were carried out for each of the two tower sites. For each simulation, 30,000 particles were released from the location of the corresponding WRF grid box used in the adjoint sensitivity calculations. The starting locations of the particles were randomly distributed within the grid box. The resulting backward trajectories were combined with the biospheric $CO_2$ flux to calculate the footprint for the receptor locations. The HYSPLIT footprints were calculated on the same grid as used in the WRF-CO2 simulations to facilitate the comparisons between the two models.

Figure 11 shows the HYSPLIT calculated footprints for the Centerville and WLEF sites at the two different vertical levels. The four figures in Fig. 11 are the HYSPLIT counterparts of the adjoint sensitivity figures in Fig. 10. A comparison between Fig. 10 and Fig 11. shows that the results from HYSPLIT and the WRF-CO2 adjoint model compare well spatially. For instance, for the receptor placed at the first vertical level at Centerville, Iowa (Fig. 10(a) and Fig. 11(a)), the footprints from both models are primarily located in Missouri and Northwestern Arkansas. Based on the horizontal wind fields at the first level, these areas were upwind of the receptor location during the simulation period. Overall, the WRF-CO2 adjoint sensitivities contain larger surface areas compared to their HYSPLIT footprint counterparts. This difference is likely caused by the more diffusive nature of tracer transport in WRF-CO2: its finite difference scheme for tracer advection contains numerical diffusion, and it also includes an explicit horizontal diffusion term in the tracer transport (Skamarock et al., 2008). A further comparison at individual grid points reveals magnitude differences between the footprints from HYSPLIT and the WRF-CO2 adjoint model. This is mainly caused by the different treatments of turbulent vertical mixing by the two models. In WRF-CO2, the PBL and convective schemes parameterize tracer vertical mixing (see Section 2.4.3). For vertical mixing, HYSPLIT either uses the PBL heights calculated by WRF or it calculates PBL heights independently by analyzing temperature profiles. The footprints shown in Fig. 11 were simulated by HYSPLIT using PBL heights from the WRF-CO2 ACM2 PBL scheme. In a separate set of

HYSPLIT simulations with PBL heights calculated from the temperature profiles, only minor differences are observed in the resulting footprints (not shown).

## 3.4 Inverse modeling tests

### 3.4.1 Inverse modeling setup

The sensitivity tests in section 3.2 have confirmed that the tangent linear and adjoint models of WRF-CO2 4DVar are correctly implemented. In this section, inverse modeling tests are conducted to confirm that the two optimization schemes described in Section 2.2 and 2.3 are also correctly implemented. The inverse modeling tests here are designed following the approach used in Henze et al. (2007). To confirm that the GEOS-Chem 4DVar code was correctly developed, Henze et al. (2007) set $\mathbf{B}^{-1} = \mathbf{0}$ and $\mathbf{R} = \mathbf{I}$ (the identity matrix) and constrained the optimizations with error-free pseudo-observations. Because $\mathbf{B}^{-1} = \mathbf{0}$, analysis deviations from the first guess cause no increase in the cost function (see Eq. (3)). This means that if the 4DVar code is correctly implemented, the optimization will converge to the true solution used to generate the pseudo-observations. Such a configuration of $\mathbf{B}$ and $\mathbf{R}$, although highly ideal and unrealistic for real applications, is an effective way to test the code accuracy in isolation from external errors. If the code is correctly implemented, the optimization will converge to the true solution used to generate the pseudo-observations. Because the background error is set to infinity ($\mathbf{B}^{-1} = \mathbf{0}$), the optimization should converge to the true solution with any first guess. A different first guess will impact the process of the convergence, but not the result: the optimization should eventually converge to the "true" solution.

Following Henze et al. (2007), inverse modeling tests here involve the following steps:

1. Run the WRF-CO2 forward model for 24 hours, using the daily mean biospheric $CO_2$ flux (Fig. 4) as the true biospheric $CO_2$ fluxes.

2. Generate pseudo-observations by saving the model simulated atmospheric $CO_2$ at all grid points of the bottom 30 vertical levels every 4 hours. This creates a set of 6 pseudo-observation files, which contain no error with respect to the true bisopheric $CO_2$ flux used in Step 1.

3. Generate a set of first guess biospheric $CO_2$ fluxes.

4. Set the background error to infinity ($\mathbf{B}^{-1} = \mathbf{0}$) and the observation error to the identity matrix ($\mathbf{R} = \mathbf{R}^{-1} = \mathbf{I}$).

5. Run the L-BFGS-B and incremental optimizations with the first guess biospheric $CO_2$ flux (Step 3), constrained by the pseudo-observations (Step 2), until the optimized biospheric flux converges to the true biopheric $CO_2$ flux (Step 1).

Steps 3-5 repeat twice for two different sets of first guess biospheric $CO_2$ fluxes:

– Case 1: set flux scaling factor $k_{co2} = 1.5$ at all surface grid point.

– Case 2: set flux scaling factor $k_{co2}$ randomly distributed between 0.5 and 1.5.

Figure 12 shows the two sets of first guess biospheric $CO_2$ as compared with the true biospheric $CO_2$ fluxes. Each point in the figures represents a surface grid point. It should be noted that because the Case 1 first guess overestimates the true fluxes by 50% at all surface grid points, the background error are perfectly correlated, implying that all off-diagonal elements in $\mathbf{B}$ should be set to unity. However, since the inverse modeling tests are designed to be driven solely by the pseudo-observations (by set-

ting $\mathbf{B}^{-1} = \mathbf{0}$), the detailed content of $\mathbf{B}$ becomes irrelevant. It should also be noted that the same set of pseudo-observations (Step 1) are used for both of the two cases of first guesses, and the pseudo-observations were not perturbed with errors, it is appropriate to set $\mathbf{R} = \mathbf{I}$ for both cases. This simply assigns all the observations equal weight in calculating the observation cost function using Eq. (4). In these inverse modeling tests, because the pseudo-observations are of $q_{co2}$ at the forward model's grid points, the mapping between model space and observation space is trivial: the observation operator ($H$), tangent linear

observation operator ($\widetilde{H}$), and adjoint observation operator ($\widetilde{H}^T$) are all simply the indentity matrix. For application with real observations, however, each type of $CO_2$ observations will need its own set of $H$, $\widetilde{H}$, and $\widetilde{H}^T$ to map between the model space and observation space.

A very simple error configuration ($\mathbf{B}^{-1} = \mathbf{0}$, and $\mathbf{R} = \mathbf{I}$) was used in the inverse modeling tests here, but such a setting is

only appropriate to confirm code accuracy using error-free observations. For real data applications, an appropriate specification of background ($\mathbf{B}$) and observation error ($\mathbf{R}$) is a critical and challenging task. Ideally the variance and covariance in $\mathbf{B}$ should be specified based on comparisons between prior fluxes and accurate flux measurements (Chevallier et al., 2006; Gerbig et al., 2006). But available flux measurements are often of insufficient amount, thus necessitating assumptions regarding the form of the background error matrix. For instance, prior flux errors were treated as uncorrelated in Gockede et al. (2010), and Roden-

beck et al. (2003) used an exponential decaying spatial correlation for the prior flux error. In another study, Peylin et al. (2005) found significantly different flux estimation resulted from varying the prior flux error correlation scale from 500 km to 2000 km. For the observation error covariance matrix $\mathbf{R}$, the spatial and temporal error correlation were often negelected in earlier inversion studies (Gurney et al., 2002; Pillai et al., 2016). With more recent inversion studies using continuous observation at towers (Law et al., 2008), airborne observations (Lauvaux et al., 2008), and satellite observations (Chevallier et al., 2005),

attempts have been made to represent the spatial and temporal correlation of observation errors. For instance, Kountouris et al. (2015) found the temporal autocorrelation time for observation data using the VPRM model is around 30 days.

### 3.4.2   Inverse modeling results

The results from inverse modeling experiments with Case 1 prior are shown in Fig. 13 and 14. Figure 13 shows the iterative

reduction of the cost function $J(\mathbf{x})$, gradient norm $\|\nabla J(\mathbf{x})\|$, and RMSE. The iteration number for Lanczos-CG is all from its inner loop, and only one outer loop is used. The figures show both L-BFGS-B and Lanczos-CG reduce the cost function monotonically. In about the first 10 iterations, the cost function reduction is more or less similar for the two optimization schemes, but Lanczos-CG starts to gradually outperform L-BFGS-B after. In gradient norm reduction, both schemes feature periodic oscillations embedded in the large scale downward trend. By comparison, Lanczos-CG has a smaller magnitude oscillation

and steeper downward trend than L-BFGS-B. It should be noted that while L-BFGS-B calculates cost function and its gradient in each iteration, Lanczos-CG only approximates these values in its inner loop. The cost function and gradient norm from Lanczos-CG shown in Fig. 13 are calculated by extra calls to the forward and adjoint models in each inner iteration, which doubles the computation cost and is not needed in actual inversion applications. Figure 13(c) shows that both optimization schemes reduce RMSE of daily biosphere flux monotonically, and Lanczos-CG achieves better reduction after about the first 10 iterations. Figure 14 shows the snapshots of the optimized daily mean biosphere flux (obtained as the product of the prior flux and the optimized scaling factor) at a selected set of iterations. These figures depict the iterative process of priors converging to the true solution.

The results of inverse modeling experiments using Case 2 prior are shown in Figs. 15 and 16. The reductions of $J(\mathbf{x})$, $\|\nabla J(\mathbf{x})\|$, and RMSE are similar to Case 1 in that Lanczos-CG substantially outperforms L-BFGS-G after about the first 10 iterations. Table 5 summarizes the results from all four inverse modeling experiments described above. It must be pointed out that these inverse modeling results are obtained from a highly unphysical setup, and they are not the expected level of performance (in terms of cost function and RMSE reduction) that would be obtained in a inversion with real observations.

## 4 Tracer lateral boundary condition

The lateral tracer boundary condition is necessary to connect regional tracer simulations to the global background tracer distribution (Gerbig et al., 2003). A number of regional inversion studies have explored the sensitivity of the estimated posterior flux to the lateral boundary condition. For instance, Schuh et al. (2010) found a 30% magnitude difference in the retrieved North America biospheric flux when boundary conditions from two different global models were used (CarbonTracker and PCTM). In an inversion study over the state of Oregon, Gockede et al. (2010) found the estimated biospheric $CO_2$ fluxes were highly sensitive to systematic changes in the advected background $CO_2$ through the lateral boundaries. To address the lateral boundary uncertainty, Lauvaux et al. (2008) used LPDM backward trajectories to calculate the atmospheric $CO_2$ sensitivity to the lateral boundary conditions, and optimized lateral boundary conditions along with surface fluxes in a synthesis inversion approach. An alternative is to use part of the observations to correct the lateral boundary error before the inversion, which then only includes surface fluxes in its state vector (Lauvaux et al., 2012). In the pseudo-observation based inverse modeling tests described in Section 3 of this work, $CO_2$ lateral boundary conditions do not contain error, and they were not included in the state vector for optimization. When WRF-CO2 4DVar is applied with real observations, uncertainties of lateral boundary conditions need to be appropriately treated. To use either approach used in Lauvaux et al. (2008) or in Lauvaux et al. (2012), the adjoint code for tracer lateral boundary conditions would need to be developed for the WRF-CO2 4DVar system.

In the WRF-Chem dynamical core, chemistry mixing ratios are updated at each time step by the advection and diffusion tendencies. Then chemistry mixing ratios at the lateral boundaries are updated with the chemistry boundary condition using the flow dependent method, which uses the horizontal wind direction to determine whether the chemistry mixing ratio at a

boundary grid point should be updated by the lateral boundry. If the horizontal wind direction indicates tracer inflow at a boundary grid, Eq. (9) will be applied to the grid point.

$$q_{co2} = q_b + q_{b,t}\Delta t \tag{9}$$

Where $q_{co2}$ represents $CO_2$ mixing ratio at a lateral boundary grid, $q_b$ and $q_{b,t}$ are the $CO_2$ mixing ratio and tendency at the correponding lateral boundary.

To develop the lateral boundary related tangent linear and adjoint code, Eq. (9). is replaced by Eq. (10) in WRF-CO2 4DVar.

$$q_{co2} = k_{co2}(q_b + q_{b,t}\Delta t) \tag{10}$$

Where $k_{co2}$ represents the $CO_2$ lateral boundary scaling factor. Please note that in Eqs. (9) and (10), the time dependence has been dropped for the sake of simplicity. The corresponding tangent linear and adjoint of Eq. (10) are given in Eqs. (11) and (12).

$$g\_q_{co2} = g\_k_{co2}(q_b + q_{b,t}\Delta t) \tag{11}$$

$$a\_k_{co2} = a\_k_{co2} + a\_q_{co2}(q_b + q_{b,t}\Delta t) \tag{12}$$

Where $g\_q_{co2}$ and $a\_q_{co2}$ are the tangent linear and adjoint variable of $q_{co2}$, and $g\_k_{co2}$ and $a\_k_{co2}$ are the tangent linear and adjoint variables of $k_{co2}$.

Most code development for tracer lateral boundary conditions are in the input_chem_data module of the chemistry directory, along with some additional code modification to enable the lateral boundary condition variables to be passed forward ($k_{co2}$ and $g\_k_{co2}$) and backward ($a\_k_{co2}$) in time. The two optimization schemes of WRFCO2-4DVar have also been implemented to allow for flexibilities in state vector specification. The user can choose to include lateral boundary conditions in the state vector to be optimized, which is a similar approach as in Lauvaux et al. (2008) (but using a 4DVar optimization). Alternatively, the user can choose to correct the lateral boundary (using the adjoint model) before to the inversion, and not to include lateral boundary in the state vector (Lauvaux et al., 2012).

When applied with real observations, whether and how to aggregate lateral boundary scaling factors is not trivial (Lauvaux et al., 2008, 2012). On one hand, including lateral boundary scaling factors without spatial aggregation will greatly increase the state vector size, likely causing the inversion to be under-constrained. On the other hand, aggregating lateral boundary scaling factors may cause aggregation error (Kaminski et al., 2001). While the actual treatment of lateral boundary scaling factor aggregation is beyond the scope of this work, a mapping mechanism has been implemented in WRF-CO2 4DVar to facilitate the aggregation. In WRFCO2-4DVar, $q_{co2}$, $g\_q_{co2}$ and $a\_q_{co2}$ are defined on the model grid, but $k_{co2}$, $g\_k_{co2}$, and $a\_k_{co2}$ are defined as 1d variables in the state vector. The mapping mechanism implemented in procedure da_cv_to_wrf and its adjoint

counterpart allows for many-to-one mappings from the 3d grid variables to the 1d state vector. This mapping mechanism allow the user flexibility in determining whether and how to aggregate the lateral boundary condition.

## 5   Summary and outlook

WRF-CO2 4DVar was developed as a data assimilation system designed to constrain surface $CO_2$ fluxes by combining an online atmospheric chemistry transport model and observation data in a Bayesian framework. Two optimization schemes were implemented for cost function minimization. The first is based on L-BFGS-B and the second is an incremental optimization using Lanczos-CG. The cost function and its gradient required by the optimization schemes are calculated by WRF-CO2 4DVar's three component models: forward, tangent linear, and adjoint models, all developed on top of the WRFPLUS system. While WRFPLUS's forward model is WRF, WRF-Chem was used as WRF-CO2 4DVar's forward model to include $CO_2$ in the system, and the tangent linear and adjoint models were modified to keep their consistency with the forward model. Like most other $CO_2$ inverse modeling systems, WRF-4DVar ignores the possible impacts of atmospheric $CO_2$ variation on the meteorology. This simplification enables the use of the same full physical parameterizations in the forward, tangent linear, and adjoint models. This configuration reduces linearization error while allowing the WRF system's large number of physical parameterizations to be used in WRF-CO2 4DVar without requiring a large amount of new code development.

WRF-CO2 4DVar's tangent linear and adjoint models were tested by comparing their sensitivities' spatial patterns with the dominant wind patterns. The results make physical sense given the meteorological transport. The accuracy of tangent linear and adjoint models was evaluated by comparing their sensitivity against finite difference sensitivity calculated by the forward model. The results show that both tangent linear and adjoint sensitivities agree well with finite difference sensitivity. Finally, the system was tested in inverse modeling with pseudo-observations, and the results show that both optimization schemes successfully recovered the true values with reasonable accuracy and computation cost.

While Lanczos-CG performs better than L-BFGS-B in the inverse modeling tests, it must be pointed out that the tests are very limited. Although a comprehensive comparison between the two optimization schemes is beyond the scope of the present paper, it is important to point out some of their differences as implemented in WRF-CO2 4DVar. First, the Lanczos-CG calls the tangent linear model in each inner loop iteration, while L-BFGS-B calls the forward model. For a tracer transport system like WRF-CO2 4DVar, the tangent linear model can skip some of the costly physics parameterizations, such as the radiation scheme. This difference means that typically the tangent linear model is faster than the forward model, and as a result Lanczos-CG runs faster than L-BFGS-B. In our inversion modeling experiments (24-hour simulation with $\Delta t = 120$ seconds, 30 processor core), it takes about 10 minutes walltime to complete one inner loop of Lanczos-CG. L-BFGS-B takes about 10% more walltime to complete one iteration.

Second, provided with the cost function and its gradient, each iteration of L-BFGS-B calculates an updated state vector from its previous iteration. In WRF-CO2 4DVar, this calculation is carried out on only the root core and broadcasted to the other process cores. In comparison, Lanczos-CG calculates the state vector increment based on the cost function gradient alone (without the need for $J(\mathbf{x})$). The calculation is carried out on each processor core. The above difference has implications for memory requirements: The main memory allocation for L-BFGS-B is its workspace array, which is about $(2 \times k + 4) \times n$, where $n$ is the size of the state vector $(x)$, and $k$ is the number of corrections used in the limited memory matrix. This memory allocation is only needed on the root core. The value of $k$ is set by the user and the recommended value is between 3 and 20. In comparison, Lanczos-CG requires memory size of about $m \times n$ on each processor core, where $m$ is the maximal inner loop iteration allowed. Although it is possible to reduce the per processor core a memory allocation from $m \times n$ to $n$ by disactivating the modified Gram-Schmidt orthonormalization step, it is typically not recommened.

Another consideration for memory requirements is related to I/O time cost. WRFPLUS saves its entire trajectory in memory to avoid expensive I/O operations. This is not a practical solution for WRF-CO2 4DVar, which is designed to run a longer simulation than the typical 6-hour run intended for WRFDA. GH15/17 implemented a second-order checkpoint mechanism to overcome the memory limit. This approach breaks the whole simulation period into sections, saves restart files at the end of each section by the forward model. This approach requires extra calls of the forward model to recalculate the trajectory for each section during backward integration (See Fig. 3 of GH15). In WRF-CO2 4DVar, a different approach was implemented to overcome this memory limit: the forward model saves the trajectory at each time step in memory, as WRFPLUS does. After a number of integration steps, the memory on each task processor core is dumped to an external file, and the memory is then reused. Each external file is marked with its starting timestamp and the processor core it belongs to. For instance, a 24-hour simulation with 120-second time step will have a total of 720 steps. If the system saves its trajectory to external files each 30 time steps, memory allocation on each task processor core is only needed for 30 steps instead of 720 steps. This will results in 24 (720/30) trajectory files on each task processor core, and the total number of trajectory files depends on the number of processor cores used. These trajectory files are read by both tangent linear and adjoint models in a similar way as standard WRF auxiliary files. In the above example, they are read in at each 30 time steps, substantially reducing I/O time compared with reading in at each step. These trajectory files are different from standard WRF auxillary files in that each file belongs to an individual processor core, rather than being shared among all processor cores. This means all model runs in an inverse experiment must use the same domain patch configuration, which is the most common practice.

In future development, we plan to implement observation operators for real observations, including those from towers, satellites, and airborne instruments. This is required for applying WRF-CO2 4DVar with real observations. As a regional inverse system, the correct treatment of tracer lateral boundary conditions is important. We plan to test the lateral boundary condition adjoint code (Section 4) in a follow-up study. In addition, future applications of WRF-CO2 4DVar with real observations must use proper treatment of observation and background error covariance, which was not tackled in the pseudo-observation tests in

the present paper.

In addition, we also plan to periodically update the WRF-CO2 4DVar system to keep up with WRF system updates. Such updates will mainly consist of replacing the forward model with the updated WRF code, and developing the tangent linear and adjoint code for the relevant updated procedures. As the variable dependence analysis (Section 2.4.1) indicates that the tangent linear and adjoint code are only needed for a portion of WRF procedures, the amount of work required for updating WRF-CO2 4DVar is manageable. In addition, future development of WRF-CO2 4DVar will also be dependent on updates to WRFPLUS, which has always been updated along with WRF.

## 6   Code availability

WRF-CO2 4DVar source code can be retrieved via https://doi.org/10.5281/zenodo.1184200

*Acknowledgements.*  The authors express their appreciation for the WRF/WRF-Chem/WRFDA/WRFPLUS development teams for making their code available in the public domain. Discussion with Joel LeBlanc of Michigan Technological Research Institute (MTRI) improved the optimization schemes implementation and presentation in this paper. The insightful and detailed comments from the three reviewers greatly improved both the model and the paper. This work was partially supported by a Central Michigan University CST research incentive fund.

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

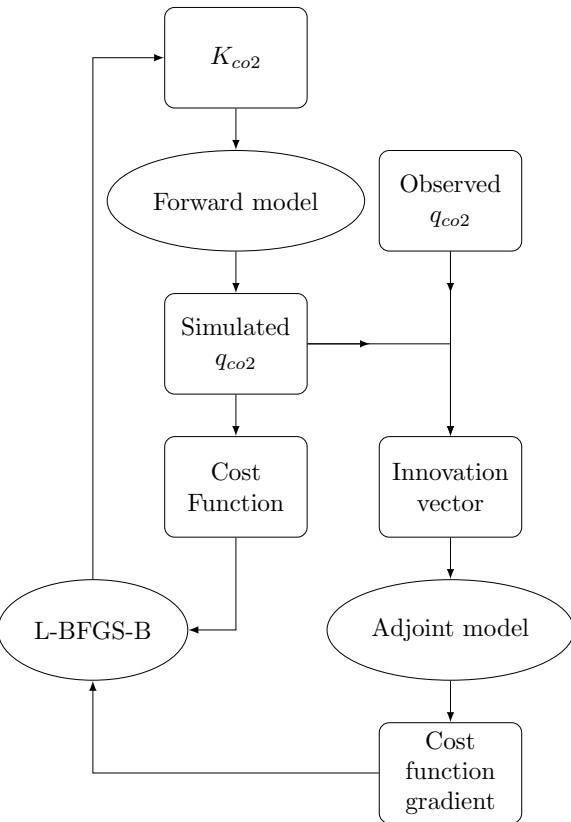

**Figure 1.** Diagram of L-BFGS-B based optimization implemented for WRF-CO2 4DVar.

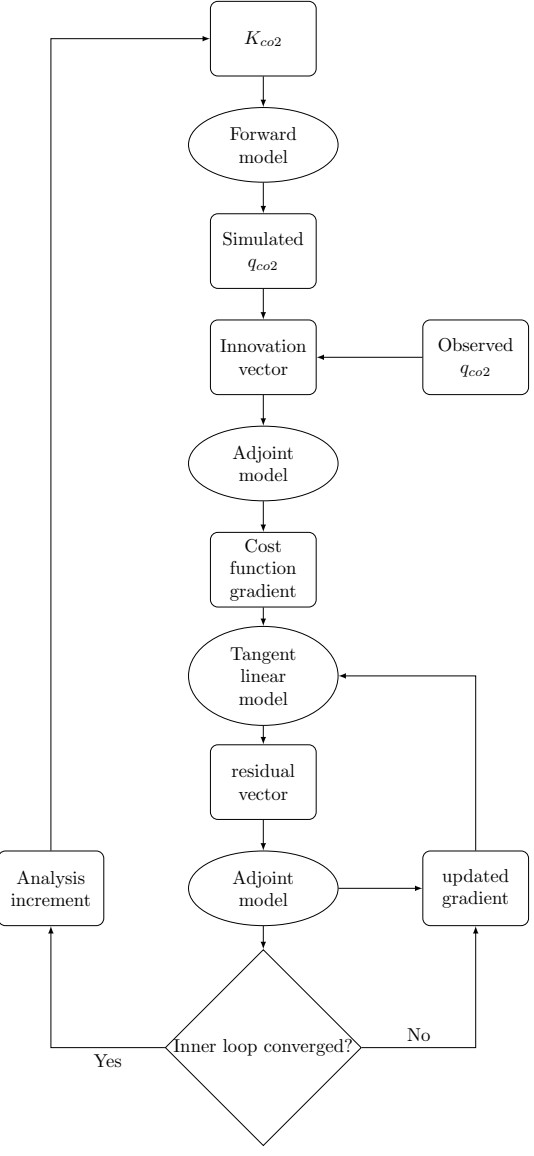

**Figure 2.** Diagram of Lanczos-CG based incremental optimization implemented for WRF-CO2 4DVar.

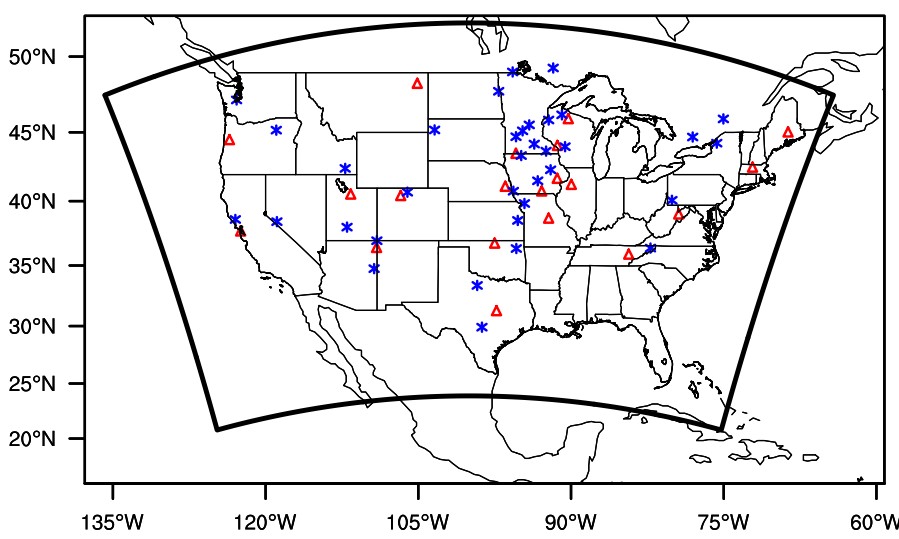

**Figure 3.** WRF-4DVar simulation domain covering the continental United State with 48 km$\times$48 km grid spacing. The domain boundary is marked by the bold dark outline. Grid cells used for evaluating sensitivities are marked: red triangles are the 20 $CO_2$ tower sites used as receptor locations; blue stars are source locations. While receptors are placed at the $1^{st}$, $5^{th}$, and $10^{th}$ vertical level at each site, all sources are at the $1^{st}$ level only.

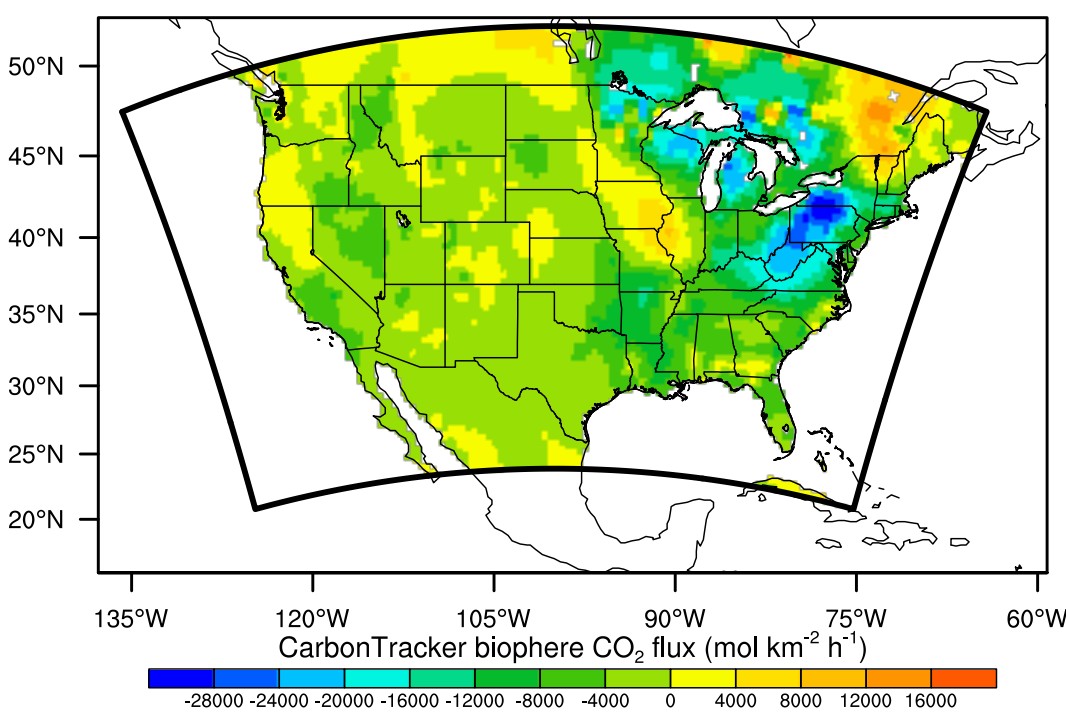

**Figure 4.** Daily mean CarbonTracker biosphere $CO_2$ flux, calculated as the arithmetic mean of the 3-hourly flux between 2011-06-02 00:00:00 UTC to 2011-06-03 00:00:00 UTC.

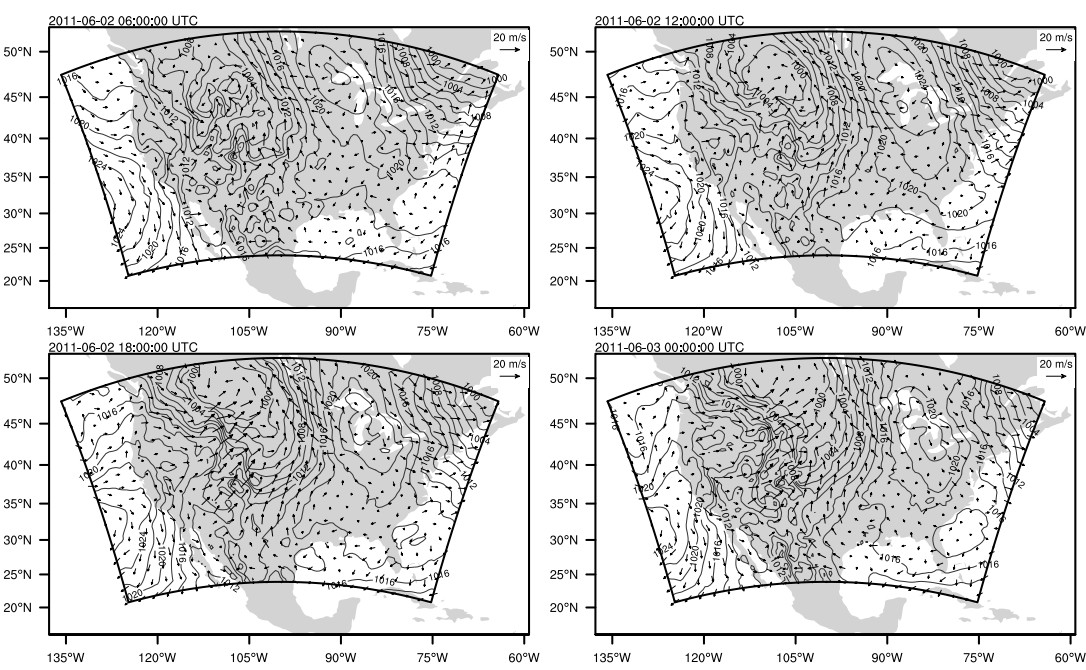

**Figure 5.** Sea Level Pressure (hPa) and horizontal wind (m s[-1]) at model's lowest vertical level plotted at 6-hour interval during the 24-hour simulation starting at 2011-06-02 00:00 UTC.

**Wind speed difference (m/s)**

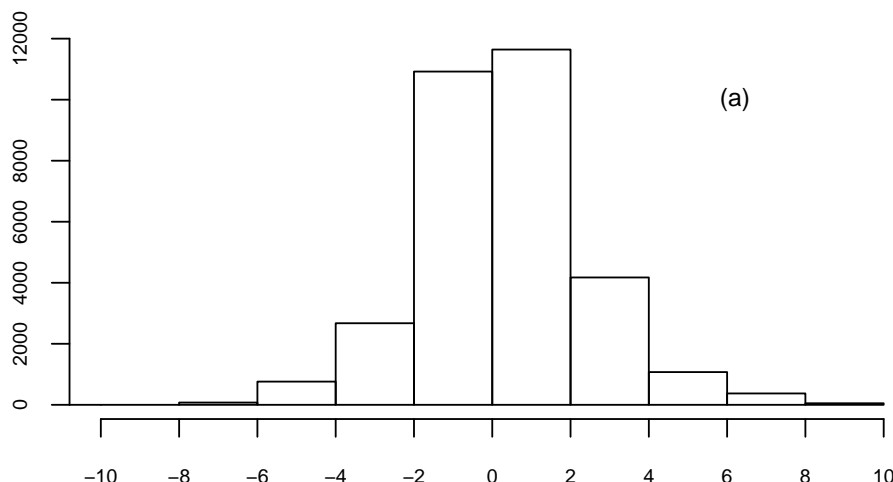

**Wind direction difference (degree)**

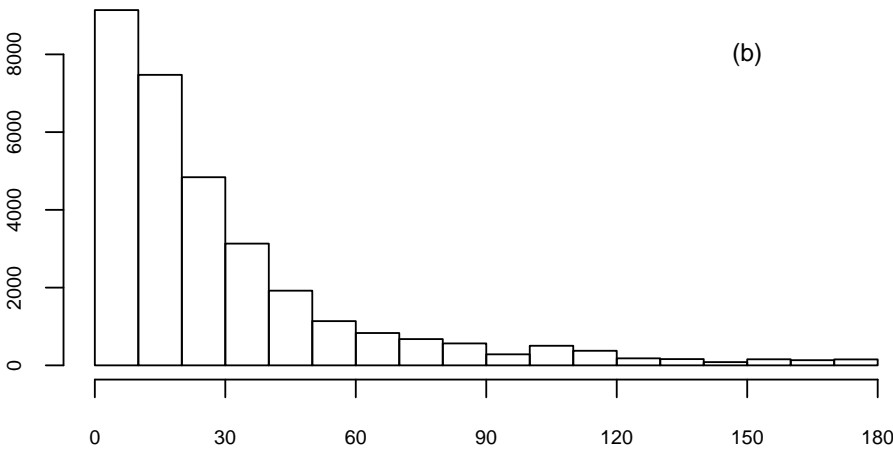

**Figure 6.** Histograms of the 10m wind speed difference (a) and wind direction difference (b) between WRF simulation and surface meteorological station measurements.

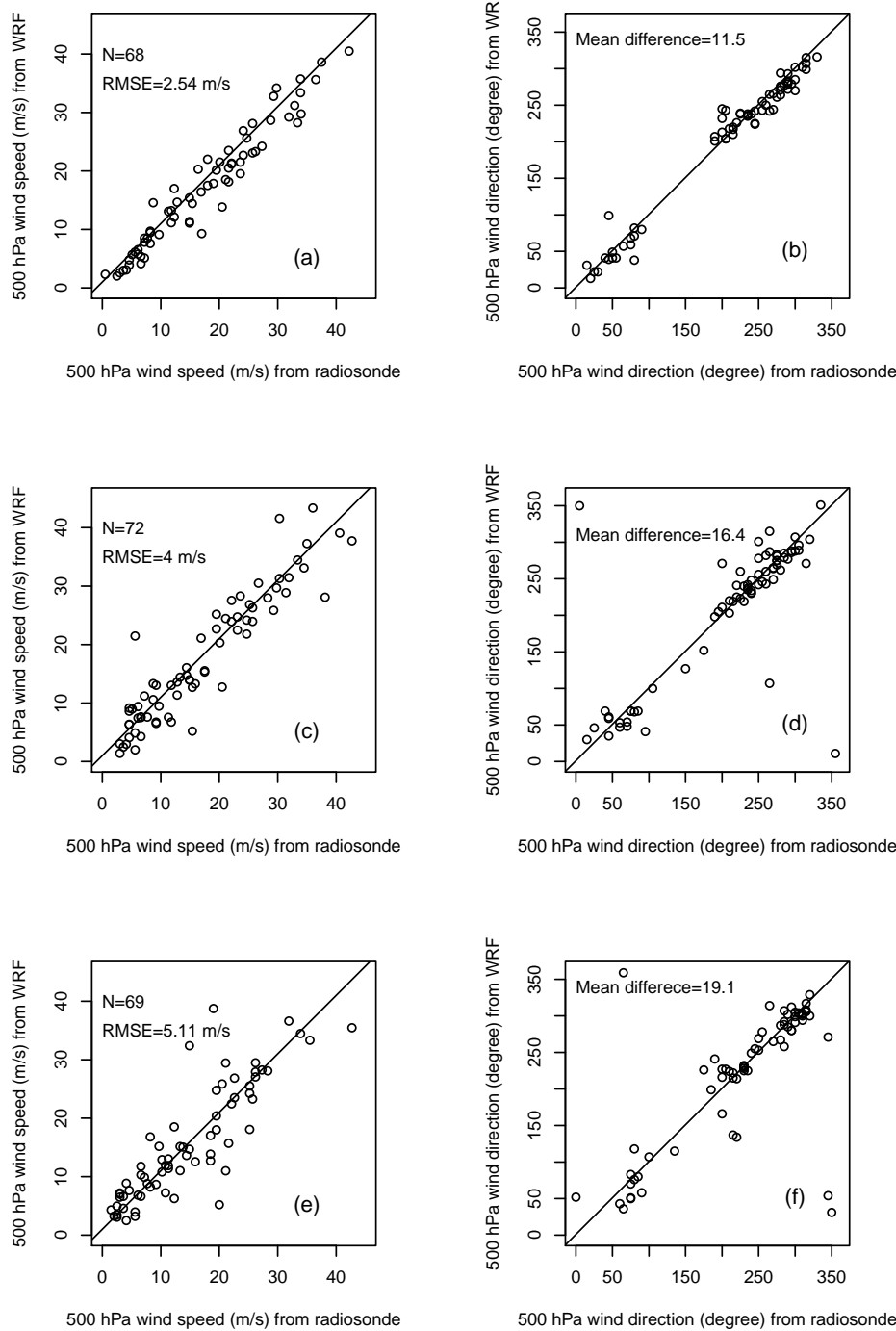

**Figure 7.** Comparison of 500 hPa wind speed and wind direction between WRF simulation and radiosonde measurements. Figures (a) and (b) are the comparison at 2011-06-02 00:00 UTC; Figures (c) and (d) are at 2011-06-02 12:00 UTC; and Figures (e) and (f) are at 2011-06-03 00:00 UTC. RMSE and relative error (RE) for wind speed and meann difference in wind direction are shown in each figure.

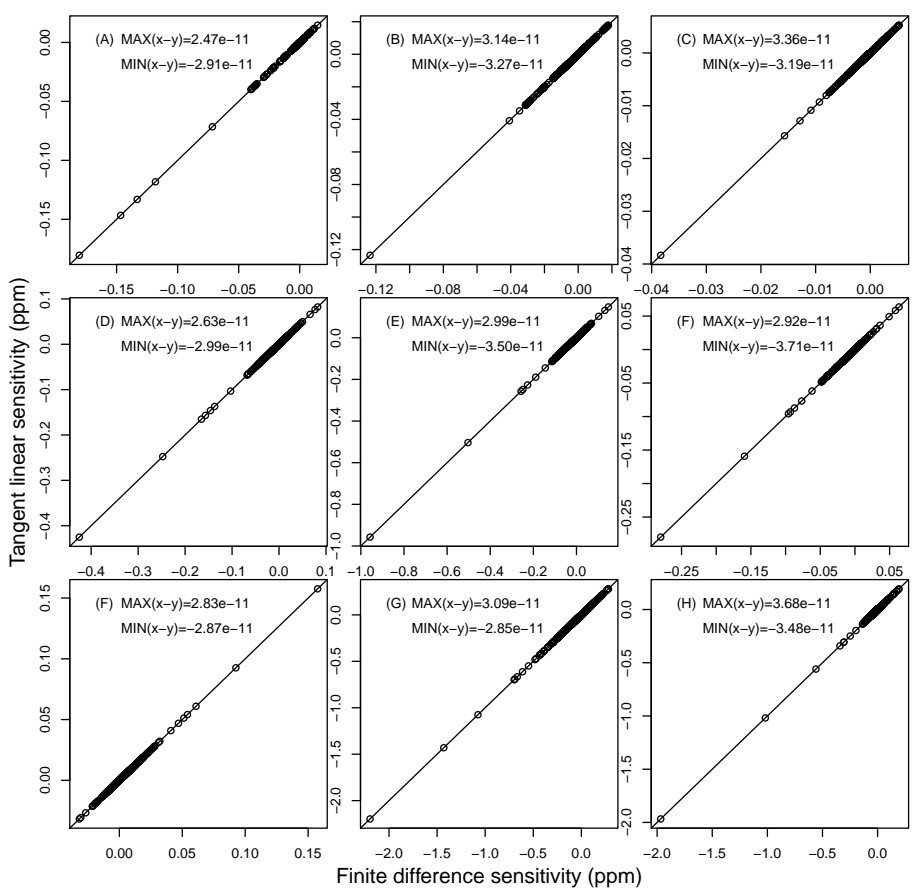

**Figure 8.** Comparison between $\partial q_{co2}/\partial k_{co2}$ calculated by finite difference (x axis) and tangent linear model (y axis) for nine source cell locations (see Fig. 3 for source locations).

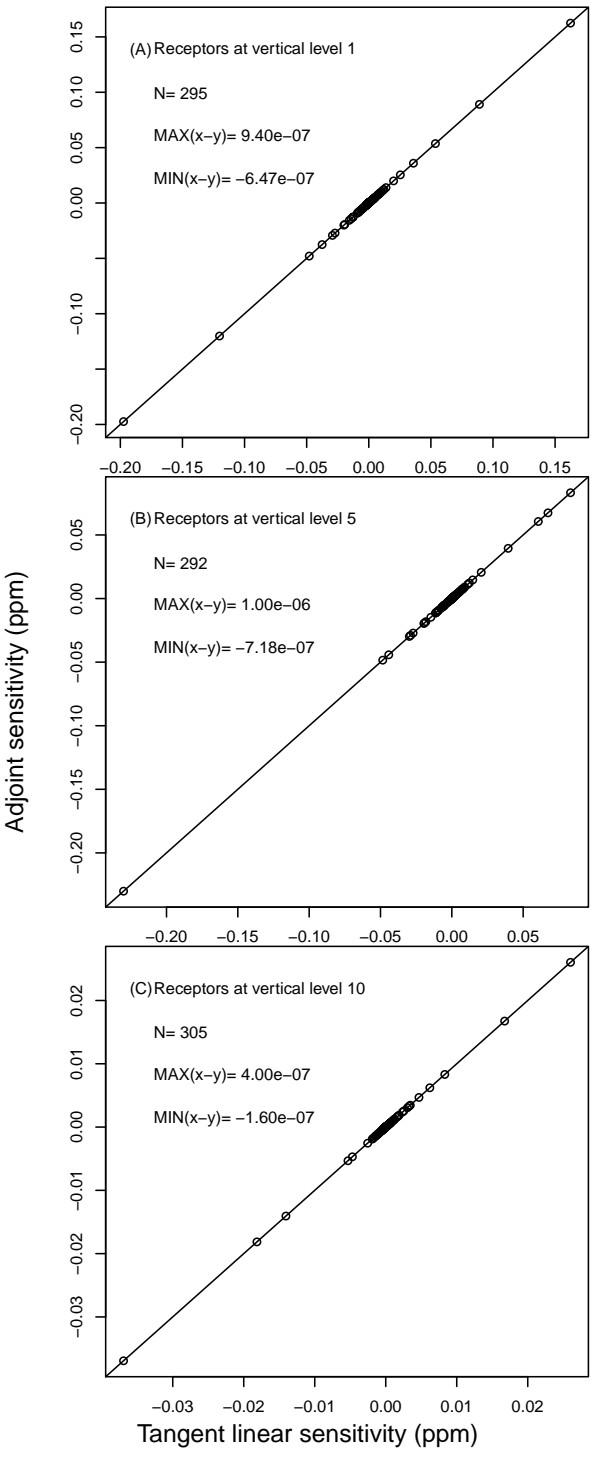

**Figure 9.** Comparison between $\partial q_{co2}/\partial k_{co2}$ calculated by the tangent linear (x axis) and adjoint model (y axis).

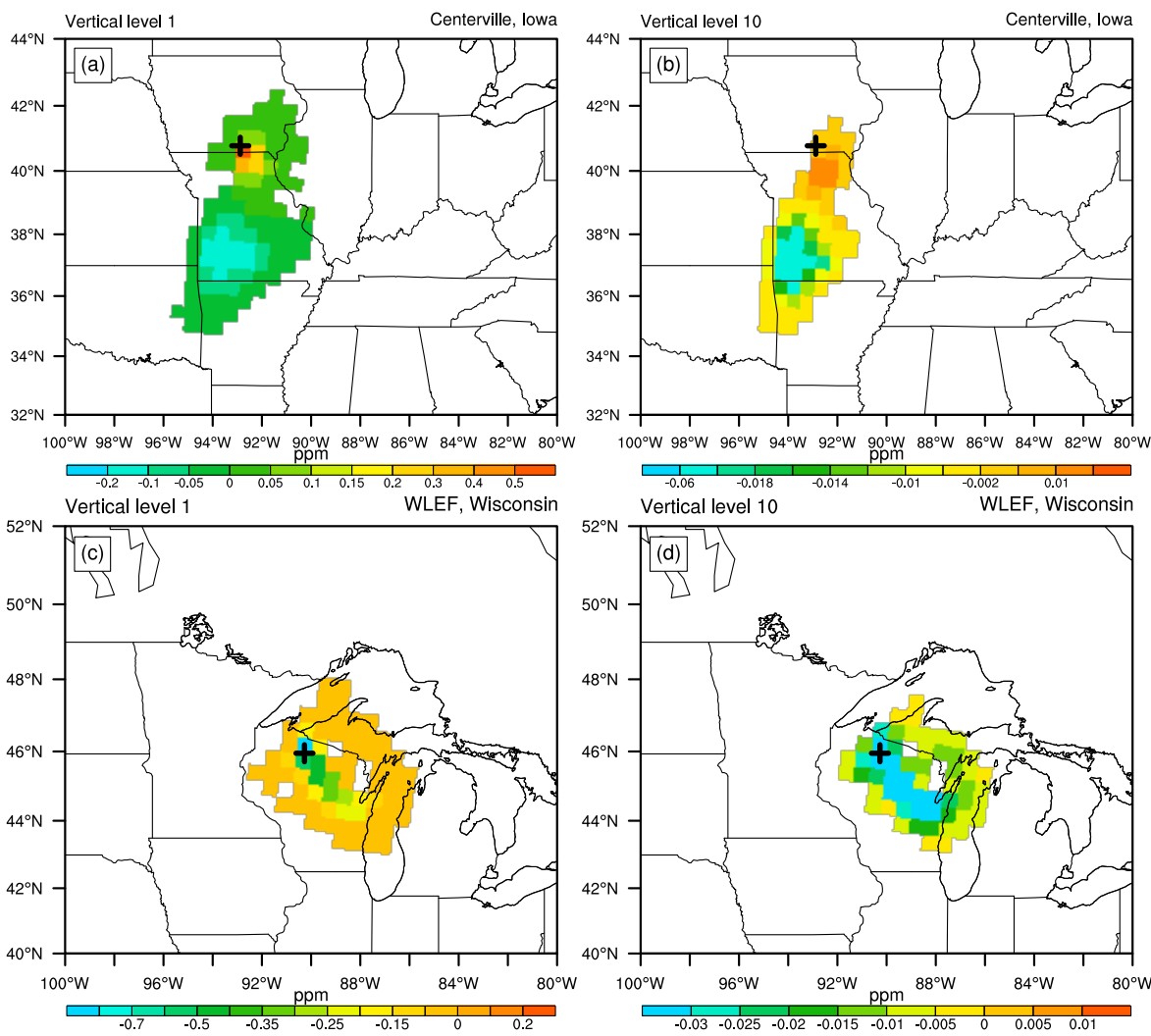

**Figure 10.** Adjoint sensensitivities calculated by the WRF-CO2 adjoint model. The top panel shows adjoint sensitivity of receptors placed at the 1st (a), and 10th (d) vertical level at Centerville, Iowa. The bottom panel shows adjoint sensitivity of receptors placed at the 1st (c), and 10th (d) vertical level at WLEF, Wisconsin. The black cross in each figure marks the corresponding tower site.

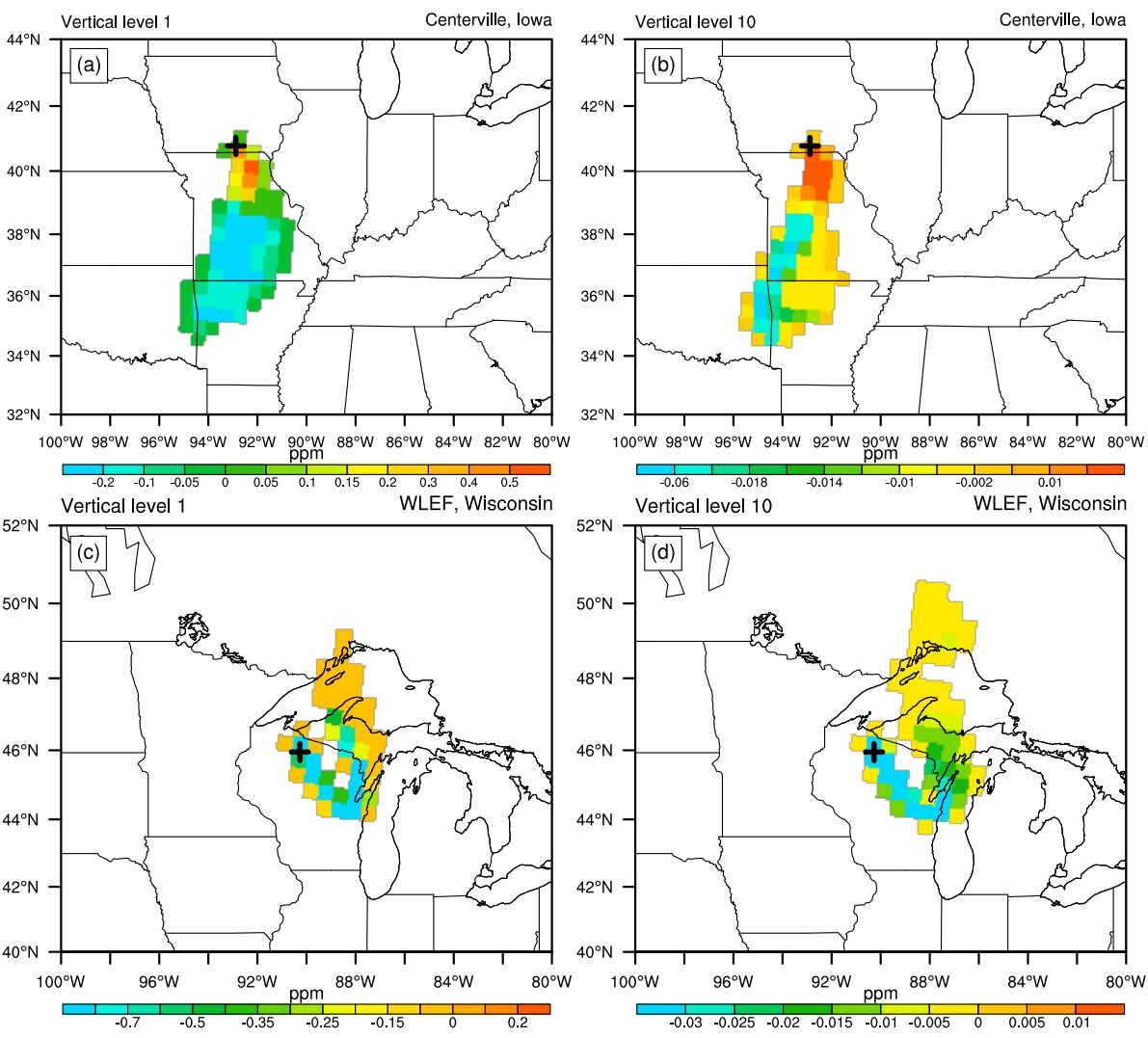

**Figure 11.** Footprints calculated using HYSPLIT backward trajectories and CarbonTracker biospheric fluxes for the tower sites at Centerville, Iowa and WLEF, Wisconsin. The receptor locations are the same as in Fig. 10. Each HYSPLIT footprint is plotted in the same color scale as its counterpart in Fig. 10 for comparision

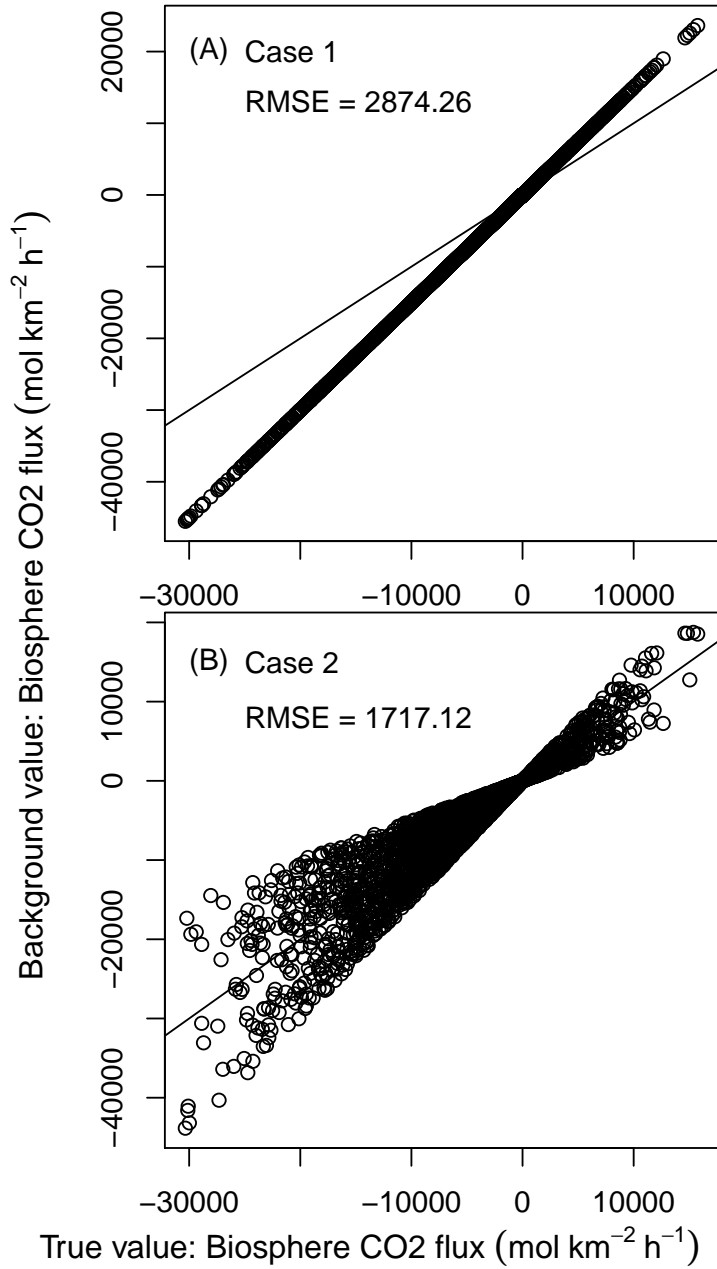

**Figure 12.** The first guess biosphere CO$_2$ fluxes used in the two inverse modeling experiments. The x-axis is true daily mean CarbonTracker biosphere CO$_2$ value (as shown in Fig 4), and y-axis is the first guess (background value). The solid line in each figure is the 1:1 line.

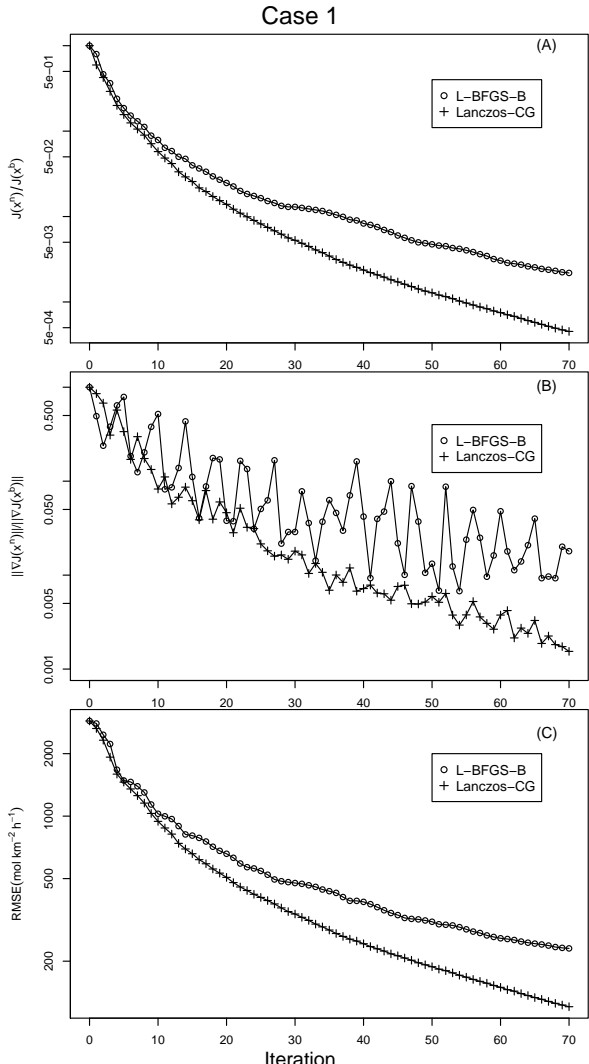

**Figure 13.** Results of inverse modeling experiment Case 1. Figure (a) shows the reduction of the cost function, represented by $J(x^n)/J(x^b)$. Figure (b) shows the reduction of the gradient norm, represented by $\left\|\nabla J(x^n)\right\| / \left\|\nabla J(x^b)\right\|$. Figure (c) shows the reduction of biospheric $CO_2$ flux RMSE.

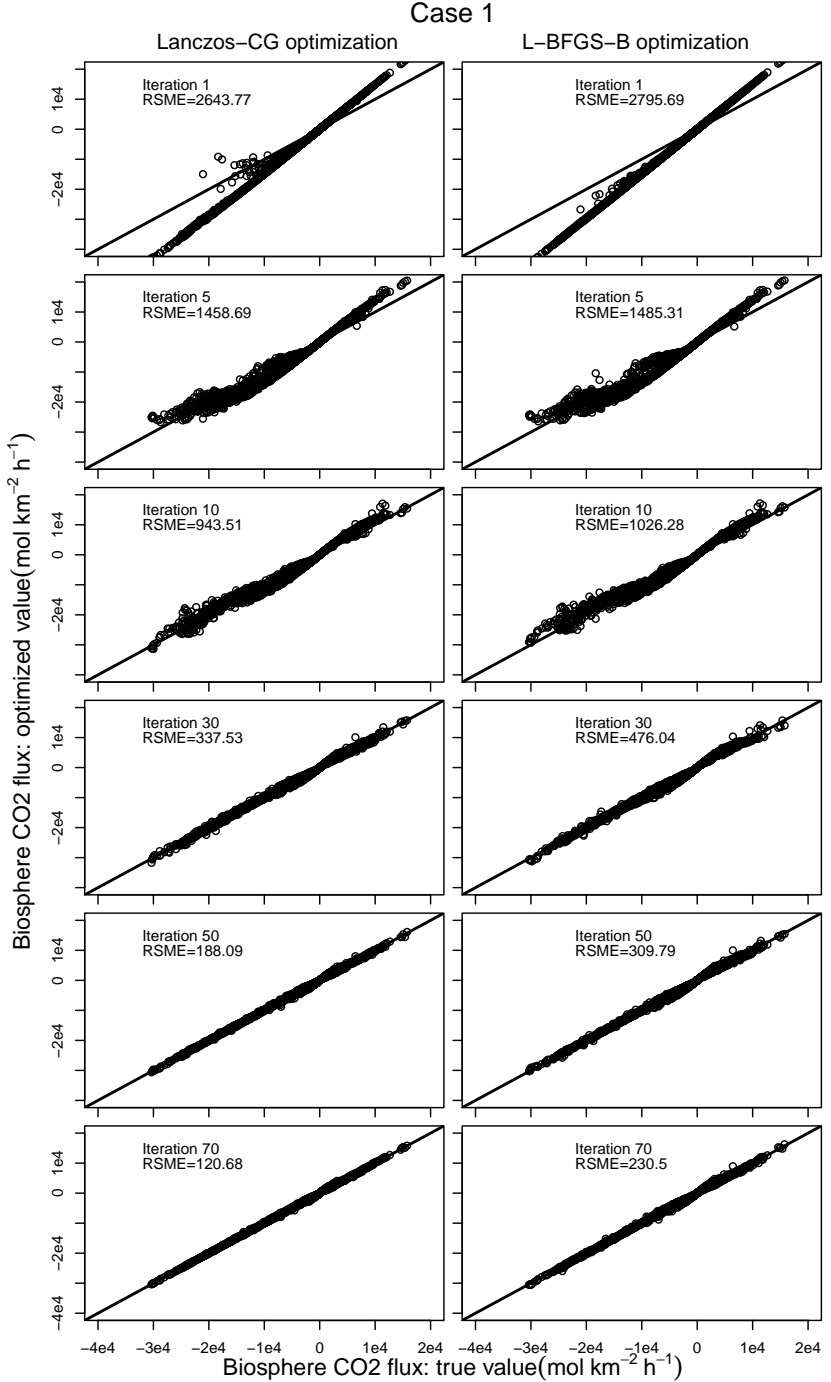

**Figure 14.** Comparison between the true and optimized $CO_2$ flux by Lanczos-CG (left column) and L-BFGS-B (right column) in inverse modeling experiment Case 1. The comparison and RMSE after the 1st, 5th, 10th, 30th, 50th, 70th iteration are shown in the figure. All iterations of Lanczos-CG are from one outer loop.

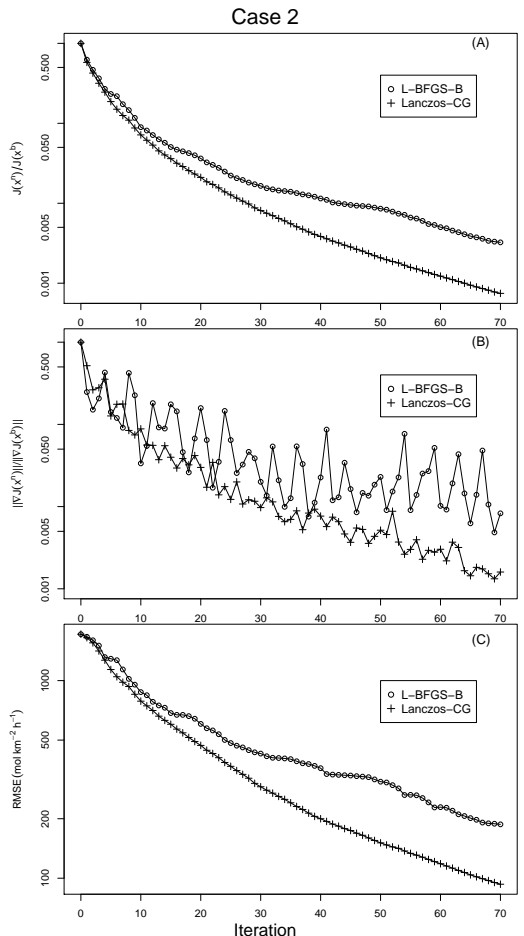

**Figure 15.** Same as Fig. 13, but for inverse modeling experiment Case 2.

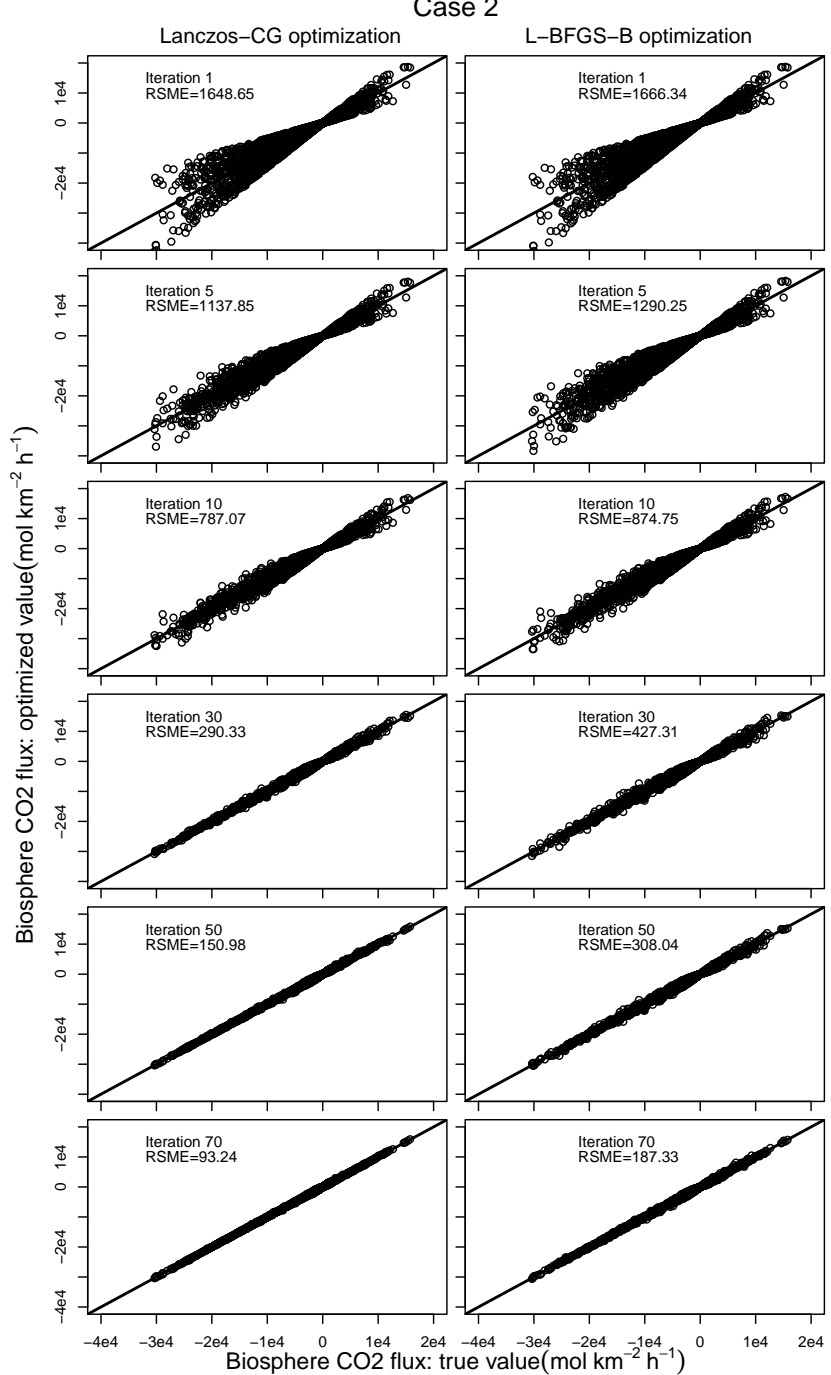

**Figure 16.** Same as Fig. 14, but for inverse modeling experiment Case 2.

**Table 1.** A list of symbols used in this article

| | |
|---|---|
| $J(\mathbf{x})$ | Cost function |
| $J_b(\mathbf{x})$ | Background cost function |
| $J_o(\mathbf{x})$ | Observation cost function |
| $\nabla J(\mathbf{x})$ | Cost function gradient |
| $\|\nabla J(\mathbf{x})\|$ | Cost function gradient norm |
| $\nabla^2 J(\mathbf{x})$ | Cost function Hessian |
| $\mathbf{B}$ | Background error covariance |
| $\mathbf{R}$ | Observation error covariance |
| $M$ | WRF-CO2 forward model |
| $\widetilde{M}$ | WRF-CO2 tangent linear model |
| $\widetilde{M}^T$ | WRF-CO2 adjoint model |
| $H$ | Observation operator |
| $\widetilde{H}$ | Tangent linear observation operator |
| $\widetilde{H}^T$ | Adjoint observation operator |
| $\mathbf{k}_{co2}$ | $CO_2$ emission scaling factor |
| $\mathbf{q}_{co2}$ | $CO_2$ mixing ratio (dry air) |
| $g\_\mathbf{k}_{co2}$ | Tangent linear variable for $CO_2$ emission scaling factor |
| $a\_\mathbf{k}_{co2}$ | Adjoint variable for $CO_2$ emission scaling factor |
| $g\_\mathbf{q}_{co2}$ | Tangent linear variable for $CO_2$ mixing ratio (dry air) |
| $a\_\mathbf{q}_{co2}$ | Adjoint variable for $CO_2$ mixing ratio (dry air) |
| $\mathbf{x}^b$ | Prior estimate of $CO_2$ emission scaling factor |
| $\mathbf{x}x^n$ | Analysis of $CO_2$ emission scaling factor |
| $\hat{\mathbf{x}}$ | Analysis increment of $CO_2$ emission scaling factor |
| $\mathbf{y}_k$ | Observation at the $k^{th}$ assimilation window |
| $\mathbf{d}_k$ | Innovation vector at the $k^{th}$ assimilation window |

**Table 2.** Summary of variable dependence analysis for developing WRF-CO2 4DVar component models on top of WRFPLUS. In the table, an 'F' means a full physics scheme is used in the forward model, tangent linear model, or the forward sweep of the adjoint model. An 'X' means a process is not needed for $CO_2$ treatment. A 'Dev' means a process does not exist in WRFPLUS and has been developed for WRF-CO2 4DVar. An 'Add' means a process for $CO_2$ is simply added using the existing WRFPLUS code for other tracers.

| Process | Forward model | Tangent linear model | Adjoint model forward sweep | Adjoint model backward sweep |
|---|---|---|---|---|
| Chemistry | X | X | X | X |
| Photolysis | X | X | X | X |
| Dry deposition | X | X | X | X |
| Wet deposition | X | X | X | X |
| Radiation | F | F | F | X |
| Surface | F | F | F | X |
| Cumulus | F | F | F | X |
| Microphysics | F | F | F | X |
| Advection | F | Add | F | Add |
| Diffusion | F | Add | F | Add |
| Emission | F | Dev | F | Dev |
| PBL | F | Dev | F | Dev |
| Convective transport | F | Dev | F | Dev |

**Table 3.** WRF-CO2 4DVar model configuration and $CO_2$ flux used in sensitivity and inverse modeling tests.

| | |
|---|---|
| Longwave radiation | Rapid Radiative Transfer Model (RRTM) |
| Shortwave radiation | Goddard shortwave |
| Microphysics | Thompson |
| Surface layer | Pleim-Xiu |
| Land surface | Pleim-Xiu |
| Planetary boundary layer | ACM2 PBL |
| Cumulus | Grell-Freitas |
| $CO_2$ advection | Positive-definite advection |
| biosphere $CO_2$ flux | CarbonTracker 2016 |
| ocean $CO_2$ flux | CarbonTracker 2016 |
| fire $CO_2$ flux | CarbonTracker 2016 |
| fossil fuel $CO_2$ flux | CarbonTracker 2016 |

**Table 4.** Summary of $CO_2$ tower sites. Sensitivity $\partial q_{co2}/\partial k_{co2}$ as calculated by WRF-CO2 4DVar's tangent linear and adjoint models is compared against finite difference sensitivity at these sites.

| Site Name | Symbol | Latitude | Longitude |
|---|---|---|---|
| Kewanee | RKW | $41.28^oN$ | $89.77^oW$ |
| Centerville | RCE | $40.79^oN$ | $92.88^oW$ |
| Mead | RMM | $41.14^oN$ | $96.46^oW$ |
| Round Lake | RRL | $43.53^oN$ | $95.41^oW$ |
| Galesville | RGV | $44.09^oN$ | $91.34^oW$ |
| Ozarks | AMO | $38.75^oN$ | $92.2^oW$ |
| WLEF | LEF | $45.95^oN$ | $9.27^oW$ |
| West Branch | WBI | $41.73^oN$ | $91.35^oW$ |
| Canaan Valley | ACV | $39.06^oN$ | $72.94^oW$ |
| Chestnut Ridge | ACR | $35.93^oN$ | $84.33^oW$ |
| Fort Peck | AFP | $48.31^oN$ | $105.10^oW$ |
| Roof Butte | AFC_RBA | $36.46^oN$ | $109.09^oW$ |
| Storm Peak Lab | SPL | $40.45^oN$ | $106.73^oW$ |
| Argle | AMT | $45.03^oN$ | $68.68^oW$ |
| Harvard Forest | HFM | $42.54^oN$ | $72.17^oW$ |
| Southern Great Plains | SGP | $36.80^oN$ | $97.50^oW$ |
| Sutro | STR | $37.75^oN$ | $122.45^oW$ |
| Hidden Peak | HDP | $40.56^oN$ | $111.64^oW$ |
| Mary's Peak | ARC_MPK | $44.50^oN$ | $123.55^oW$ |
| KWKT | KWT | $31.31^oN$ | $97.32^oW$ |

**Table 5.** Summary of inverse modeling experiment results. The reductions of cost function $J(x)$, gradient norm $\|\nabla J(x)\|$, and RMSE are given as the ratio to their respective starting values. Results of the two experiment cases are the values after 70 iterations.

|  | Case 1 | |
| --- | --- | --- |
| Reduction in | L-BFGS-B | Lanczos-CG |
| $J(\mathbf{x})$ | $2.23 \times 10^{-3}$ | $4.72 \times 10^{-4}$ |
| $\|\nabla J(\mathbf{x})\|$ | $2.0 \times 10^{-2}$ | $1.7 \times 10^{-3}$ |
| RMSE | $8.01 \times 10^{-2}$ | $4.19 \times 10^{-2}$ |

|  | Case 2 | |
| --- | --- | --- |
| Reduction in | L-BFGS-B | Lanczos-CG |
| $J(\mathbf{x})$ | $3.31 \times 10^{-3}$ | $7.76 \times 10^{-4}$ |
| $\|\nabla J(\mathbf{x})\|$ | $4.84 \times 10^{-3}$ | $1.32 \times 10^{-3}$ |
| RMSE | $1.09 \times 10^{-1}$ | $5.43 \times 10^{-2}$ |