# Peer review of "Development of the WRF-CO2 4DVar assimilation system v1.0"

_Geoscientific Model Development, 2016_

## Short Comment (SC1) · 16 Dec 2016

Dear authors,

in my role as Executive editor of GMD, I would like to bring to your attention our Editorial version 1.1:

http://www.geosci-model-dev.net/8/3487/2015/gmd-8-3487-2015.html

This highlights some requirements of papers published in GMD, which is also available on the GMD website in the 'Manuscript Types' section:

http://www.geoscientific-model-development.net/submission/manuscript_types.html

In particular, please note that for your paper, the following requirement has not been met in the Discussions paper:

- "The main paper must give the model name and version number (or other unique identifier) in the title."

Please add a version identifier for the published WRF-CO2 4DVar version in the title upon your revised submission to GMD.

Yours,

Astrid Kerkweg

---

## Referee Comment (RC1) · Anonymous Referee #1 · 6 Jan 2017

General comments:

1) The introduction includes quite a bit of information related to offline and online meteorology. However, I have a hard time understanding why the online system was chosen if the feedback of CO2 to the meteorology was ignored for the convenience. Why not instead choose development on an offline regional model? Also, will the system be updated periodically to account for the regular updates in the WRF model family?

2) As the authors noted, meteorology is critical to the quality of CO2 transport. Throughout the paper, I am surprised that there is no evaluation of the WRF meteorology, but this could be easily done considering the numerous observations available within the CONUS domain. In fact, I do have some concerns on the WRF setup as in specific comments 2)-3) below.
3) The comparison of L-BFGS-B and the CG has been done before. It'd be good to relate the prior results to yours and highlight any unique findings from your work.

4) Information on computation requirement and cost would be helpful.

Additionally, here are some places where clarifications or corrections are needed:

1) The cost functions, etc, are not quite consistent with literature on the similar topic. Vectors should be in bold.

2) Page 9, Line 26: Was indirect soil nudging enabled when PX LSM was used? It is recommended to enable it in retrospective analysis because little testing has been done for running PX with the indirect soil nudging disabled. See the WRF users guide and literature.

3) Page 10: Met IC/LBC from CFS on which resolution? Potential problems of downscaling that to 48km should be discussed. Again, some model evaluation should be added.

4) Page 10: any biomass burning emissions included? Does daily emission include any diurnal variability? Please include the emission amount in Section 3.1, to help understand the figure and results in Section 3.4.

5) Page 10, line 7-8: More details on the global WRF-Chem simulation is needed—I assume it was done on a coarser resolution than 48 km. Would the sensitivity and other tests in the global domain differ much from the regional results? Some simple comparison can highlight the benefit of using a regional scale 4dvar system.

6) Figure 1 and 2 need to be cited in the text. As red is already included in your emission color scheme, I suggest using a non-red color to show the locations of towers sites in Figure 3.

7) It'd be good to show Figure 5 and 6 along with trajectory calculations as in some prior works.

Some typos and grammar errors:

1) Page 5, line 19: according -> according to

2) Page 12, line 28: facotr -> factor

3) Page 37, Table 5 caption: givne ->given

―――――――――――――――

---

## Referee Comment (RC2) · Anonymous Referee #2 · 11 May 2017

The manuscript, "Development of the WRF-CO2 4DVar assimilation system", describes exactly that. The authors have created linearized model versions of the WRF-Chem CO2 transport mechanisms, validated their performance against finite difference approximations, and demonstrated their utility in a simplified pseudo-data experiment. The introduction covers much of the relevant literature necessary to get to the same starting point as the authors, and we make a few additional suggestions below. The adjoint and tangent linear model developments are described thoroughly, and would be helpful for any person working their way through the code at a later time. The adjoint model evaluation falls a little short, and we provide some suggestions for ways it could be improved. The pseudo-data inversion test, while quite simple and unrealistic, demonstrates that the inversion framework is working. It is a first step that undoubtedly took considerable effort, but needs some improvements in the application of the new

tool. There is no discussion of the statistical nature of 4D-Var, which is paramount to that method's success with real data and its being labeled a Bayesian inversion technique. We have several specific comments as to how the discussion could be made more precise and also miscellaneous technical corrections.

Major Comments:

- Section 2.3: Incremental 4D-Var is used to optimize nonlinear systems. But the CO2 tracer simulation is inherently linear. Thus, I don't really see at this point what the benefit would be of an incremental formulation, nor how updating the inner loop with an outer loop integration would provide any additional information. Thus, the importance of including both of these methods and their comparisons for $CO_2$ inversions needs further justification.

- Fig 8: As the authors surely know, adjoint sensitivities should agree with the tangent linear sensitivities to numerical precision. The differences between these sensitivities vs the finite difference sensitivities, given that the latter match the tangent linear sensitivities, is an indication that the adjoint model is not error free. For the cases tested here, the errors are manageable, yet there is no guarantee that the errors would not grow for longer simulations. The authors should thus continue to debug their adjoint code, possibly by performing this type of test around the tangent linear and adjoint code of individual physics components developed here (such as ACM2 PBL mixing). If they can not resolve the code bugs this way, they should at least perform additional tests using different receptor locations and simulations of increasing (and decreasing) length to examine how the numerical errors may be accumulating.

- p. 12, line 7: It is not required that a grid cell be both a receptor and a source to have non-zero sensitivity, as evidenced by Figures 5 and 6. The source and receptor must simply be significant (large source and large perturbation to concentration due to that particular source). Choosing them to be the same grid

cell is likely to produce good agreement between the adjoint and finite difference methods even when the advection, PBL, or convective transport adjoint may be incorrect. Did the authors choose identical grid cells for source and receptor? If so, then additional tests are required to prove that the adjoint code works as described. Additionally, validating the convective transport adjoint and tangent linear codes requires demonstration during a period when the subgrid cumulus parameterization is active and for sources and receptors near that phenomenon. The authors should present some indicator of cumulus activity in an additional figure.

- p. 12, lines 16-17: Similar to the comment above, choosing the observations to be in the lowest model layer reduces the importance of having the adjoint and tangent linear treatments of vertical mixing. The observations should be spread more thoroughly, vertically.

- From a software perspective, I'm a bit confused about the distinction between the present work and that of GH15/16, where the adjoint of the BC tracer is developed. So, essentially the update here is that BC has been changed to CO2, and convective transport has been added? On a broader note, would it be beneficial to the community to view these as two options within a single chemical 4D-Var system, rather than as two different models? I realize this likely results from development of these systems over time, in parallel, but thinking to the future I wonder if a model merge would be in order. To illustrate my point, imagine if rather than a consolidated WRF-Chem model, we had separate WRF-Chem-BC, WRF-Chem-CO2, WRF-Chem-CO, . . . etc. models. That would clearly hinder development of the tool as a whole, which shares many common elements across the different tracer simulations.

Further Comments:

1. p. 2, line 33: The reference to Streets 2013 is a bit odd, as that paper is specifically a review of remote sensing based constraints on emissions and focuses mainly on reactive trace gases and aerosols. As the present work doesn't seem geared towards remote sensing observations, some other references to literature on regional CO2 inverse modeling would be a better fit here.

2. p. 2, line 9: Probably more correct to say "instead they directly compute the product of the Jacobian with a forcing vector, which is the gradient used for optimizing the state vector."

3. p. 2, line 10: The notion that posterior error can not be calculated from a variational inversion is outdated. Posterior error can be calculated analytically using the Lanczos vectors from a CG minimization in the incremental 4D-Var framework following Fisher and Courtier (1995) - as currently done in operational weather forecast centers such as ECMWF and the UK Met Office - for minimal additional cost. Efficient posterior error estimate for non-incremental 4D-Var frameworks are described in Bousserez et al., QJRMS, 2015, including previous works on Monte Carlo (e.g., Chevallier et al., JGR, 2007) and stochastic (Rabier and Courtier, QJRMS, 1992) methods.

   Fisher M, Courtier P. 1995. Estimating the Covariance Matrices of Analysis and Forecast Error in Variational Data Assimilation, Technical Memorandum 45. ECMWF: Reading, UK.

4. The introduction states that both bottom-up and top-down approaches are used, but does not say why that is the case. It is recommended to move the first paragraph of Section 4 to the introduction.

5. p. 2., line15: The GEOS-Chem $CO_2$ 4D-Var system is also part of JPL's Carbon Monitoring System, e.g. Liu, Bowman, Lee, et al., Tellus B, 2014; Liu, Bowman, and Lee, JGR, 2016.

6. p. 2, line 18: Also Chevallier et al., JGR, 2007.

7. p. 2: 2: For regional CO2 inversions, the list isn't entirely complete, see also Alden, Miller, Gatti, et al., Global Change Biology, 2016; Chan, Chan, Ishizawa, et al., GMDD, 2016. There are others, but I think it suffices to say the literature review could be a bit more comprehensive (or, alternatively, scoped / phrased as to be more narrow).

8. p. 2, line 33: Here and in several other places, the authors use the phrase "influence function" without every having defined it.

9. Equations 3,4: define superscript $n$ in this context.

10. Equations 6,7: I understand incremental 4D-Var, but I think still the authors should rigorously define the superscript $n$ in this context for the sake of completeness, which I believe should differentiate between inner and outer loops. Also, incremental 4D-Var is usually employed with a square-root preconditioning, which I don't see here.

11. Fig 1: The way that the observations fit into this diagram doesn't make sense, since currently it implies the arrow coming out of the right side of the Simulated box passes information both to the right and left. Some separate arrows from the Observation box seem to be needed.

12. Fig 2: Despite the caption, this doesn't really show how the CG method is implemented to anyone already not familiar enough with incremental 4D-Var to know that it likes between the "no" and Tangent linear model and involves an updated estimate to the (preconditioned) increment.

13. p. 3, lines 11-13: This sentence is grammatically incorrect, the phrasing is confusing, and the conclusion is drawn weakly. What is the "potential" that online

transport based inversion systems have demonstrated? At a bare minimum, add a reference (e.g., Grell and Baklanov, 2011) or remove that statement.

14. p. 3, line 18, p .6, line 3, and p .6, line 23: The original reference for WRFPLUS is Xiao et al. (2008) [DOI: http://dx.doi.org/10.1175/2008MWR2235.1]. The version you use (v3.6) includes the work by Zhang et al. (2013) [mentioned elsewhere in your text], and should be included in these references. Huang et al. (2009) specifically used WRFPLUS for 4D-Var in WRFDA, but did not develop WRFPLUS. Barker et al. (AMS, 2005) originally developed WRFDA (for 3D-Var). These two latter references should be used as references for WRFDA. Barker et al. (2012) is an update on software development for WRFDA, and doesn't even mention WRFPLUS. The appropriate references need only be given at the first mention of these particular models, and do not need to be repeated throughout as references to the entire model. The exception is when discussing a particular aspect of those works.

15. Throughout, the author should use bold characters for vector notation (i.e., $x$, $y$, $q$, $k_{co2}$). These would be particularly illustrative on p. 10, lines 21, 24, and 31 to indicate whether the denominator or numerator of $\partial q_{co2}/\partial k_{co2}$ is a vector.

16. p. 5, line 7 and p. 14, line 23. You mention that L-BFGS-B can be used to calculate the posterior covariance, which is true although robustness of this approach with regards to the initial inverse Hessian estimate is an issue when using this algorithm, see Bousserez et al., QJRMS, 2015. As mentioned elsewhere in our review and this manuscript, Lanczos CG can also be used to estimate posterior error due to the eigen decomposition (well documented). Thus, the ability to calculate posterior error estimations is not a valid distinction between these two. Further, calculation of posterior error is not included for either method in this work. So while this could be mentioned in the introduction or discussion of future work, the methods section should only refer to methods that are actually used in

this work or ones that provide reasoning for why you used a particular approach.

17. p. 5, the term "cost function" is used 13 times on this page alone. It is suggested to reduce "cost function gradient" to "gradient".

18. A comparison of Lanczos CG and LBFGS-B based solely on cost function reduction and RMSE is not sufficient. The authors should be more instructive and explicit as to the tradeoffs between them. In regards to p. 6, line 20, and p. 14, line 21: How much less memory does Lanczos CG require for your particular application, as a percentage? Is that a good reason for choosing it over L-BFGS-B in this case? Are there ways to reduce the memory requirements of each? The most accurate Lanczos CG algorithm requires storing all basis vectors and performing full reorthogonalization after each iteration. Do you include that step? If not, why? This is a salient topic, since you discuss the loss of conjugacy later in the manuscript. Also, how can Lanczos CG be adapted for parallel computation in a way that differs from L-BFGS-B? The name of an algorithm or a reference should be included. What are the respective wall-clock times of the two methods? Lastly, It's also not clear why one is more amenable to parallel programming than the other (p. 14, line 21), as both are sequential techniques, unless that is strictly a consequence of the aforementioned memory requirements.

19. p. 6, line 20: Lanczos CG provides approximations of both the leading eigenvalues and eigenvectors (eigenmodes), not only the former.

20. p. 7, lines 8-9: Does VPRM calculate fluxes at the grid-scale in every time step? You can scale fluxes whether they are provided online or offline.

21. p. 7, line 15: So a tagging scheme for source specific $CO_2$ has been implemented as well? This might present an interesting feature for testing the adjoint sensitivities and 4D-Var system, or performing low-dimensional analytic inversions.

22. p. 7, line 32: Well, to be more precise about the wording, the adjoint is just the backward sweep, although for nonlinear systems it would need information from the forward sweep. I'm not sure what that information would be though, for a linear CO2 simulation.

23. p. 10, line 29: The emission-normalized sensitivities can also be found by dividing the full sensitivities by the emissions. Two separate simulations are not required. You might say that you calculated them this way, but to suggest "this is done" in general by a specific approach is misleading.

24. p. 11, line 5: I realize this is more of a numerical demonstration than scientific result, but it is strange to define the adjoint forcing for tower observations to be at the surface rather than tower height, as in practice these types of measurements would have a greater emissions footprint (hence the rational for using a tower …).

25. Section 3.3: It seems like accuracy should be evaluated first, before presenting the sensitivity results in section 3.2.

26. Equation 8: What value used for $\triangle x$? It can sometimes be difficult to find a perturbation value that balances truncation and roundoff error when using this equation to verify adjoint sensitivities.

27. p. 12, line 23: By assuming $\mathbf{B}^{-1} = \mathbf{0}$ and $\mathbf{R} = \mathbf{I}$, the pseudo-data case ignores how uncertainties will affect the convergence of Lanczos-CG and LBFGS-B. How would the performance of these two approaches differ with imperfect observations? With an unbiased prior? Determining the correct treatment of $\mathbf{B}$ and $\mathbf{R}$ is an active research area, which the authors do not address. Do the authors plan to explore more realistic covariance definitions in the future? At a minimum, this should be discussed in Section 4.

28. p. 12, line 32-24: Did the authors confirm a loss of conjugacy in the Lanczos basis vectors? Also, did the authors make any attempt to force conjugacy through full re-orthonormalization (e.g., Modified Gram Schmidt)? That mechanism is built in to release version 3.6 of WRFDA. While re-orthonormalization uses extra memory, that resource requirement is often very small relative to that of the model integrations. The authors should justify a decision that adds iterations to the optimization. After including full re-orthonormalization, the number of iterations for Lanczos CG to converge in each outer iteration should be proportional to the degrees of freedom (DOF) constrained by the chosen observations (see, e.g., Rodgers, 2000), entirely independent of the conjugacy issue. At that point, the necessity of multiple outer iterations would be caused by a nonlinearity in the forward model, possibly the PBL treatment or convective transport. The authors make no attempt to characterize such a nonlinearity that would necessitate using a nonlinear optimization strategy.

29. Section 3.4: While interesting and valuable, numerically, there should be some statement with regards to the unphysical nature of the test setup, to emphasize that this is strictly a numerical test and not the expected level of performance (in terms of cost function reduction or RMSE) that would be obtained in a real inversion.

30. In Section 4, the third paragraph needs a topical sentence. Also, the two sentences "We evaluated ... sensitivity." should be combined into one and made more concise.

31. p. 15, lines 1-11: While mentioning the ability to use different aggregation techniques may illuminate a budding area of research to the reader, the details given should be both accurate and concise. The authors' discussion of smoothing and aggregation error (i.e., from Turner et al., 2015) are based in the assumption that no correlation is used in $\mathbf{B}$. Taken out of context, this could be very confusing to

the reader. Full non-ambiguous coverage of that topic would require more than a paragraph, but such a description is not appropriate for this section. Indeed, large portions of Section 4 (paragraphs 4, 5, and 6) ought to be rewritten or removed. Many of the references are out of date, and do not represent the state of the science.

32. p. 15, line 18-21: Other areas to improve upon would be more accurate treatment of data and model (i.e. transport and representational) errors in $R$, and error correlations in $B$, and posterior error estimation.

33. Section 5: I'm not sure this meets the requirements of GMD, and may delay the publication of this work until the code is publicly available.

Technical Corrections:

1. Add appropriate punctuation to Eqs. 1, 2, 3, 4, 6, and 7.

2. p. 1, line 21: Remove "inversion" at the end of the sentence, as it is implied in the first half of this statement.

3. p. 2, line 16: "LDMZ" should be changed to "LMDZ"

4. p. 2, line 17: Change "inverse" (noun) to "invert" (verb).

5. p. 3, line 3: "LPDM" is undefined. Possibly define and change "Lagangian particle backward trajectory model" to "Lagangian particle dispersion model (LPDM)" on p. 2, line 33.

6. p. 4, line 12: "Where" to "where"

7. The first term in parentheses in Eqs. 3, 4, and 6 need transpose operators. Additionally, it would be less confusing if brackets and braces are used in addition to parentheses where warranted.

8. p. 6, line 9: Correct "innoviation" to "innovation"

9. p. 6, line 11: Remove "Eq. 7", since you are referencing the very next line of the text.

10. p. 6, line 20: Correct "lead" to "leading"

11. p. 7, lines 12, 16: "inner" to "inert"

12. p. 7, lines 14-16. Combine the two sentences that both state this category does not apply to CO2.

13. p. 7, line 20: "(Zhang et al.," to "Zhang el al., ("

14. p. 7, line 30: Correct "simplied" to "simplified"

15. p. 8, lines 12 and 16: Correct "inner" to "inert"

16. p. 8, line 12: Correct "use" to "uses"

17. p. 9, line 1: Change "chemistry vertical mixing" to "vertical mixing of chemical species"

18. p. 9, line 3: "dynamical" to "the dynamical"

19. p. 9, line 24: "set up" to "setup"

20. p. 10, line 5: "simulation spans" to "simulations span"

21. p. 10, line 7: "condition" to "conditions"

22. p. 11, line 3: The reference should be to Gerbig et al. (2008). Also, use the correct parenthetical format for inline references.

23. p. 11, line 3: Change "footprint at a receptor" to "footprint of a receptor"

24. p. 11, lines 9, 11, 12, 15: The figure references are off by 1.

25. p. 11, line 18: Correct "no shown" to "not shown"

26. p. 12, line 23: Correct "identify" to "identity"

27. p. 12, line 28: Correct "facotr" to "factor"

28. p. 12, line 30-31: Lanczos-CG is repeated twice. Also, use either "Lanczos CG" or "Lanczos-CG" throughout the document.

29. p. 13, lines 2 and 21: cost function needs an article, such as "the"

30. p. 13, lines 2-4: Add commas before and after "by the 30th iteration".

31. p. 13, lines 16-17: The opening to this sentence, "Starting at 2336.5 mol km-2 h-1," is confusing or out of place.

32. p. 13, lines 24: change "the Lanczos" to "Lanczos" for consistency

33. p. 13, line 26: extra "the"

34. p. 14, line 8: Change "system" to "systems"

35. p. 14, line 10: Modify, "Such configuration", which is grammatically incorrect.

36. p. 14, line 11: Change "incurring" to "requiring"

---

## Author Response (AR1)

General comments:

1) The introduction includes quite a bit of information related to offline and online mete-
orology. However, I have a hard time understanding why the online system was chosen
if the feedback of CO2 to the meteorology was ignored for the convenience. Why not
instead choose development on an offline regional model? Also, will the system be
updated periodically to account for the regular updates in the WRF model family?

*Regarding the rational for developing on-line chemistry transport based 4DVar system without
considering the $CO_2$ feedback to meteorology.*
***First**, a major benefit of using an online chemistry transport model is that it provides meteorology
fields at much finer grid spacing and time interval. For high resolution regional inverse modelings,
$CO_2$ vertical transport is primarily driven by resolved vertical wind velocity, instead of parameterized
physical schemes. Offline models, driven by archived analysis or meteorology model output with much
larger time interval, have difficult simulate chemistry transport fine grid spacing.*
***Second**, in WRF system, $CO_2$ can impact meteorology fields through the radiation schemes
(longwave/shortwave). As a regional $CO_2$ inverse modeling system, WRF-CO2 4DVar is designed to
run in short period time (hours to weeks) for constrain emission flux with observation data. For such
time span are insignificant in most cases. For the applications WRF-CO2 4DVar is designed for,
including CO2 impacts on meteorology will require a large amount of code development while offering
limited performance improvement. This can be done in the future, but it is beyond the scope of the
present paper.*

*Regarding the future WRF system update:*
*We do plan to keep developing and updating the WRF-CO2 4DVar system. In addition to inclusion of
observational operators for application with real observation data, we do plan to periodically update
the system to keep up with the WRF system. We note that many updates of WRF system are not relevant
to WRF-CO2 4DVar system, such as those for physical schemes and chemical mechanisms not used in
$CO_2$ transport. Because WRF-CO2 uses a subset of WRF system, we only need to keep updating those
relevant procedures, most of which are in the dynamical core (for advection and diffusion), convective
chemistry transport (in chemistry module), and planet boundary layer schemes that treat chemistry
transport (in physics module). The amount of work required for these updating are manageable for us.*

2) As the authors noted, meteorology is critical to the quality of CO2 transport.
Throughout the paper, I am surprised that there is no evaluation of the WRF mete-
orology, but this could be easily done considering the numerous observations available
within the CONUS domain. In fact, I do have some concerns on the WRF setup as in
specific comments 2)-3) below.

*We agree with the referee that accurate meteorology simulation is of critical importance: error in
meteorology will lead the inverse system to mistakenly assign transport error to fluxes sources. Thus it
is imperative to ensure the quality of the meteorology field when applying the system to invert real
observational data.*

*At the present stage, WRF-CO2 4DVar system does not include any observational operator (and their
TL/AD counterparts), thus it is not ready for applying real observational data yet. The objectives of the
present paper are to (1) develop and test the accuracy of the tangent linear and adjoint models, and (2)
to implement the two iterative optimization schemes and test their effectiveness with synthetic data*

*(pseudo observation). In such pseudo observation based tests, both the observed and simulated CO2 are generated by the same meteorology but different $CO_2$ flux through scaling factor).This setup ensures that meteorology is error-free, and no transport error is present in the inverse system.*

*With this said, we completely agree with the referee that meteorology must be evaluated before the system is used with real observations. In response, we conducted comparison of the meteorology simulated by WRF-CO2 forward model against CFSv2. Since CFSv2 is an analysis which assimilated a large amount of quality controlled observations, it can be used in lieu of observational data here.*
*The inverse experiments described in the manuscript span the 24 hours starting at 2011-06-02 00:00 UTC. We interpolated CFSv2 to the WRF grid, and compare the two datasets at et 6-hour interval.*

*Figure 1 shows sea level pressure and horizontal wind at first vertical level from WRF (left column) and CFSv2 (right column).*
*Figure 2 shows horizontal wind and geopotential at the $30^{th}$ vertical level.*

*These comparisons indicate WRF simulated meteorology is close to the analysis and is valid for the purpose of the pseudo observation based tests used in the present paper.*

[Figure]

*Figure 1. Sea level pressure (background) and horizontal wind at the first vertical level (arrows) as simulated by WRF (left column) and interpolated from CFSv2 (right columns). The four figures are plotted at 6-hour interval for the 24 hour simulation period (2011-06-02 00:00 UTC to 2011-06-03 00:00 UTC).*

[Figure]

1)Figure 2. Geopotential and horizontal wind at the 30$^{th}$ vertical level (arrows) as simulated by WRF (left column) and interpolated from CFSv2 (right columns). The four figures are plotted at 6-hour interval for the 24 hour simulation period (2011-06-02 00:00 UTC to 2011-06-03 00:00 UTC).

3) The comparison of L-BFGS-B and the CG has been done before. It'd be good to relate the prior results to yours and highlight any unique findings from your work.
*In Section 4of the revised manuscript, comparison between L-BFGS –B and Lanczos-CG regarding memory and computation cost are added, and related to previous research findings (Guerrette and Henze 2015).*

4) Information on computation requirement and cost would be helpful.
*Detailed information about memory requirement and walltime are added in Section 4 of the revised manuscript.*

**Additionally, here are some places where clarifications or corrections are needed:**
1) The cost functions, etc, are not quite consistent with literature on the similar topic. Vectors should be in bold.
*We accidently dropped the transpose operator equations 3,4, and 6. These are fixed.*
*All vectors in equations and inline text are in bold now.*

2) Page 9, Line 26: Was indirect soil nudging enabled when PX LSM was used? It is recommended to enable it in retrospective analysis because little testing has been done for running PX with the indirect soil nudging disabled. See the WRF users guide and literature.
*The indirect soil nudging was not used in the experiments described in the original manuscript. We greatly appreciate the referee brought it to our attention. All the new simulations reported in the revised manuscript were conducted with the indirect soil nudging activated .*

3) Page 10: Met IC/LBC from CFS on which resolution? Potential problems of down-scaling that to 48km should be discussed. Again, some model evaluation should be added.

*CFSv2 analysis data of 1x1 degree horizontal grid are used to generate the meteorology initial and boundary condition. For model evaluation, please refer our response your general comment #2, and Figure 1 and 2 of this document.*

Potential problem associated with downscaling should be added.
*For purpose of model development and testing, the simulation matches well with analysis data (Figure 1 and 2) and the meteorology are valid for testing model accuracy and inverse modeling test. We do note that more care should be exercised when WRF-CO2 4DVar is used with actual observation data and potential transport error.*

4) Page 10: any biomass burning emissions included? Does daily emission include any diurnal variability? Please include the emission amount in Section 3.1, to help understand the figure and results in Section 3.4.
*Biomass burning emission:*
*In the simulations described in the original manuscript, biomass burning emissions were not included. In the revised manuscript, we switched from using EDGAR/CASA emission flux to CarbonTracker optimized fluxes. All four fluxes (fire, biosphere, fossil fuel, and ocean) are used. So, biomass burning*

*emission is included in the simulations reported in the revised manuscript.*

***Diurnal variability:***
*Because CarbonTracker fluxes are of 3-hour interval, diurnal variability is included. However we need to point out that in the inverse experiments, emission scaling factor is applied to the mean daily value of biosphere flux. This means no diurnal variability in the inverse experiments. Our inverse experiment is constrained by error-free pseudo observations, which is an ideal configuration to prove that WRF-CO2 inverse framework works. We acknowledge that diurnal flux variability probably should to be included in application with real observations, which will require an different setting for the scaling factor (such as separate scaling factors for photosynthesis and respiration). As these considerations are beyond the scope of the present paper, they are not addressed here.*

***Emission amount:***
*Figure 3 (below) shows the sign and magnitude of daily mean biosphere flux used in all the simulations reported in the revised text.*

[Figure]

*Figure 3. Daily mean CarbonTracker biophsere CO2 flux used in calculating sensitivies and inverse modeling experiments. The daily mean value is calculated as the arithmetic mean of the 3-hourly fluxes from 2011-06-02 00:00 UTC to 2011-06-03 00:00 UTC.*

5) Page 10, line 7-8: More details on the global WRF-Chem simulation is needed. I assume it was done on a coarser resolution than 48 km. Would the sensitivity and other tests in the global domain differ much from the regional results? Some simple comparison can highlight the benefit of using a regional scale 4dvar system.

*The global WRF-chem simulation was conducted with a 256 (east-west) by 128 (north-south) grid, which is about 156x156 km in horizontal. The WRF global model code includes polar filtering procedures for the converging latitude at high latitudes. These filtering procedures are located in multiple modules in the dynamical core. Because they are only needed for global model, we did not*

*develop their tangent linear and adjoint code for WRF-CO2 4DVar (which is designed for regional inverse only).This mean we can not run WRF-Chem for an global domain inversion to compare with WRF-CO2 4DVar in regional inversion.*

*In the new experiments reported in the revised manuscript, chemistry initial and boundary conditions are **not** obtained by global WRF-Chem simulation anymore, but are from CarbonTracker $CO_2$ mole fraction (global 3x2 degree, 3-hourly interval product, 2016 version).The interpolation from CarbonTracker grid to WRF is described in Section 3.1of the revised manuscript.*

6) Figure 1 and 2 need to be cited in the text. As red is already included in your emission color scheme, I suggest using a non-red color to show the locations of towers sites in Figure 3.

*The optimization scheme diagram in Figure 1 and 2 are now cited in the text. In addition both diagrams have been improved for clarity in response to the second referee's comments.*

*Figure 3 are redrawn. In response to the second referee's comments, the sensitivity (TL/NL/AD)tests in the revised manuscript are done at the 20 towers (as receptors) and a different set of 35 locations (as sources). For clarity, we split the original Figure 3 to two separate figures: the Figure 3 marks the source and receptor locations and Figure 4. show the biosphere flux magnitude (please refer to our response to your comment #4*

7) It'd be good to show Figure 5 and 6 along with trajectory calculations as in some prior works.

*Our understanding is that trajectory calculations are carried out using Lagrangian particle dispersion models, such as LPDM, FlexPart, or Hysplit, in backward trajectory mode. As an adjoint based inverse system, WRF-CO2 4DVar does not calculate backward trajectory (it directly computes the product of the Jacobain with a forcing vector). So, we can not plot trajectory calculation. We did improve the footprint (adjoint sensitivity) plot by adding the results from receptors placed at the $10^{th}$ vertical level (in addition to the $1^{st}$ level).*

Some typos and grammar errors:
1) Page 5, line 19: according -> according to
Fixed.

2) Page 12, line 28: facotr -> factor
Fixed.

3) Page 37, Table 5 caption: givne ->given
Fixed.

**Response to Anonymous Referee #2**

The manuscript, "Development of the WRF-CO2 4DVar assimilation system", describes exactly that. The authors have created linearized model versions of the WRF-Chem CO2 transport mechanisms, validated their performance against finite difference approximations, and demonstrated their utility in a simplified pseudo-data experiment. The introduction covers much of the relevant literature necessary to get to the same starting point as the authors, and we make a few additional suggestions below. Theadjoint and tangent linear model developments are described thoroughly, and would be helpful for any person working their way through the code at a later time. The adjoint model evaluation falls a little short, and we provide some suggestions for ways it could be improved. The pseudo-data inversion test, while quite simple and unrealistic, demonstrates that the inversion framework is working. It is a first step that undoubtedly took considerable effort, but needs some improvements in the application of the new tool. There is no discussion of the statistical nature of 4D-Var, which is paramount to that method's success with real data and its being labeled a Bayesian inversion technique. We have several specific comments as to how the discussion could be mademore precise and also miscellaneous technical corrections.

We thank the referee for the time they have taken to improve the paper with their insightful and detailed comments. Below is a summary of the major work conducted in response to the referee's comments. The point-to-point response is in the following sections.

- Model code debugging: as the referee point out, the adjoint model was not error free. To address the problem, we systematically debugged the model code. Errors were isolated, identified and corrected. The evaluation through sensitivity calculation confirms that the three model components (NL/TL/AD) match as expected.

- Optimization experiment: in the revised text, synthetic observation data are from 30 vertical levels from bottom up. They were from the bottom level only in the original text.

- In the footprint calculations, receptors are now placed at $1^{st}$, $5^{th}$, and $10^{th}$ . They were placed on the $1^{st}$ level only in the original text.

- TL/AD/FD sensitivity comparison. (1) tangent linear, adjoint, and finite difference sensitivities are calculated for source and receptors cells at different locations in both horizontal and vertical). (2) The receptor cells are placed at the $1^{st}$, $5^{th}$, and $10^{th}$ vertical levels at each tower site.

- Cumulus activity indication: extra variables are implemented in the model to track when/where the convective tracer transport is activated during the simulation. This information is plotted and used to ensure there are sources and receptors located within or near the cumulus activity. As the referee pointed out, this is necessary to evaluate the accuracy of the newly developed TL/AD code of the convective chemistry transport scheme (module_ctrans_grell).

**Major Comments:**
• Section 2.3: Incremental 4D-Var is used to optimize nonlinear systems. But the
CO2 tracer simulation is inherently linear. Thus, I don't really see at this point
what the benefit would be of an incremental formulation, nor how updating the
inner loop with an outer loop integration would provide any additional information.
Thus, the importance of including both of these methods and their comparisons
for CO2 inversions needs further justification.

We thoroughly examined the model code for the linearity pointed out by the referee. We
found that WRF-CO2 is linear except at certain situation when positive definite chemistry
transport is used. Our examination shows that the ACMPBL and convective transport
(ctrans_grell) are both linear with respect to CO2. With positive definite chemistry advection,
nonlinearity can be introduced when the predicted minimum possible CO2 is negative at a
grid point. This will trigger a renormalization procedure, which is nonlinear. We confirmed
this nonlinearity through examining finite difference sensitivity around grid point where the
above mentioned renormalization is artificially triggered. In order to trigger the
renormalization, we created large horizontal CO2 gradients. We do not believe such large
CO2 gradient is very common in nature, but may be possible for wild fire emission. In the
24-hour simulation used in this paper, the renormalization was not triggered in any grid cell,
thus the system is linear.

The outer loop updating used in the original text was necessitated by the error in the adjoint
model code, and we mistakenly attributed it to the loss of conjugacy. We greatly appreciate
the referee's insightful comment. With the corrected adjoint model, Lanczos-CG based
incremental optimization does not need the outer loop update: only one outer loop iteration is
applied in all inverse modeling experiments.

Our inverse experiment results (with the corrected adjoint model, and observation at 30
vertical levels) show that Lanczos-CG converge substantially faster than L-BFGS-B.
Although we are aware that this performance difference may be specific to our experiment
setup, we consider it is necessary to include both optimization schemes in WRF-CO2 4DVar
for future applications.

• Fig 8: As the authors surely know, adjoint sensitivities should agree with the tangent
linear sensitivities to numerical precision. The differences between these
sensitivities vs the finite difference sensitivities, given that the latter match the
tangent linear sensitivities, is an indication that the adjoint model is not error free.
For the cases tested here, the errors are manageable, yet there is no guarantee
that the errors would not grow for longer simulations. The authors should
thus continue to debug their adjoint code, possibly by performing this type of test
around the tangent linear and adjoint code of individual physics components developed
here (such as ACM2 PBL mixing). If they can not resolve the code bugs
this way, they should at least perform additional tests using different receptor locations
and simulations of increasing (and decreasing) length to examine how
the numerical errors may be accumulating.

Yes, we are aware that adjoint model was not error free. We thank the referee for reminding us to correct it. Following the referee's suggestion, we debugged the individual processes in isolation. We also modified the code to test all three models in a single time step mode. Code errors were identified and corrections were made. We evaluated the updated code by comparing the adjoint sensitivity against the tangent linear and finite sensitivities. As suggested by the referee, these sensitivities are calculated with sources and receptors at different grid cells and the receptors cells are placed at multiple vertical levels. (See Fig. 2 next page for the source and receptor placement. We also ensured that there are sources and receptors placed within or near cumulus activities for testing the convective transport code. (See Fig. 3 for cumulus indicator).

[Figure]

*Figure 1. Comparison between tangent linear sensitivity (x-axis) and adjoint sensitivity (y-axis). The sensitivities are organized into three groups for receptors placed at the $1^{st}$, $5^{th}$, and $10^{th}$ vertical levels. Because there 35 sources (red starts Fig. 2) and 20 tower sites (red triangles in Fig. 2), there are 700 (35x20) pairs of tangent linear and adjoint sensitivities for each group. Sensitivities with absolute values less than $10^{-10}$ are not included in comparison, resulting in 295, 292, and 305 pairs of comparisons. The solid lines in each figure are the 1:1 lines. Because the values are very close, we summarize each figure by the minimum and maximum difference between the tangent linear and adjoin sensitivities, instead of the slope and r-squared.*

• p. 12, line 7: It is not required that a grid cell be both a receptor and a source
to have non-zero sensitivity, as evidenced by Figures 5 and 6. The source and
receptor must simply be significant (large source and large perturbation to concentration
due to that particular source). Choosing them to be the same grid cell is likely to produce
good agreement between the adjoint and finite difference methods even when the advection,
PBL, or convective transport adjoint may be incorrect. *Did the authors choose identical grid
cells for source and receptor?* If so, then additional tests are required to prove that the adjoint
code works as described. Additionally, validating the convective transport adjoint and
tangent linear codes requires demonstration during a period when the subgrid cumulus
parameterization is active and for sources and receptors near that phenomenon. The authors
should present some indicator of cumulus activity in an additional
figure.

**Placement of the source and receptor cells**: In the original manuscript, the source and
receptors were the same grid cells. They were the bottom level grid cells where the 20 towers
are located.
We agree that code should be tested with sources and receptors placed at different grid cells
as long as there are discernible impact during the simulation period. To address this, we
conducted the sensitivity calculations with the updated model in a more systematic approach:
(1) Receptors are still placed at the 20 tower sites in horizontal, but at each tower site, 3
receptors are placed at different vertical levels: level 1, 5, and 10. (2) Sources are at a
different set of 35 grid cells placed around the receptors. The placement of sources and
receptors used in the revised manuscript is in Figure 2 below.

[Figure]

Figure 2. Placement of the sources (blue stars) and receptors (red triangles). Receptors are the
placed at $1^{st}$, $5^{th}$, and $10^{th}$ vertical levels of grid cell of the 20 towers (Table 4 of the
manuscript). Being the surface flux source, all sources are place at $1^{st}$ vertical level.

**Indicator of cumulus activities:**

To ensure some sources and receptors are placed within or near cumulus activities, we implemented an counting mechanism within the convective transport code (module_ctrans_grell.F in the chemistry directory). In WRF, whether chemistry species at each grid cell is vertically transported at a given time step by the cumulus process is determined by a number of tests and marked by a pair of flags (one for deep convection and another for shallow convection). These two flags are reset at each time step. We added two variables to track the two flags across time steps: each time the convective transport process is triggered at a grid cell, its count increases by one. We refer the new variables "convective tracer transport trigger count" and used it to examine the cumulus activities. Comparison of Figure 3 (below) and Figure 1 confirms there are sources and receptors placed within or near cumulus activities.

[Figure]

Fiure 3. Convective tracer transport trigger count plotted at 6-hour intervals at vertical level 5. These counts are vertical level specific as the convective transport activations are determined for each vertical level. The figures show that deep convections are trigger during a large portion of the 24-hour simulation at the Pacific northwest and Midwest (center round Iowa). A comparison with source/receptor cells placement in Figure 1 confirms that there are

sources/receptors placed around cumulus activities, and thus tangent linear and adjoint code of the convective transport are indeed tested.

• p. 12, lines 16-17: Similar to the comment above, choosing the observations to be in the lowest model layer reduces the importance of having the adjoint and tangent linear treatments of vertical mixing. The observations should be spread more thoroughly, vertically.

We agree with the referee.
To address this issue, new pseudo observation data are generated from the forward model run at the first 30 vertical levels (out of a total of 50 levels). This means that there are 30 observations at each horizontal grid. We emphasize this setting is for the sole purpose of testing the inverse system with ideal error-free synthetic data. The inverse experiment results are shown in Fig. 10-13 of the revised manuscript.

• From a software perspective, I'm a bit confused about the distinction between the present work and that of GH15/16, where the adjoint of the BC tracer is developed. So, essentially the update here is that BC has been changed to CO2, and convective transport has been added? On a broader note, would it be beneficial to the community to view these as two options within a single chemical 4D-Var system, rather than as two different models? I realize this likely results from development of these systems over time, in parallel, but thinking to the future I wonder if a model merge would be in order. To illustrate my point, imagine if rather than a consolidated WRF-Chem model, we had separate WRF-Chem-BC, WRF-Chem-CO2, WRF-Chem-CO, . . . etc. models. That would clearly hinder development of the tool as a whole, which shares many common elements across the different tracer simulations.

Yes, we developed WRFCO2 4dvar system as part of an effort proposing to NASA's carbon science program. Our development is in parallel with G15/16, and no collaboration has been involved yet.
We agree with the referee that coordinated code development will benefit the community. We will contact Dr. Henze's group for collaborating on future code consolidation/merging.

**Further Comments:**
1. p. 2, line 33: The reference to Streets 2013 is a bit odd, as that paper is specifically a review of remote sensing based constraints on emissions and focuses mainly on reactive trace gases and aerosols. As the present work doesn't seem geared towards remote sensing observations, some other references to literature on regional CO2 inverse modeling would be a better fit here.

This reference has been removed.

2. p. 2, line 9: Probably more correct to say "instead they directly compute the product of the Jacobian with a forcing vector, which is the gradient used for optimizing

the state vector."

This sentence has been corrected following the referee's suggestion.

3. p. 2, line 10: The notion that posterior error can not be calculated from a variational inversion is outdated. Posterior error can be calculated analytically using the Lanczos vectors from a CG minimization in the incremental 4D-Var framework following Fisher and Courtier (1995) - as currently done in operational weather forecast centers such as ECMWF and the UK Met Office - for minimal additional cost. Efficient posterior error estimate for non-incremental 4D-Var frameworks are described in Bousserez et al., QJRMS, 2015, including previous works on Monte Carlo (e.g., Chevallier et al., JGR, 2007) and stochastic (Rabier and Courtier, QJRMS, 1992) methods.
Fisher M, Courtier P. 1995. Estimating the Covariance Matrices of Analysis and Forecast Error in Variational Data Assimilation, Technical Memorandum 45. ECMWF: Reading, UK.

We thank the referee for correcting us. This statement about posterior error calculation has been corrected accordingly.

4. The introduction states that both bottom-up and top-down approaches are used, but does not say why that is the case. It is recommended to move the first paragraph of Section 4 to the introduction.

The related paragraphs have been rearranged following the suggestion.

5. p. 2., line15: The GEOS-Chem CO2 4D-Var system is also part of JPL's Carbon Monitoring System, e.g. Liu, Bowman, Lee, et al., Tellus B, 2014; Liu, Bowman, and Lee, JGR, 2016.

Thanks for pointing this out. The text has been modified accordingly.

6. p. 2, line 18: Also Chevallier et al., JGR, 2007.

Thanks. This has been fixed.

7. p. 2: 2: For regional CO2 inversions, the list isn't entirely complete, see also Alden, Miller, Gatti, et al., Global Change Biology, 2016; Chan, Chan, Ishizawa, et al., GMDD, 2016. There are others, but I think it suffices to say the literature review could be a bit more comprehensive (or, alternatively, scoped / phrased as to be more narrow).

Agree. The regional CO2 inversion review has been strengthened with the following the two additional literatures.

8. p. 2, line 33: Here and in several other places, the authors use the phrase "influence

function" without every having defined it.

Definition of the influence function has been added.

9. Equations 3,4: define superscript n in this context.
Definition for superscript n has been added.

10. Equations 6,7: I understand incremental 4D-Var, but I think still the authors should
rigorously define the superscript n in this context for the sake of completeness,
which I believe should differentiate between inner and outer loops. Also, incremental
4D-Var is usually employed with a square-root preconditioning, which I
don't see here.

Text has been added to define the superscript n and how it changes differently within the inner
and outer loop of the incremental optimization.
The square-root preconditioning is not applied in the pseudo-data based inversion
experiments. As explained in Section 3.4 of the manuscript, background error matrix is set to
infinity and observation error matrix is set to the identity matrix. This is realized in the code
by setting the cost function equal to the observation cost function, and the cost function
gradient to the observation cost function gradient. For this setup, we believe the
preconditioning does not need to be applied.

11. Fig 1: The way that the observations fit into this diagram doesn't make sense,
since currently it implies the arrow coming out of the right side of the Simulated
box passes information both to the right and left. Some separate arrows from the
Observation box seem to be needed.

Two separate arrows has been added out of the modeled and simulated box to avoid the
possible confusion in data flow direction. (See Fig 1 in the updated manuscript.)

12. Fig 2: Despite the caption, this doesn't really show how the CG method is implemented
to anyone already not familiar enough with incremental 4D-Var to know
that it likes between the "no" and Tangent linear model and involves an updated
estimate to the (preconditioned) increment.

We added to two addition boxes (for residual vector and updated gradient respectively)
between the tangent linear and adjoint model. This helps explain the data flow between the
the TL an AD models in the inner loop. Also the 'exit' box is changed to 'inner loop
converged' box to emphasize the condition to exit the inner loop. (See Fig. 2 of the update
manuscript.)

13. p. 3, lines 11-13: This sentence is grammatically incorrect, the phrasing is confusing,
and the conclusion is drawn weakly. What is the "potential" that online transport based
inversion systems have demonstrated? At a bare minimum, add a reference (e.g., Grell and
Baklanov, 2011) or remove that statement.

We chose to have the statement removed.

14. p. 3, line 18, p .6, line 3, and p .6, line 23: The original reference for WRFPLUS
is Xiao et al. (2008) [DOI: http://dx.doi.org/10.1175/2008MWR2235.1]. The version
you use (v3.6) includes the work by Zhang et al. (2013) [mentioned elsewhere
in your text], and should be included in these references. Huang et al.
(2009) specifically used WRFPLUS for 4D-Var in WRFDA, but did not develop
WRFPLUS. Barker et al. (AMS, 2005) originally developed WRFDA (for 3D-Var).
These two latter references should be used as references for WRFDA. Barker et
al. (2012) is an update on software development for WRFDA, and doesn't even
mention WRFPLUS. The appropriate references need only be given at the first
mention of these particular models, and do not need to be repeated throughout
as references to the entire model. The exception is when discussing a particular
aspect of those works.

We really appreciate the referee to clear this up for us. The references are fixed.

15. Throughout, the author should use bold characters for vector notation (i.e., x, y,
q, kco2). These would be particularly illustrative on p. 10, lines 21, 24, and 31 to
indicate whether the denominator or numerator of @qco2/@kco2 is a vector.
All vectors in the equations and inline text have been changed to bold face characters. At
place when a symbol can be either vector or scalar, it is kept as non-bold character.

16. p. 5, line 7 and p. 14, line 23. You mention that L-BFGS-B can be used to calculate
the posterior covariance, which is true although robustness of this approach
with regards to the initial inverse Hessian estimate is an issue when using this
algorithm, see Bousserez et al., QJRMS, 2015. As mentioned elsewhere in our
review and this manuscript, Lanczos CG can also be used to estimate posterior
error due to the eigen decomposition (well documented). Thus, the ability
to calculate posterior error estimations is not a valid distinction between these
two. Further, calculation of posterior error is not included for either method in this
work. So while this could be mentioned in the introduction or discussion of future
work, the methods section should only refer to methods that are actually used in
this work or ones that provide reasoning for why you used a particular approach.

We thank the referee for correct us on this issue. The text related to posterior covariance
calculation between the two optimization schemes have been removed from the manuscript.

17. p. 5, the term "cost function" is used 13 times on this page alone. It is suggested
to reduce "cost function gradient" to "gradient".
"Cost function gradient" has been change to "gradient" through the text except where full
term is needed to avoid ambiguity.

18. A comparison of Lanczos CG and LBFGS-B based solely on cost function reduction
and RMSE is not sufficient. The authors should be more instructive and
explicit as to the tradeoffs between them. In regards to p. 6, line 20, and p. 14,

line 21: How much less memory does Lanczos CG require for your particular application, as a percentage? Is that a good reason for choosing it over L-BFGS-B in this case? Are there ways to reduce the memory requirements of each? The most accurate Lanczos CG algorithm requires storing all basis vectors and performing full reorthogonalization after each iteration. Do you include that step? If not, why? *This is a salient topic*, since you discuss the loss of conjugacy later in the manuscript. Also, how can Lanczos CG be adapted for parallel computation in a way that differs from L-BFGS-B? The name of an algorithm or a reference should be included. What are the respective wall-clock times of the two methods? Lastly, It's also not clear why one is more amenable to parallel programming than the other (p. 14, line 21), as both are sequential techniques, unless that is strictly a consequence of the aforementioned memory requirements.

(1) Regarding the comparison between L-BFGS-B and Lanczos CG, we added discussion about the memory requirement, and parallel implementation related issue in Section 4.
(2) Regarding the reorthogonalization. Yes the reorthogonaliation is implemented in the code used in the original optimization experiment. The referee is correct that the degradation of the Lanczos CG with increased inner loop iteration was not caused by the loss of conjugacy, but the error in the adjoint model (and thus the calculated gradient vector). After we corrected the adjoint model, the need for the second outer loop does not exist anymore.
(3) Walltime used in our experiment with the two schemes are documented and added in the text.
(4) References for the L-BFGS-B Fortran code (Algorithm 788) compiled in WRF-CO2 4DVar is added.

19. p. 6, line 20: Lanczos CG provides approximations of both the leading eigenvalues and eigenvectors (eigenmodes), not only the former.

Thanks. This statement has been corrected.

20. p. 7, lines 8-9: Does VPRM calculate fluxes at the grid-scale in every time step? You can scale fluxes whether they are provided online or offline.

We agree with the referee that the fluxes can be scaled whether they are from offline data files or calculated by online model (VPRM). Because running the VPRM model requires additional datasets (satellite derived vegetation indexes and land cover classification maps) and some parameter tuning, we choose to use offline CarbonTracker $CO_2$ fluxes instead. Both methods are valid, but using the offline files allows us to focus on the core code development by avoiding some extra input data preparation.

21. p. 7, line 15: So a tagging scheme for source specific CO2 has been implemented as well? This might present an interesting feature for testing the adjoint sensitivities and 4D-Var system, or performing low-dimensional analytic inversions.

The present model code does not include tagging scheme for specific $CO_2$ sources. But as the chemistry 4d variable in all three models (NL/TL/AD) includes separate variables for each individual fluxes, tagging scheme can be implemented with some minor code modification in the future development.

22. p. 7, line 32: Well, to be more precise about the wording, the adjoint is just the backward sweep, although for nonlinear systems it would need information from the forward sweep. I'm not sure what that information would be though, for a linear $CO_2$ simulation.

The statement about the adjoint and forward/backward sweep has been revised.
Regarding the referee's question about what information from the forward sweep would be needed by the adjoint (the backward sweep): (1) Meteorology state variables: In the dynamical core, advection and diffusion of chemistry species are carried in each of the three sub-steps of the Runge-Kuta loop. At a given time step, at the start of the backward sweep, only the meteorology state variables at the last sub-step is available while all three all needed. This requires the forward sweep to save (push to local stack) the meteorology at each sub-step to be used in the backward sweep. (2) $CO_2$ mixing ratio: as explained in our response to the referee's major comment #1, nonlinearity can be present in WRF-$CO_2$ when the positive definite advection predicts the minimum $CO_2$ mixing at a given grid cell to be negative and trigger the renormalization. When such nonlinearity occurs, cost function gradient depends on not only perturbation but also background value of $CO_2$.

23. p. 10, line 29: The emission-normalized sensitivities can also be found by dividing the full sensitivities by the emissions. Two separate simulations are not required. You might say that you calculated them this way, but to suggest "this is done" in general by a specific approach is misleading.

We agree and the text has been revised to avoid misleading the readers.

24. p. 11, line 5: I realize this is more of a numerical demonstration than scientific result, but it is strange to define the adjoint forcing for tower observations to be at the surface rather than tower height, as in practice these types of measurements would have a greater emissions footprint (hence the rational for using a tower . . . ).
We agree. We conducted new experiment with the improved code, and set the adjoint forcing at the 1$^{st}$ and 10$^{th}$ vertical level of the WRF model grid. Footprint figures (Figs 8 of the revised manuscript) are redrawn with the new simulation results. We also noted the difference of footprints between the adjoint forcing at the 1$^{st}$ and 10$^{th}$ levels, as the referee pointed out in his comment.

[Figure]

*Figure 4. The adjoint sensitivities (footprint) of tower sites at Centerville, Iowa (top panels) and WLEF, Wisconsin (lower panels). At each site, the adjoint sensitivities are calculated twice: one with receptor placed a the $1^{st}$ vertical level, and another at the $10^{th}$ level.*

25. Section 3.3: It seems like accuracy should be evaluated first, before presenting the sensitivity results in section 3.2.

The order has been switched so that the model accuracy evaluation is presented before the sensitivity spatial pattern.

26. Equation 8: What value used for delta_x? It can sometimes be difficult to find a perturbation value that balances truncation and roundoff error when using this equation to verify adjoint sensitivities.

The delta_x used in final calculation is 0.1. Prior to the final calculation for finite difference sensitivity, we conducted test using delta_x ranging from 0.01 to 1.0 to assess the impact of

magnitude of delta_x. The results indicate there is virtually no difference between 0.01, 0.1, and 1.0 regarding the finite difference value. We attribute this to the fact that $CO_2$ tracer transport is linear. The impact of delta_x on the finite difference sensitivity is documented in Figure 5 below. This figure is included in the supplement document, but not in the manuscript.

[Figure]

Figure 5. Impacts of the delta_x on finite difference sensitivity calculation. Finite difference of a source is calculated three times, with delta_x set to 1.0, 0.1, and 0.01 respectively. The results are compared grid cell to grid cell. Figure (c) is the histogram of difference in finite difference sensitivity calculated with delta_x=0.01 and delta_x=0.1. Figure (d) is the histogram of difference between detal_x=0.01 and 1.0. Both the scatterplots and histograms show the difference in finite difference caused by delta_x is negligible.

27. p. 12, line 23: By assuming B−1 = 0 and R = I, the pseudo-data case ignores how uncertainties will affect the convergence of Lanczos-CG and LBFGS-B. How would the performance of these two approaches differ with imperfect observations? With an unbiased prior? Determining the correct treatment of B and R is an active research area, which the authors do not address. Do the authors plan to explore more realistic covariance definitions in the future? At a minimum, this should be discussed in Section 4.

*We agree with the referee that these are very important issues when applying the system to actual observation data. As the referee pointed out, treatment of background and observation error covariance is an active research area. Our objective of the present paper is to develop and test the TL/AD model and the optimization scheme. In-depth discussion on how the two optimization schemes perform with actual data is beyond the present paper's cope. But we did add a statement to remind the readers the nature of the pseudo-data based inverse experiments.*

28. p. 12, line 32-24: Did the authors confirm a loss of conjugacy in the Lanczos basis vectors? Also, did the authors make any attempt to force conjugacy through full re-orthonormalization (e.g., Modified Gram Schmidt)? That mechanism is built in to release version 3.6 of WRFDA. While re-orthonormalization uses extra memory, that resource requirement is often very small relative to that of the model integrations. The authors should justify a decision that adds iterations to the optimization. After including full re-orthonormalization, the number of iterations for Lanczos CG to converge in each outer iteration should be proportional to the degrees of freedom (DOF) constrained by the chosen observations (see, e.g., Rodgers, 2000), entirely independent of the conjugacy issue. At that point, the necessity of multiple outer iterations would be caused by a nonlinearity in the forward model, possibly the PBL treatment or convective transport. The authors make no attempt to characterize such a nonlinearity that would necessitate using a nonlinear optimization strategy.

*In the original text, we did not examine conjugacy. The loss of conjugacy was a mere guess, and turned out to be a wrong one. We really appreciate the referee pointing it out. As we explained in debugging the adjoint model, the degradation of the incremental inner loop was caused by the inaccuracy of adjoint model as opposed to the loss of conjugacy.*

*Yes, re-orthonormalization is implemented WRF-CO2 following the WRFDA code. This means that loss of conjugacy was not possible, but we did not realize this while writing the original manuscript. This has been corrected in the revision.*

[Figure]

Figure 6. The results of the inverse experiment (Case 1). Reduction of the cost function and gradient norm are expresses as ratio to their respective starting value. Only one outer loop is used in the Lanczos-CG optimization.

29. Section 3.4: While interesting and valuable, numerically, there should be some statement with regards to the unphysical nature of the test setup, to emphasize that this is strictly a numerical test and not the expected level of performance (in terms of cost function reduction or RMSE) that would be obtained in a real inversion.

*Thanks for pointing this out. We added description about the unrealistic nature of the experiment setup and that real observation data application will need more careful treatment of the errors.*

30. In Section 4, the third paragraph needs a topical sentence. Also, the two sentences "We evaluated ... sensitivity." should be combined into one and made more concise.
A topic sentence has been added and the two sentences have been combined.

31. p. 15, lines 1-11: While mentioning the ability to use different aggregation techniques may illuminate a budding area of research to the reader, the details given should be both accurate and concise. The authors' discussion of smoothing and aggregation error (i.e., from Turner et al., 2015) are based in the assumption that no correlation is used in B. Taken out of context, this could be very confusing to the reader. Full non-ambiguous coverage of that topic would require more than a paragraph, but such a description is not appropriate for this section. Indeed, large portions of Section 4 (paragraphs 4, 5, and 6) ought to be rewritten or removed. Many of the references are out of date, and do not represent the state of the science.

The discussion about aggregation and errors has been removed in the revised manuscript, as they are not central to the present paper's objectives. In its place, we added detailed discussion about memory and computation requirement of the two optimization schemes.

32. p. 15, line 18-21: Other areas to improve upon would be more accurate treatment of data and model (i.e. transport and representational) errors in R, and error correlations in B, and posterior error estimation.

Thanks! These suggestions have been added in the revised manuscript.

33. Section 5: I'm not sure this meets the requirements of GMD, and may delay the publication of this work until the code is publicly available.

Accept. The source code has been submitted to zenodo at https://doi.org/10.5281/zenodo.839260
 Future development will be made available to the public access in the same fashion too.

**Technical Corrections:**
1. Add appropriate punctuation to Eqs. 1, 2, 3, 4, 6, and 7.

Fixed.

2. p. 1, line 21: Remove "inversion" at the end of the sentence, as it is implied in the first half of this statement.
Fixed.

3. p. 2, line 16: "LDMZ" should be changed to "LMDZ"
It is corrected.

4. p. 2, line 17: Change "inverse" (noun) to "invert" (verb).
It is fixed.

5. p. 3, line 3: "LPDM" is undefined. Possibly define and change "Lagangian particle backward trajectory model" to "Lagangian particle dispersion model (LPDM)" on p. 2, line 33.
Definition for LPDM is added, and the Lagrangian particle backward trajectory model is changed to LPDM.

6. p. 4, line 12: "Where" to "where"
Fixed.

7. The first term in parentheses in Eqs. 3, 4, and 6 need transpose operators. Additionally, it would be less confusing if brackets and braces are used in addition to parentheses where warranted.
Thanks for pointing out the missing transpose operators. They are added. The two equation presentation has been improved by using brackets and braces.

8. p. 6, line 9: Correct "innoviation" to "innovation"
Fixed.

9. p. 6, line 11: Remove "Eq. 7", since you are referencing the very next line of the text.
Fixed.

10. p. 6, line 20: Correct "lead" to "leading"
Fixed

11. p. 7, lines 12, 16: "inner" to "inert"
Fixed

12. p. 7, lines 14-16. Combine the two sentences that both state this category does not apply to CO2.
Fixed

13. p. 7, line 20: "(Zhang et al.," to "Zhang el al., ("

Fixed

14. p. 7, line 30: Correct "simplied" to "simplified"
Fixed

15. p. 8, lines 12 and 16: Correct "inner" to "inert"
Fixed

16. p. 8, line 12: Correct "use" to "uses"
Fixed

17. p. 9, line 1: Change "chemistry vertical mixing" to "vertical mixing of chemical species"
Fixed

18. p. 9, line 3: "dynamical" to "the dynamical"
Fixed

19. p. 9, line 24: "set up" to "setup"
Fixed

20. p. 10, line 5: "simulation spans" to "simulations span"
Fixed

21. p. 10, line 7: "condition" to "conditions"
Fixed

22. p. 11, line 3: The reference should be to Gerbig et al. (2008). Also, use the correct parenthetical format for inline references.
The Reference is fixed.

23. p. 11, line 3: Change "footprint at a receptor" to "footprint of a receptor"
Fixed

24. p. 11, lines 9, 11, 12, 15: The figure references are off by 1.
Fixed

25. p. 11, line 18: Correct "no shown" to "not shown"
Fixed

26. p. 12, line 23: Correct "identify" to "identity"
Fixed

27. p. 12, line 28: Correct "facotr" to "factor"
Fixed

28. p. 12, line 30-31: Lanczos-CG is repeated twice. Also, use either "Lanczos CG" or "Lanczos-CG" throughout the document.
Fixed.

29. p. 13, lines 2 and 21: cost function needs an article, such as "the"
Fixed.

30. p. 13, lines 2-4: Add commas before and after "by the 30th iteration".
Fixed

31. p. 13, lines 16-17: The opening to this sentence, "Starting at 2336.5 mol km-2 h-1," is confusing or out of place.
The sentence has been rephrased.

32. p. 13, lines 24: change "the Lanczos" to "Lanczos" for consistency
Fixed.

33. p. 13, line 26: extra "the"
Fixed.

34. p. 14, line 8: Change "system" to "systems"
Fixed

35. p. 14, line 10: Modify, "Such configuration", which is grammatically incorrect.
Text is modified to correct the grammatical error.

36. p. 14, line 11: Change "incurring" to "requiring"
Fixed

A list of major changes made in the revised manuscript

- Model code debugging: as the referee point out, the adjoint model was not error free. To address the problem, we systematically debugged the model code. Errors were isolated, identified and corrected. The evaluation through sensitivity calculation confirms that the three model components (NL/TL/AD) match as expected.

- Optimization experiment: in the revised text, synthetic observation data are from 30 vertical levels from bottom up. They were from the bottom level only in the original text.

- In the footprint calculations, receptors are now placed at $1^{st}$, $5^{th}$, and $10^{th}$ . They were placed on the $1^{st}$ level only in the original text.

- TL/AD/FD sensitivity comparison. (1) tangent linear, adjoint, and finite difference sensitivities are calculated for source and receptors cells at different locations in both horizontal and vertical). (2) The receptor cells are placed at the $1^{st}$, $5^{th}$, and $10^{th}$ vertical levels at each tower site.

- Cumulus activity indication: extra variables are implemented in the model to track when/where the convective tracer transport is activated during the simulation. This information is plotted and used to ensure there are sources and receptors located within or near the cumulus activity. As the referee pointed out, this is necessary to evaluate the accuracy of the newly developed TL/AD code of the convective chemistry transport scheme (module_ctrans_grell).

- Chemistry initial and boundary conditions used in the simulations have been changed to CarbonTrack2016 CO2 mol fraction.

- The four CO2 fluxes (fossil fuel, fire, biosphere, and ocean) have been changed to CarbonTracker2016 fluxes.

- Comparison has been conducted between WRF simulated meteorology and that interpolated from CFSv2.

[revised manuscript text omitted]

---

## Author Response (AR2)

**(1) Summary of major changes in this revision**

Major changes made in this round of revision include the following:

- In section 1 (Introduction), we added statements to explain the difference between the online and offline chemistry transport models. This provides a detailed answer to the second reviewer's question about why we developed WRFCO2 4DVar without considering the feedback from CO2 to the meteorology.

- In section 3.1 (Model setup), we added:
    - A discussion of the potential problems of downscaling coarse resolution global reanalysis data used as WRFCO2's initial and boundary conditions.

    - evaluations of WRF simulated meteorology fields (horizontal wind at surface and upper level air) using in-situ measurements (surface stations and radiosonde). Quantitative summaries of the evaluation are provided in error statistics and two new figures: Fig. 6 for the surface level, and Fig. 7 for 500 hPa.

- In section 3.3 (Spatial patterns of adjoint sensitivities), we added backward trajectory simulations using the HYSPLIT Lagrangian model. The resulting trajectories are plotted in a new figure (Fig. 9) to accompany the adjoint sensitivity figure (Fig. 8). This provides a more comprehensive view of the source-receptor relationship as suggested by the second reviewer.

- In section 4 (Summary and outlook), we added an explanation about our strategy for future development of WRFCO2 4DVar to keep up with new updates to the WRF system.

Reviewer #1

*The authors have done a commendable job in their revisions. They have thoroughly addressed my previous comments, with careful attention to detail, and have added much more rigorous testing and evaluation of their model to this paper, as well as additional discussions. The revised work is more accurate, and more complete. I only have a few small comments and technical corrections they could make before publishing, without need for further review. Page.Line numbers are in reference to the track-changes version submitted for review.*

Thanks! We greatly appreciate the reviewer's insightful comments, which helped us in improving the model code and manuscript presentation.

*Response to referee #1 comments are fine, but it does not draw from any text in the manuscript, original or revised. Some content should be added to the introduction so that these questions do not persist for subsequent readers.*

Agree. In this round of revision, we incorporated addition statements and paragraphs in the manuscript to fully address the comments from referee#1. Please refer to our point-by-point response to the referee, which we think better documents the changes made to the manuscript compared with the previous response to the reviewer.

*6.10 - 6.12: If you're not going to show the results, then they need to be described more rigorously and quantitatively. Just saying that they "closely match" is not sufficiently informative.*

Agree. Responding to the referee's suggestion to use in-situ measurements to provide quantitative evaluation, we compared WRF's simulated surface and upper level winds against weather station and radiosonde measurements. RMSE, relative error, and mean difference are provided for the wind speed and wind direction comparison. Please refer to our point-by-point response to the reviewer's comments.

*2.4: Amazon —> the Amazon*
Fixed
*2.5: applied regional … approach. —> applied a regional… approaches.*
Fixed
*3.34: a L-BFGS-B —> an L-BFGS-B*
Fixed
*Eq. 1: remove "\times"*
Fixed
*Eq 4: since the authors changed K to k in the text, I'm not sure what K in the equation means.*
The lower case k has been changed to the upper case, and the text reads as "*The summation K indicates …*"
*5.4: of conjugate —> of the conjugate*

Fixed

5.15: it's —> its

Fixed

5.27: it's —> its

Fixed

6.17: and gradient are —> the gradient is

Fixed

11.10: conducted comparison —> conducted comparisons

Fixed

11.15: sensitivity calculation —> the sensitivity calculations

Fixed

11.19: resulting to —> resulting in

Fixed

13:32 matches —> agrees with

Fixed

13.32 souhernly

Fixed

14.2: is in much —> is a much

Fixed

14.20: plotS

Fixed

17.16: How long is "slightly longer"? 1%? 10%?

The text has been changed to "about 10% more wall time to…."

Page 17 and 18: You mean "processor core" instead of "process", throughout.

Fixed

18.15: extra call of …trajectory —> extra calls of the… the trajectory

Fixed

18.23: trajectory —> the trajectory

Fixed

Reviewer #2

We greatly appreciate the referee's time and effort to help improve our manuscript. We provide point-by-point responses to his/her further comments in the following sections. The referee's comments are in red, and our responses are in black.

*Regarding complying with GMD requirements--Although the author added version 1.0 in the title, they also need to specify the version of WRF(WRFPLUS) used in this development for our information. Furthermore, they should introduce us the flexibility of this development to future versions of WRF(WRFPLUS). Sharing the code to us at this moment and expanding the future development plans are helpful. Unfortunately, this report is limited to the quality of this manuscript and does not include evaluation of the code.*

Response:

1.  Thanks for pointing out the version number for WRF/WRFPLUS. In the abstract section of the revised manuscript, we specified the version of WRF (v3.6) and WRFPLUS (v3.6) that were used to develop our code (WRFCO2 4DVar v1.0).

2.  Regarding the future development of WRFCO2 4DVar to keep up with WRF/WRFPLUS updates, we added the following statements in the "Summary and outlook" section.
    *"In addition, we also plan to periodically update the WRFCO2 4DVar system to keep up with WRF system updates. Such updates will mainly consist of replacing the forward model with the updated WRF code, and developing the tangent linear and adjoint code for the relevant updated procedures. As the variable dependence analysis (Section 2.4.2) indicates that the tangent linear and adjoint code are only needed for a portion of WRF procedures, the amount of work required for updating WRFCO2 4DVar is manageable. In addition, future development of WRFCO2 4DVar will also be dependent on updates to WRFPLUS, which has always been updated along with WRF."*

3.  The source code of WRFCO2 4DVar v1.0 has been posted for free public access at https://doi.org/10.5281/zenodo.839260. This has been done in our last round of revision. The background, technical details, and evaluations of the code are documented in this manuscript. In addition, we will responsive to questions from potential users and provide assistance if needed.

*I am glad that the authors have corrected a few obvious mistakes in the previous version that I pointed out. However, the answers to several of my initial comments and questions are not satisfactory. At least one more iteration is needed before this paper can be considered for publication. Specifically, my further comments are:*

Response:

We appreciate the referee's patience, and we have better addressed his/her comments and questions this time in our response below:

*Model IC/BC and evaluation: thanks for adding the WRF-CFSv2 comparisons. Plotting them side by side is such a qualitative illustration; please quantify their differences as well. Considering that CFSv2 data are on a coarser resolution than WRF, and are used as WRF IC/BC in this work, a more convincing way to evaluate WRF (or/and CFSv2) is to compare them with in-situ measurements. There are plenty of them over the US. You can use NCEP composite observations, for example.*

Response:

We agree with the referee that using in-situ measurements to evaluate WRF is more convincing since CFSv2 is used for IC/BC in the simulation. For this revised manuscript, we evaluated WRF simulated wind fields using in-situ measurements and have provided a quantitative summary of evaluation results. Wind fields (wind speed and direction) are chosen for evaluation because they are the meteorological phenomena that most directly impacts tracer transport.

The following four paragraphs and Figures 2 and 4 are added to the revised manuscript. Figure 1 and 3 are included in the supplement material.

*"In order to be useful for applications which employ real observational data, WRF-CO2 4DVar requires accurate simulation of the meteorological fields by the forward model, in addition to accurate tangent linear and adjoint models. Because transport error can only be partially accounted for in the 4DVar system through the observation error variance, it is imperative to minimize errors due to inaccurate simulation of meteorological processes as much as possible. Although the present paper uses pseudo-observation data (which have zero transport error by definition) in its inversion experiments, future applications with true observational data will require vigorous evaluation of the model simulated meteorology and associated transport error. In the following, the forward model simulated horizontal winds at the surface and 500 hPa constant pressure surface are evaluated using in-situ measurements from weather stations and radiosondes."*

*"For the surface level, WRF simulated 10m winds are compared against surface weather station measurements archived in the NOAA Integrate Surface Dataset (Smith et al., 2011). Hourly*

*surface wind measurements from more than 2,000 stations within the WRF domain are used for the evaluation. Comparisons of wind speed and wind direction are carried out at the top of each hour during the 24 hour simulation period starting at 00:00 UTC 02 June 2011. Excluding missing observations, this results in 31,745 valid data pairs, which are summarized in the histograms of Fig. 6. RMSE for the hourly wind speed is 2.16 m s$^{-1}$ and the mean difference in the hourly wind direction is 29.4$^o$.”*

*“For the upper level, WRF simulated 500 hPa horizontal winds were compared against radiosonde measurements from 90 stations obtained from the NOAA/ESRL radiosonde database (https://ruc.noaa.gov/raobs/). Since most stations release balloon at 00:00 and 12:00 UTC, WRF winds were compared against the radiosonde measurements at a 12 hour interval during the 24 hour simulation period. The results are shown in Figure 7: RMSE of wind speed is 2.54, 4.0, and 5.11 m s$^{-1}$, at 2 June 00:00 UTC, 2 June 5 12:00 UTC, and 3 June 00:00 UTC, respectively. Wind direction difference between WRF and radiosonde is 11.5$^o$, 16.4$^o$, and 19.1$^o$ at the three times. Locations of the weather stations and radiosonde sites used in the evaluations can be found in the supplement document.”*

*“The above-described evaluations using in-situ measurements indicate that the meteorological simulation is of adequate accuracy for the pseudo observation based inverse modeling tests conducted in this paper. Future applications with true observational data will need to quantify the simulation error in the 4DVar system.”*

[Figure]

Figure 1. Locations of the meteorological stations used for evaluating the WRF simulation. The bold dark outline marks the boundary of the WRF simulation domain. The surface meteorology data are obtained from NOAA Integrated Surface Dataset. (Figure 6 of the supplement document).

[Figure]

Figure 2. Histogram of the 10m wind speed difference (a) and wind direction difference (b) between WRF simulation and surface meteorological station measurements. This is Figure 6 of the revised manuscript.

[Figure]

Figure 3. Locations of the radiosonde stations used for evaluating the WRF simulation. The bold dark outline marks the boundary of the WRF simulation domain. The radiosonde data are obtained from NOAA/ESRL radiosonde database. (Figure 7 of the supplement document).

[Figure]

Figure 4. Comparison of 500hPa wind speed and wind direction between WRF simulation and radiosonde measurements. Figures (a) and (b) are the comparisons at 00:00 UTC, 02 June 2011,  Figures (c) and (d) are at 12:00 UCT 02 June 2011, and Figures (e) and (f) are at 00:00 UTC 03 June 2011. RMSE and relative error (RE) for wind speed and mean difference in wind direction are shown in each figure.  (This is Figure 7 of the revised manuscript.

*Moreover, directly downscaling IC from coarse resolution datasets (especially that the land states were from a significantly different Noah LSM, on coarser-resolution land surface model than the PX LSM used in this work) is problematic and can lead to very poor WRF performance. Although the authors are aware of this issue, they failed to clearly discuss the impacts.*

Response

Thanks for reminding us. The following statements are added to the Model setup section (section 3.1.4, immediately preceding our new documentation of the quantitative accuracy of the WRF simulation.)

*"In the model setup, the initial and boundary meteorological conditions are generated by downscaling the CFSv2 data. Downscaling coarse resolution global reanalysis data could lead to poor WRF performance. Although this potential problem is not a concern for the present pseudo data based inversion experiments, it must be properly treated in the future application with true observation data. Error in the initial condition will lead to erroneous emission source attribution, especially for inversions with short assimilation window."*

*2) The authors' response to my general comment 1: I don't get why offline model runs cannot achieve as fine grid spacing as online runs do. Can you elaborate on this?*

Response:

First, we would like to clarify that we did not mean the offline model cannot employ fine grid spacing comparable to online runs. Our last response was not clearly worded, and we apologize for the confusion. We intended to state the that: because the offline model's driving meteorology fields (wind speed, wind direction, and PBL height ...) are often provided by reanalysis data, which are of a fixed grid spacing. Attempting to run an offline model with substantially finer grid spacing than the driving meteorology dataset does not make much sense, since the spatial downscaling (interpolation) is not able to introduce the small scale atmospheric processes that are not resolved by the original reanalysis meteorology data.

With this said, we do agree with the referee that an offline model can be run at the same fine grid spacing as an online model, if the required meteorology data used for initial and boundary conditions are provided by an NWP (numerical weather prediction) model or reanalysis at the same grid spacing.

*I sort of agree with the statement related to timestep, but the authors should provide a quantitative estimate of such impact on offline vs online, e.g., based on literature review.*

Response:

To provide a quantitative estimate of the difference between offline vs. online transport models, the following statements are added to the introduction section.

*"In the WRF system, CO2 mixing ratio variations can impact the meteorology fields through the radiation scheme. This feedback is ignored inWRF-CO2 4DVar and instead CO2 is modeled as a passive tracer.WRF-CO2 4DVar is designed for regional inversions with a short assimilation window (days to weeks). For such applications, the potential feedback of CO2 variation on the meteorology is insignificant for most cases. Including such impacts would require a large mount of new code development while offering very limited performance improvement. By ignoring the CO2 feedback, WRF-CO2 4DVar can use the full version of most WRF physics schemes in its tangent linear and adjoint models to minimize the linearization error (Tremolet, 2004)."*

*"Compared with offline regional inversion systems, WRF-CO2 4DVar has an advantage provided by the close one-way coupling between meteorological processes and chemistry transport. For example, adequately resolving CO2 vertical transport in high resolution regional simulations is crucial and vertical motions in the atmosphere exhibit significant temporal variability. Grell et al. (2004) shows that less than 40% of the total vertical velocity variability in a 3 km resolution simulation is captured by 1-hour output interval. He estimate that the meteorological output interval must be less than 10 minutes in order to capture more than 85% of the variability in cloud resolving simulations. In WRF-CO2 4DVar, CO2 transport runs at the same time step as the meteorology, avoiding the problems facing its offline counterparts."*

*3) The authors' response to my additional comments 7: Not sure why you cannot calculate trajectories using the tools you listed (e.g., HYSPLIT), using WRF fields? Several publications demonstrate adjoint along with Lagrangian analysis to provide a comprehensive view of the source-receptor relationships.*

Response:

We misinterpreted the referee's original comment: we thought the referee was asking us to generate backward trajectories using the WRFCO2 adjoint code. Realizing that the referee is asking us to calculate trajectories using a Lagrangian transport model driven by the WRF meteorology, we have now done so. The following paragraph has been added to "spatial pattern of adjoint sensitivities" (Section 3.3), along with Fig. 11.

"To provide a comparative view of the source-receptor relations, backward trajectories of particles released from the Centerville and WLEF sites were also calculated using the Lagrangian model HYSPLIT (Stein et al., 2015). WRF-CO2 forward model simulated meteorology saved at 1-hour intervals is used to drive the HYSPLIT trajectory calculations. Two sets of simulations were carried out for each of the two tower sites: particles were released from the height approximates the first vertical level in WRF in the first set, and from the tenth level in the second. Each set of simulation consists of 300 particles, and their starting locations were uniformly distributed within a 48x48 km grid box corresponding with the WRF grid spacing. The HYSPLIT backward trajectory simulation results wre plotted in Fig. 11. To avoid cluttering the figures, the trajectories are plotted using only 60 out the 300 particles released in each case. The backward trajectories demonstrate similar spatial patterns as the adjoint sensitivities (Fig. 10). On the other hand, there are discernible differences between the two, which can be attributed to that representation of some tracer transport processes in HYSPLIT that are different from WRF, especially diffusion and convective transport."

[Figure]

Figure 5. The Backward trajectories of particles released from Centerville, Iowa at the first (a) and 10th (b) vertical level, and WLEF, Wisconsin at the first (c) and 10th (d) vertical level. (This is Figure 11 of the revised manuscript.)

*My last complaint is on the presentation of this manuscript and the rebuttal. Normally introduction should include what's been done in prior works and the limitations which will be addressed in this work. Referred articles in the results/discussion section should be closely related to the point to make. However, a portion of introductions and discussions are unrelated to goals and highlights of this work, so they are actually confusing and a bit misleading. The documents (especially the rebuttal) contain so many grammar mistakes and are hard to read. Careful proofreading is needed.*

Response:

We think the presentation of the revised manuscript is of good quality, but we did not do a good job checking our grammar in our rebuttal to the first referee. We apologize and have exercise greater care in writing this response and the accompanied revision. We have revised sections of the manuscript in response to the referee's specific suggestions, as previously presented in this document. Specifically, in the introduction section, a statement about the rationale for developing the adjoint code of WRF-$CO_2$ without considering the chemistry feedback is added. In the model setup section, a discussion of the potential problem of downscaling coarse resolution global scale reanalysis for initial and boundary conditions is added, along with an improved presentation of accuracy evaluation of the WRF simulation. In the summary and outlook section, a discussion about future code development is now included. Lacking specific comments by the referee related to the material in the introduction and discussion, we are unable to ascertain exactly where additional improvements can be made.

[revised manuscript text omitted]

---

## Author Response (AR3)

Authors' Response to Reviewer Comments

The paper presents a newly developed WRF-CO2 4DVar assimilation system. The authors have documented the full system and provided a reasonable demonstration of the system. The system is well-documented and based on known optimization methods. The adjoint model has been carefully developed and implemented for tracer applications, and therefore provide an important step toward a full 4DVar system for regional inversions of fluxes. The main concerns and technical comments listed hereafter need to be addressed before final publication.

The reviewer has provided very insightful and constructive comments about our manuscript and the model code. We greatly appreciate the reviewer's time and effort for helping us improve our work. We have carefully considered the reviewer's comments and conducted further code development, simulation experiments, and manuscript revision to address these comments. This revision has substantially enhanced our understanding of the broad topic of atmospheric tracer data assimilation, in addition to improving the quality of this manuscript and our model code. For this, we express our sincere gratitude toward the reviewer. Detailed point-by-point responses are given following each comment below.

Note: all the references used in this response can be found in the reference of the revised manuscript.

**General comments:**

[1]- Missing definition of error covariances (both observations and prior flux errors): Here the system has been described and tested without any details on the construction of error covariances. The inverse framework needs to address that problem. The authors should include a paragraph about it.

In addition, one of the tests used an incorrect observation error covariance matrix R, inconsistent between the fully correlated errors (bias of 50% everywhere) and the R matrix (diagonal terms only - no correlation). The test should reconcile or address this problem.

The reviewer's comment on error matrices improved our understanding of this critical issue, especially with regards to real-data applications. The inverse modeling tests we used in this paper follow the same approach of Henze et al. (2007) regarding the error matrix specification and error-free synthetic data. But we failed to provide a clear justification for using the idealized error specification and a discussion of the need for more vigorous error matrix treatment in real-data applications. To address this, we have completely rewritten the inverse modeling section to: (1) provide a clear explanation of the experiment designs, with details about the **B** and **R** specification in light of the error-free synthetic data used in our inverse modeling tests. (2) A review section of error specification for real-data applications from the past studies. Please see Section 3.4.1 (Page 16 line 5- Page 17 line 26) of the revised manuscript for the detailed changes.

Regarding the reviewer's concern of the incorrect error covariance matrix R used in the Case 1 inverse modeling test (case 1: prior flux scaling factor set to 1.5 at all surface grid points),

we believe it was caused by our original manuscript's lack of clear description of the inverse modeling test setup. To address this issue, we provide a detailed explanation below and in the revised manuscript (Section 3.4.1). First, the synthetic data inversions used in our paper use a highly idealized configuration for the sole purpose of testing our model code accuracy. This is the same approach used in Henze et al. (2007) for testing the GEOS-Chem adjoint model. In this aspect, our inverse modeling tests are different from the synthetic data inversions typically used in observation system simulation experiments (OSSE). In OSSE type inversions, synthetic data are generated by a transport model driven by a true flux, then errors are added to the model simulated atmospheric $CO_2$ before being used to constrain the inversion. This process of generating synthetic data for OSSE inversion is described in detail in Chevallier et al. (2007). Since the synthetic observations contain errors, the observation error matrix must properly represent the error (both the variance and covariance) in this type of synthetic data inversions.

In comparison, the inverse modeling tests used in our paper are designed to test the adjoint model and optimization code accuracy, instead of an OSSE type synthetic inversion. This leads to a key difference in our inverse modeling test setup: *no errors were added to the forward model simulated $CO_2$*. Because the synthetic-observations contain no error, the observation error matrix **R** has no error correlation. The idea is to start with any prior flux, constrained by the error-free synthetic observations, the 4DVar system should converge to the true flux, thus confirming our assimilation system has been correctly implemented. In addition to the error-free observations, another key feature is that the background error matrix **B** is set to infinity (by setting $\mathbf{B}^{-1}=0$ ) and the observation error **R** is set to the identity matrix. The combination of the three features (error-free observations , $\mathbf{B}^{-1}=0$, and **R**=**I**), means that: (1) the background cost function is always zero: deviations of the analysis from the prior fluxes will incur no cost because $\mathbf{B}^{-1}=0$. (2) All the synthetic observations are given the same weight in the observation cost function because (**R**=**I**). (3) The 4DVar system is driven by the synthetic observations only, and the system iteratively decreases the cost function by moving the analysis from a given first guess flux to the true flux. Because deviations from the prior flux incur no cost and the observations contain no error, the system will eventually converge to the true flux. This is highly idealized setting, but is an effective way to examine assimilation system code accuracy, as done in Henze et al. (2007).

Given the above descriptions, we consider it is reasonable to use the same error configuration ($\mathbf{B}^{-1}=0$ and **R**=**I**) for the two cases of inversions (Case 1 and 2). Because both inversions were constrained by the same set of error-free synthetic-observations, **R**=**I** is appropriate to assign all observations the same weight in cost function calculations. As the reviewer correctly pointed out that for Case 1, the first guess fluxes is perfectly correlated because the flux scaling factor is set to 1.5 for all surface grid points, thus all the off-diagonal elements of the background error matrix **B** (not the observation error matrix **R**) should be set to 1.0. However, this detail in **B** becomes irrelevant because **B** is set to infinity so that analysis deviation will incur no cost (so that the analysis can converge to the true flux). We hope this will provide a satisfactory response to the reviewer's concern. Please also see the rewritten Section 3.4.1 (Page 16 line 5- Page 16 line 27) of the revised manuscript for more details.

**[2]**- Boundary conditions: The inversion system needs to address this problem which is not trivial and could increase significantly the cost of the 4DVar. In addition, the optimization becomes more complicated as the two unknowns (boundary inflow and surface fluxes) could be optimized to correct for concentrations. The authors need to address that problem, and if not solving for, should describe a path toward implementing this part of the code. At this point, the WRF-CO2 4DVar is not usable for real-data inversions because of this gap. This is a major weakness in the current study.

We thank the reviewer for pointing out the importance of the tracer lateral boundary condition for our regional inversion system. To address this issue, we further developed our current 4DVar system to include the capacity of tracer lateral boundary condition optimization. The new code development includes: (a) the adjoint and tangent linear code of the tracer lateral boundary condition, and (b) expanding the two variational optimization schemes to accommodate the lateral boundary conditions in the state vector. The tracer lateral boundary condition adjoint and tangent linear code development are rather straightforward because they share the same code for dynamic (advection/diffusion), physics (ACM2 PBL) and chemistry (convective tracer transport in the chemistry module) with the surface flux code already developed in WRF-CO2 4DVar. Please note that WRF-Chem system does not use the relaxed boundary zone for chemistry variables (as compared to other scalar variables), and we followed the same treatment in the adjoint code development.

We have confirmed the accuracy of the tangent linear and adjoint code of the lateral boundary condition. However, we have not used the new code to test the combined boundary condition and flux optimization. As the reviewer pointed out, including lateral boundary conditions in the inversion is not trivial because it greatly increases the size of the state vector. Based on past regional inversion studies, we recognize that it may not be practical to optimize the lateral boundary conditions at the model grid resolution, and thus aggregation (both spatial and temporal) of the boundary condition scaling factor is necessary. Given this consideration, we implemented a mechanism in WRF-CO2 4DVar to allow a flexible mapping from the 3d lateral boundaries into the 1d state vector. This mapping gives the user the flexibility to determine whether and how to aggregate the lateral boundary condition scaling factors. While this new code has not been tested for the inverse modeling experiments in this manuscript, we do plan to test it in a follow-up study.

The description of the tracer lateral boundary condition code development is in Section 4 of the revised manuscript (Page 18 line 15-Page 20 line 2). We have reposted the WRF-CO2 4DVar code and it now includes the update with the tracer lateral boundary conditions.

 **[3]**- Adjoint evaluation: Instead of a qualitative comparison of adjoint sensitivity and back-trajectories, the authors can compute the actual HYSPLIT footprints, and combine them with prior fluxes to compare to the adjoint sensitivity results. This analysis is fast considering the short simulation period and that the tools to compute tower footprints are publicly available. This evaluation would reinforce the confidence in the adjoint model.

We appreciate the reviewer's suggestion regarding the comparison. Through conducting further simulations, footprint calculations, and comparisons, our understanding of Eulerian vs.

Lagrangian tracer transport has been substantially enhanced, in addition to the improvement of the present manuscript.

In calculating the HYSPLIT footprint, we released 30,000 particles from each receptor location. Our sensitivity tests found that further increasing the particle number has negligible impact on the resulting footprint calculation (results not shown). All the backward trajectories used in calculating the footprint (Fig. 1 below and Fig. 11 of the revised text) were simulated with HYSPLIT using the PBL heights provide by the WRF generated meteorological data.

We summarized the new simulations and compared the resulting HYSPLIT footprints with that calculated by WRF-CO2 adjoint model (Fig. 2 below and Fig. 11 in the revised text). Note the adjoint sensitivities have been redrawn in a raster format (from the contours used in the original manuscript) to facilitate comparison with the HYSPLIT footprints. Since our adjoint model accuracy has been confirmed in this paper and the HYSPLIT model has been used in numerous studies, we provide a discussion for attributing the differences between the two sets of footprints. We believe that these differences are at least partially caused by: (1) the finite difference advection of WRF and Lagrangian advection of HYSPLIT. (2) The different treatment of vertical mixing in the boundary layer.

Through working through this part of the revision suggested by the reviewer, we realized that the detailed comparison between an Eulerian adjoint model and a Lagrangial model is of vital importance for real-data applications, but such comparisons are not been well documented in the literature. We acknowledge that our current simulations and analysis could be further improved regarding the impact of PBL and cumulus schemes, and advections. We plan to carry out further investigation along this direction to help us better understand and treat transport model errors for CO2 inversions in a follow-up work.

The above described HYSPLIT footprint calculations and comparison can be found at Page 15 Line 13 – Page 16 Line 2 of the revised manuscript.

[Figure]

Figure 1. Footprints calculated using HYSPLIT backward trajectories for the four receptors (two vertical levels at each of the two tower sites). The black cross marks the tower location. These HYSPLIT footprints are calculated and plotted on the same grid and color scale as their WRF-CO2 adjoint sensitivity counterparts (Fig 2 next page).
Note: this figure is Fig. 11 of the revised manuscript.

[Figure]

Figure 2. Footprint calculated by the WRF-CO2 adjoint model at four receptor locations (two vertical levels at each of the two tower locations). The black cross in each figure marks the location of the tower. Each footprint is plotted using same color scale with its HYSPLIT counterpart in Fig. 1 (last page). Note: this figure is Fig. 10 of the revised manuscript.

**Technical comments:**

[1] P1-L16-21: This paragraph needs to include more references. Only two papers are cited, somehow arbitrarily. Please add the references corresponding to the different statements. Chevallier et al., 2005 is a good reference but some of the major papers should also be cited here.

This paragraph has been completely rewritten to provide an organized review of the atmospheric CO2 inversion studies. Please see page 1 lines 16-24 of the revised text for details.

[2] P1-L22: An ensemble approach would also need a CTM. Correct the statement.

This sentence has been corrected in the revised manuscript (Page 2 line 3-4).

[3] P1-L23: What about the boundary conditions? You are developing a regional system within a bounded domain. The state vector also includes prior information related to the boundary inflow. Revise the sentence.

The sentence has been revised to include the CO2 lateral boundary condition in the state vector. Please see the page 2 line 1-2 of the revised manuscript. Regarding tracer lateral boundary conditions, please also refer to our response to general comment [1].

[4] P2-L1: Same comment related to boundary conditions. Some bibliography on this problem is needed here. Background conditions are critical to limited-domain inversions.

Code development for the tracer lateral boundary conditions and related literature reviews has been added in a new Section 4 (Page 18 line 15-Page 20 line 2 of the revised manuscript).

[5] P2-L2-3: Why do you bring the dimensions of the Jacobian in the introduction while not discussing it further in this paragraph?

The sentence about Jacobian matrix dimensions has been deleted.

[6] P2-L5-6: THis is only true for single tracer simulations in Eulerian mode. One can run multiple tracers in one simulation, or run millions of particles at once in a Lagrangian framework. Please revise this statement.

This sentence has been corrected. Please see page 2 line 10-14 of the revised text for details.

[7] P2-L6-11: The idea here is to explain the advantage of using a variational approach. While the arguments are correct, the justification lacks some clarity and a more rigorous description of the variational approach (minimzation using the gradients, adjoint model,...). There are also some disadvantages to the variational approach that should be stated here (no explicit posterior uncertainty, problem of convergence,...). This is an important paragraph that highlights the value of your work. You should revise this text to explain the major technical features in variational methods, and link it with the previous examples of 3d/4d-var studies in the following paragraphs.

The description of 4DVar assimilation has been rewritten to include the reviewer's suggestions. The new paragraph is in page 2 line 21-28 of the revised text.

[8] P2-L19: CarbonTracker is using a lagged Ensemble Kalman Smoother, not a variational approach. Revise the statement.
Thanks for pointing this out. CarbonTracker has been removed from the list of 4DVar

assimilations. It is discussed as an example for ensemble Kalman smoother in the revised manuscript (page 1 line 22, and page 2 line 16-19).

[9] P2-L13-26: this paragraph focuses entirely on 4d-var systems. What about other variational approaches?

Regarding other variational approaches: although there are a number of studies using 3DVar assimilation to estimate atmospheric chemistry mixing ratios, we have not found any 3DVar inversion studies for CO2 surface fluxes. This is most likely because 3DVar does not include a CTM and thus can not link fluxes to atmospheric mixing ratio. To strengthen this section, we added discussion about ensemble Kalman filter smoother (see our response to technical comment [8]). Page 2 line 16-19 of the revised manuscript.

[10] P2-29: More examples are needed here. Gerbig et al. (2009) is not an inversion but an overview on regional inverse modeling strategies. Include previous regional inversion studies with their achievements.

This has been fixed. In the revised text, a number of references to regional inverse modeling are given in page 3 line 10-24.

[11] P2-L31: GEOS-Chem is a global system not optimla for regional applications. Multiple examples of regional inversions have been published over the years.

GEOS-Chem reference has been removed from the regional inversion application list.

[12] P2-L34: Include references for LPDM-based inversion studies.

References to Lagrangian Particle Dispersion Model based inversion are in Page 3 Line 10-24 of the revised manuscript.

[13] P2-L35: "assimilated meteorology". Do you mean "meteorological analyses"?

"assimilated meteorology" has been corrected as "meteorological analyses"

[14] P3-L1-10: These are the studies you need to describe up front. This paragraph should be merged with the previous one.

The description about the past global and regional inversion studies have been moved together. Page 2 line 30- Page 3 line 24 of the revised manuscript.

[15] P3-L12: This statement should come before describing inversions and methods.

This statement has been modified and moved to the beginning of the first paragraph (page 1 line 16-17 of the revised manuscript).

[16] P3-L13: The differences in CMS products is an illustration but other examples would be better suited here. Consider inter-comparison studies in particular (e.g. Peylin et al. (2013) for CO2).

The CMS reference has been replaced with inversion inter-comparison studies including (Gurney et al 2002 and Peylin et al 2013), and the sentence has moved to page 1 line 16-17 in the revised manuscript.

[17] P3-L14-15: "high priority". Confusing. What do you mean here? Please rephrase.

Considering that the need for developing a high resolution 4DVar system has been made in other statements, this sentence is removed in the revised manuscript.

[18] P3-L29: The problem of convective transport for tracers is not trivial, and most of the convective schemes in WRF do not even produce mass fluxes explicitly. Some of them use a simple parameterization with 1D variables while other schemes only provide mass fluxes at the cloud base/top which is insufficient for convective mass transport. Few of the schemes have a full 3D mass flux, and mass conservation is not even guaranteed. Please explain with more details here which schemes would allow for convective transport of tracers with "limited new code development".

We agree with the reviewer's opinion about the importance of the tracer convective transport. To address this issue, the convective transport in Section 2.4.3 has been strengthened into a separate paragraph, which now includes details as suggested by the reviewer. In the revised text, we point out the following facts about the WRF-Chem tracer convective transport: (1) the tracer convective transport in WRF-Chem is treated by a simplified Grell convective scheme in the chemistry module, instead of by the cumulus schemes in the physics package. (2) This simplified Grell scheme for tracer convective transport uses convective precipitation rain rate to scale the base mass flux. This is a rather crude presentation comparing to the stochastic ensemble closure used in Grell and Freitas (2014).

Please see page 10, line 16-25 of the revised manuscript for the detailed changes.

[19] P3-L34: Justify this statement with a quantity or a reference.

Two references have been added justify this statement.

Please see page 4, line 9-10 of the revised manuscript for the detailed change.

[20] P4-L1: Typo. Subscript for "2" missing in "CO2".

Fixed.

[21] P4-L1-4: Provide a simple quantity instead of several sentences. For example, calculate the

impact on the solar energy when considering higher CO2 concentrations at the local scale. Or cite a study describing it.

To provide a quantitative assessment of $CO_2$ impacts on meteorology for short term simulations, we conducted the following sensitivity study. Using the model set up as described in Section 3.1 of the manuscript, two 48-hours simulations were conducted. The $CO_2$ mixing ratio is set to 391 ppm in the first and 500 ppm in the second simulation. Because the two simulations are identical otherwise, differences in meteorology fields were attributed to the $CO_2$ difference. The results show that magnitude of difference in horizontal wind and temperature are very small: Mean/standard deviation of U, V, and T at the end of the 48 hours simulation are 0.0794/0.1408 m/s, 0.0791/0.1459 m/s, and 0.0366/0.0614 K.

The above sensitivity results are summarized in page 4 line 9 -14 of the revised text.

Note: Radiative schemes in WRF-Chem do not use the predicted $CO_2$ mixing ratio (from the chemistry module) for radiative transfer calculations, they instead use either fixed values or interpolated climatological values.

[22] P4-L10: Typo. "estimated"

Fixed.

[23] P4-L6-12: This is even more relevant for turbulence with eddy turn-over times in minutes.

Thanks for pointing this out. The sentence has been revised to include the importance of online chemistry transport regarding the turbulence mixing. Page 4 line 20 of the revised manuscript.

[24] P4-L22: Replace "observational data" by "atmospheric observations" or "observations"

Changed to "observations"

[25] P4-L22 (also L23): Add "the" to "flux", or replace "flux" by "fluxes".

Changed to "the CO2 flux".

[26] P4-L25: Replace "emission" by "fluxes". "emissions" refer to positive fluxes only.

"emission" has been replaced by "fluxes" here and a number of other places in the text to avoid confusion.

[27] P5-Eq 2-4: Describe the variables used in the equation (x, x0, y, H, and M) and refer to Table 1 for the full list of symbols.

A description of variables used in the equations and reference to Table 1 have been added as suggested by the reviewer. Page 5 Line 19-23 of the revised manuscript.

[28] P5-L10: Why "essentially"? Is it something else?

The word "essentially" has been removed from the sentence.

[29] P5-L11: Replace "emission" by "flux"

Fixed.

[30] P5-L12: Is a lagged approach? Or independent flux estimates? Refer to later sections to describe the approach.

WRF-CO2 4DVar is implemented for independent flux estimates. Reference to later sections for detailed descriptions has been added.

[31] P7-L17: Replace "CO2 related processes" by "physical and dynamical processes involved in the atmospheric transport of CO2"

The phrase has been replaced as suggested.  Please see page 8 line 1 of the revised manuscript.

[32] P7-L20: Typo. "to keep"

Fixed.

[33] P7-L22: To be clear for the readers who are not familiar with WRF, the chemistry module was added, which is more accurate than stating that WRF-Chem "replaced" WRF. WRF is still used in WRF-Chem.

The sentence has been rephrased as the reviewer suggested (Page 8 line 2-3 of the revised manuscript).

[34] P7-L24-25: The tracer here has no impact on the code. One can substitute CO2 by CO or CH4. Assuming there is no chemistry involved, the transport of an inert and mass-free tracer has no incidence on the selected gas. Revise the sentence.

The sentence has been revised as suggested by the reviewer. Page 8 Line 5-9 of the revised manuscript.

[35] P7-L27: It means that you removed the GHG option and used a passive tracer. It would be clear to the future users if you state that. WRF-CO2 assumes that VPRM is used for the biogenic component. Otherwise it becomes equivalent to the original passive tracer mode. Clarify.

The sentence has been revised to clarify that VPRM is not used and CO2 is treated as a passive tracer. Please see page 8 line 5-9 of the revised text.

[36] P8-L25: Typo. "was"

Fixed.

[37] Section 2.4.2: This section is clear and helpful for future code development. One comment

here: Additional thoughts on joint optimization of meteo and CO2 data would be useful. How could a combined assimilation be implemented using your 4DVar system?

Thanks. A paragraph has been added about possible extension to a joint optimization of meteorology and $CO_2$ fluxes. This new paragraph (page 13, line 1-19 of the revised manuscript) is placed after the WRF meteorology field validation and discussed in the context of addressing transport error and bias for $CO_2$ inversion. This new paragraph discusses the possibility of using a combined meteorology and $CO_2$ 4DVar inversion to correct transport model errors. It also points out that if $CO_2$ feedback on meteorology is not considered (which is insignificant for short term simulation, please see our response to technical comment [21]), the primary code modification for expanding to such a joint 4DVar system will be in the optimization schemes, because the adjoint code for meteorology (in WRFPLUS) and $CO_2$ (WRF-CO2 4DVar) has already been implemented.

Please see Page 13 Line 1-19 of the revised manuscript for the detailed change.

[38] P9-L29: The argument to select the PBL scheme is understandable, but lacks some scientific background. Is ACM2 a good option for CO2 turbulent mixing in the PBL? Several schemes have been tested with known systematic errors. The selection process should be based on model performances rather than technical reasons. You should at least comment on the model performances of the ACM2 PBL scheme. Add some metrics or published studies to assess the ACM2 schemes.

The justification for choosing ACM2 for PBL parameterization has been strengthened and forms a separate paragraph in the revised manuscript. It includes a brief description of ACM2 and its performance assessment in the published literature.

Please see page 10 line 4-14 of the revised manuscript for the detailed changes.

[39] P9-L31: To clarify this sentence, you need to explain that this scheme is parameterized based on precipitation rate. It is not the actuall mass flux from the Grell scheme but rather a crude representation of the vertical convective transport. Clarify in the text.

This has been clarified as suggested by the reviewer. Please see page 10 line 19-21 of the revised manuscript.

[40] P10-L27: "biospheric"

Fixed.

[41] P10-L27: Biospheric fluxes vary diurnally from negative to positive values depending on the time of day. For this reason, inversion problems need to address separately the two components or by time of day (night versus day) or by component (respiration versus photosynthesis). The daily mean will be irrelevant when transported into the concentration space as the timing of the atmospheric mixing is coupled to the timing of the fluxes and therefore cannot be simply averaged over an entire day. The 4DVar could be used for 3-hourly fluxes,

which would be more accurate and avoid biases due to day/night components. This problem has been discussed in several papers (e.g. Gourdji et al., 2012 - https://www.biogeosciences.net/9/457/2012/bg-9-457-2012.html). Whereas this problem will not be critical in a pseudo-data study, it will be critical in real-data inversions.

We agree with the reviewer about separating the biospheric fluxes by time in inversion. The present implementation of WRF-CO2 4DVar allows the flexibility for choosing the temporal resolution of fluxes inversion: the fluxes can be optimized at the same temporal resolution as the flux data allow. For instance, as pointed out by the reviewer, the optimization can be carried out on 3-hourly fluxes when using the CarbonTracker fluxes as prior. We added texts to clarify this issue and to emphasize that for real-data application, time varying fluxes should be used.

Please see Page 11 Line 18-23 of the revised manuscript for the detailed change.

[42] P10-L31 to P11-L4: The definition of boundary conditions by a global model is highly uncertain and cannot be ignored in the optimization process. Several studies have discussed that problem (Trusilova et al., 2010; Schuh et al., 2010; Goeckede et al., 2011; Lauvaux et al., 2012). See the general comment. If the problem is not solved in this paper, it should be highlighted as a major shortcoming in this study.

We agree with the reviewer that it is imperative for a regional inversion to address the boundary condition. In response, we have developed this part of code. Please refer to our detailed response to major comment [2].

[43] Figure 5: Conventional contour lines to illustrate pressure systems and frontal structures would be much easier to catch for the readers.

Figure 5 has been replotted following the reviewer's suggestion. To avoid clustering the figure, the state boundaries are not included.

[Figure]

Figure. Sea level pressure (hPa) and horizontal wind (m s⁻¹) at models's lowest vertical level plotted during the 24-hour simulation starting at 2011-06-02 UTC

[44] P11-L22: Not only variances but covariances as well.

Thanks! Corrected.

[45] P11-L29 to P12-L7: This evaluation is limited to wind speed and direction but is not directly representative of CO2 concentration errors. Clarify here how these errors could be used to inform about transport model error variances and covariances.
From Figure 6, the wind speed is biased high, which is consistent with other studies. How would this problem be considered in the 4DVar framework?

Errors in meteorology fields will cause transport model error, which is a major part of the data-model mismatch error. For real observation applications, transport model errors are specified as part of the observation error covariance matrix R. Large transport error should be represented by higher variance/covariance in R to reduce the weight of the data-observation mismatch.

We think the wind bias can be addressed by nudging the wind field toward the observations. For instance, Gupta et al. (1997) found that nudging model simulated winds in the boundary layer to the radar wind profiles substantially improved estimates of plume dispersion. Another approach for addressing meteorology simulation bias will be a joint meteorology and CO2 assimilation. We lay out a brief pathway toward expanding our current system to such a joint meteorology/CO2 4DVar assimilation system in the future.

The above statements has been incorporated in the revised manuscript (Page 13 Line 1-19). Please also see our response to technical response [37].

[46] Section 3.2: The evaluation is convincing and seems to confirm that the adjoint has been correctly implemented. but please explain why the differences are considered "accurate". Cite other references for the threshold. For example: why is "10^-10" acceptable?

Citations to other adjoint code development accuracy assessments have been added for reference in the revised text. The added citations include: Henzen et al (2007) for GOES-Chem 4DVar system and Meirink et al. (2008) for TM5 4DVar system.

[47] Section 3.3: This section remains qualitative and not highly informative. The adjoint sensitivity seems to agree with the overall shape of the footprint. This comparison would be more convincing if the footprints were computed using HYSPLIT and combined with the prior fluxes. A short simulation (24 hours in your case) is really inexpensive for a Lagrangian model and would provide an independent evaluation of your adjoint transport.

This technical comment is the same as the major comment [3]. Please see our detailed response in the major comment section.

[48] P15-L1: The definition of R is inconsistent for case 1. If all pixels are multiplied by 1.5, the error correlations are equal to 1 over the entire domain. But you defined the R matrix with an independent error space. Clarify.

This technical comment is the same as the major comment [1]. Please see our detailed response in the major comment section.

[revised manuscript text omitted]